



# Exploring hydrologic post-processing of ensemble streamflow forecasts based on Affine kernel dressing and Nondominated sorting genetic algorithm II

Jing Xu[1], François Anctil[1], and Marie-Amélie Boucher[2]

[1]Department of Civil and Water Engineering, Université Laval, 1065 avenue de la Médecine, Québec, Québec, Canada;
[2]Department of Civil and Building Engineering, Université de Sherbrooke, 2500 Boul. de l'Université, Sherbrooke, Québec, Canada

**Correspondence:** Jing Xu (jing.xu.1@ulaval.ca)

**Abstract.** Forecast uncertainties are unfortunately inevitable when conducting the deterministic analysis of a dynamical system. The cascade of uncertainty originates from different components of the forecasting chain, such as the chaotic nature of the atmosphere, various initial conditions and boundaries, inappropriate conceptual hydrologic modelling, and the inconsistent stationarity assumption in a changing environment. Ensemble forecasting proves to be a powerful tool to represent error

growth in the dynamical system and to capture the uncertainties associated with different sources. However, space still exists for improving their predictive skill and credibility through proper hydrologic post-processing. We tested the post-processing skills of Affine kernel dressing (AKD) and Non-dominated sorting genetic algorithm II (NSGA-II). Those two methods are theoretically/technically distinct, yet however share the same feature that both of them relax the parametric assumption of the underlying distribution of the data (i.e., streamflow ensemble forecast). AKD transformed ensemble and the Pareto fronts

generated with NSGA-II demonstrated the superiority of post-processed ensemble in efficiently eliminating forecast biases and maintaining a proper dispersion with the increasing forecasting horizon.

**Keywords:** Hydrologic ensemble prediction systems (H-EPS), Hydrologic post-processing, Affine kernel dressing (AKD), Evolutionary multiobjective optimization, Non-dominated sorting genetic algorithm II (NSGA-II).

## 1 Introduction

Hydrologic forecasting is crucial for flood warning and mitigation (e.g., Shim and Fontane, 2002; Cheng and Chau, 2004), water supply operation and reservoir management (e.g., Datta and Burges, 1984; Coulibaly et al., 2000; Boucher et al., 2011), navigation, and other related activities. Hydrologic models are typically driven by dynamic meteorological models in order to issue forecasts over a medium range horizon of 2 to 15 days (Cloke and Pappenberger, 2009). Some predictive uncertainty is then inevitable given the limits of knowledge and available information (Ajami et al., 2007). In fact, those uncertainties

occur all along the different steps of the hydrometeorological modelling chain (e.g., Liu and Gupta, 2007; Beven and Binley, 2014). These different sources of uncertainty are related to deficiencies in the meteorological forcing, mis-specified hydrologic initial and boundary conditions, inherent hydrologic model structure errors, and biased estimated parameters (e.g., Vrugt and



Robinson, 2007; Ajami et al., 2007; Salamon and Feyen, 2010; Thiboult et al., 2016). Substantive theories are proposed in order to represent and reduce the different sources of uncertainties and to enhance forecasting skill.

The superiority of ensemble forecasting systems in quantifying the propagation of predictive uncertainties (over deterministic systems) is now well established (e.g., Cloke and Pappenberger, 2009; Palmer, 2002; Seo et al., 2006; Velázquez et al., 2009; Abaza et al., 2013; Wetterhall et al., 2013; Madadgar et al., 2014). Meteorological ensemble prediction systems (M-EPSs) (e.g., Palmer, 1993; Houtekamer et al., 1996; Toth and Kalnay, 1997) are operated worldwide by national agencies such as the European Centre for Medium-Range Weather Forecasts (ECMWF), the National Center for Environmental Prediction (NCEP), 30    the Meteorological Service of Canada (MSC), and more.

Sequential data assimilation techniques, such as the particle filter (e.g., Moradkhani et al., 2012; Thirel et al., 2013) and the ensemble Kalman filter (e.g., Evensen , 1994; Reichle et al., 2002; Moradkhani et al, 2005; McMillan et al., 2013) provide an ensemble of possible re-initializations of the initial conditions, expressed in the hydrologic model as state variables, such as soil moisture, groundwater level and so on. Besides, hydrologic models bring their own load of uncertainties because of 35    simplifications and limited knowledge.

Multimodel schemes were proposed to increase performance and decipher structural uncertainty (e.g., Duan et al., 2007; Fisher et al., 2008; Weigel et al., 2008; Najafi et al., 2011; Velázquez et al., 2011; Marty et al., 2015; Mockler et al., 2016). Thiboult et al. (2016) compared many H-EPS, accounting for the three main sources of uncertainties located along the hydrometeorological modelling chain. They pointed out that EnKF probabilistic data assimilation provided most of the dispersion for 40    the early forecasting horizons but failed in maintaining its effectiveness with increasing lead times. A multimodel scheme allowed sharper and more reliable ensemble predictions over a longer forecast horizon.

Usually, in a hydrologic ensemble prediction system (H-EPS) framework (e.g., Schaake et al., 2007; Cloke and Pappenberger, 2009; Velázquez et al., 2009; Boucher et al., 2012; Abaza et al., 2017), the post-processing procedure over the atmospheric input ensemble is often referred as pre-processing, while post-processing aims at improving the hydrologic ensemble 45    forecasting outputs. Statistical hydrologic post-processing techniques for rectifying biases and dispersion errors (i.e., too narrow/too large) are numerous, as reviewed by Li et al. (2017). It is noteworthy that many hydrologic variables, such as discharge, follow a skewed distribution (i.e., low probability associated to the highest streamflow values), which complicates the task. Furthermore, hydrologic forecasts and observations are typically autocorrelated.

When considering any hydrologic ensemble post-processing approaches that can estimate the probability density directly 50    from the data (i.e., ensemble forecast) without assuming any particular underlying distribution, both the kernel ensemble dressing and the non-dominated sorting genetic algorithm II (NSGA-II) could be good choices. Silverman (1986) firstly proposed the kernel density smoothing method to estimate the distribution from the data by centering a kernel function K that determines the shape of a probability distribution (kernel) fitted around every data point (i.e., the bias-corrected ensemble member). The smooth kernel estimate is then the sum of those kernels. As for the choice of bandwidth h of each dressing kernel, Silverman's 55    rule of thumb finds an optimal h by assuming that the data is normally distributed. Improvements to the original idea were soon to follow. For instance, the improved Sheather Jones (ISJ) algorithm is more suitable and robust with respect to multimodality (Wand and Jones, 1994). Roulston and Smith (2003) rely on the series of "best forecasts" (i.e., best-member dressing) to com-





pute the kernel bandwidth. Wang and Bishop (2005) as well as Fortin et al. (2006) further improved the best member method. The later advocated that the more extreme ensemble members are more likely to be the best member of raw under-dispersive

forecasts, while the central members tend to be more "precise" for over-dispersive ensemble. They proposed the idea that different predictive weights should be set over each ensemble member, given each member's rank within the ensemble. Instead of standard dressing kernels that act on individual ensemble members, Bröcker and Smith (2008) proposed the affine kernel dressing (AKD) by assuming an affine mapping between ensemble and observation over the entire ensemble. The mapping parameters are determined from the training data simultaneously with the other dressing parameters. They approximate the

distribution of the observation given the ensemble.

Other post-processing techniques, like the Non-dominated sorting genetic algorithm II (NSGA-II), are now common (e.g., Liong et al., 2001; De Vos and Rientjes, 2007; Confesor and Whittaker, 2007). Such techniques are conceptually linked to the multiobjective parameter calibration of hydrologic models using Pareto approaches. Indeed, formulating a model structure or representing the hydrologic processes using a unique global optimal parameter set proves to be very subjective. Multiple opti-

mal parameter sets exist with satisfying behavior given the different conceptualizations, albeit not identical Beven and Binley (1992). For example, Brochero et al. (2013) utilized the Pareto fronts generated with NSGA-II for selecting the "best" ensemble from a hydrologic forecasting model with a pool of 800 streamflow predictors, in order to reduce the H-EPS complexity. Given the single-model H-EPSs studied here, the hydrologic ensemble is generated by activating two forecasting tools: the ensemble weather forecast and the EnKF. Henceforth, enhancing the H-EPS forecasting skill by assigning different credibility to

ensemble members becomes preferred than reducing the number of members. Multiple objective functions (i.e., here, verifying scores) for evaluating the forecasting performances of the H-EPS are selected to guide the optimization process. The expected output is a group of solutions, also known as Pareto fronts, that can give the trade-offs between different objectives.

In this study, the daily streamflow ensemble forecasts issued from five single-model H-EPSs over the Gatineau River (Province of Québec, Canada) are post-processed. Details about the study area, hydrologic models, and hydrometeorologic

data are described in Section 2. Section 3 explains the methodology and training strategy of Affine kernel dressing (AKD) and Non-dominated sorting genetic algorithm II (NSGA-II) methods, in parallel with the scoring rules that evaluate the performance of the forecasts. Specific concepts associated with those scores are also introduced in this section. Predictive distribution estimation based on the five single-model H-EPSs configurations, which lack accounting for the model structure uncertainty, is presented in Section 4. The strengths of both statistical post-processing methods in improving the forecasting skill are

compared and analyzed as well. Conclusion follows in Section 5.

## 2 The H-EPSs

Figure 1 illustrates the study area: the Gatineau River located in southern Québec, Canada. It drains 23,838 km2 of the Outaouais and Montréal hydrographic region and experiences a humid continental climate. The river starts from Sugar Loaf Lake (47° 52 54N, 75° 30 43W) and joins the Ottawa River some 400 km later. The average daily temperature is about -3°C

in winter while the temperature spectrum is 10°C-22°C in summer (Kottek et al., 2006). The hydrologic regime of the study



area is generally wet, cold, and snow-covered. The largest flood typically appears in spring or early summer (i.e., from March to June) from snowmelt and rainfall. Autumnal rainfall often leads to a lesser peak between September and November (Figure 2).

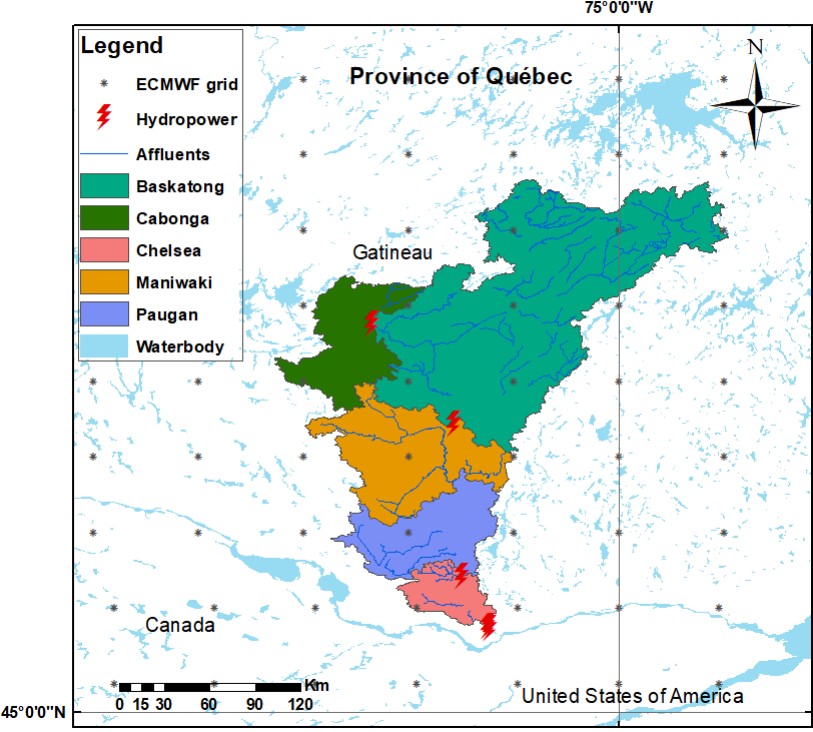

**Figure 1.** The five sub-catchments of the Gatineau River. The red thunder marks locate the dams while the original ECMWF grid points, before downscaling, are marked using black stars.

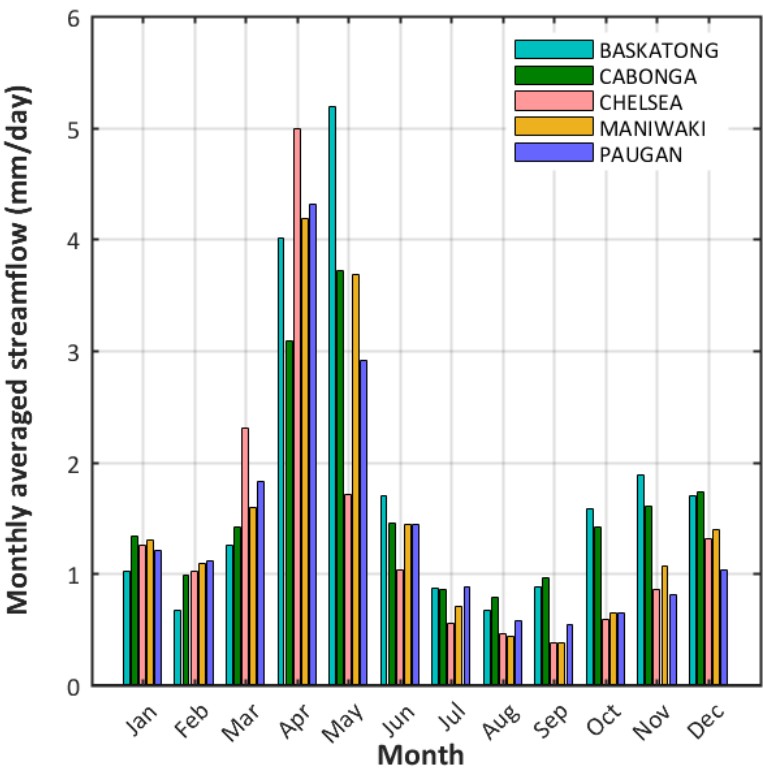

**Figure 2.** Monthly averaged hydrographs.

For operational hydrologic modelling, reservoir operation, and hydroelectricity production, the whole catchment has been

divided up into five sub-catchments: Baskatong, Cabonga, Chelsea, Maniwaki, and Paugan, identified by different colors in

Figure 1. The sub-catchments are modelled independently from one another, in order to inform a decision model operated by

Hydro-Québec (e.g., Movahedinia, 2014). Dams are identified in Figure 1 as red thunder marks. The two most upper ones

allow the existence of large headwater reservoirs, while the other three are run-of-the-river installations. The daily streamflow

($m^3/s$) time series entering the reservoirs were constructed by the electricity producer from a diversity of local information

and made available to the study along with spatially averaged minimum and maximum air temperature ($°C$) and precipitation

($mm$) for each sub-basin.

The time series extend from January 1950 to December 2017. The study focuses on the last 33 years (1985-2017) to avoid

the increased bias and variability caused by missing values within the record. Table 1 summarizes the various hydroclimatic

characteristics of the Gatineau River sub-catchments. Potential evapotranspiration is calculated from the temperature-based

Oudin et al. (2005) formulation.



**Table 1.** Hydroclimatic characteristics of five sub-catchments of the Gatineau River.

| Name | Lat. | Lon. | Catchment Area ($km^2$) | Reservoir Area ($km^2$) | Tmax (°C) | Tmin(°C) | Mean annual Q ($mm$) | Mean annual P ($mm$) |
|---|---|---|---|---|---|---|---|---|
| Cabonga | 47.21 | -76.59 | 2,665 | 426.4 | 8.7 | -3.2 | 1.35 | 977 |
| Baskatong | 47.21 | -75.95 | 13,057 | 315.7 | 8.3 | -3.8 | 1.49 | 1,031 |
| Maniwaki | 46.53 | -76.25 | 4,145 | 0 | 9.9 | -2.1 | 1.24 | 961 |
| Paugan | 46.07 | -76.13 | 2,790 | 35.2 | 10.4 | -1.3 | 1.29 | 972 |
| Chelsea | 45.70 | -76.01 | 1,142 | 10.4 | 10.9 | -0.5 | 1.27 | 957 |

The meteorological ensemble forecasts were retrieved from the European Center for Medium-Range Weather Forecasts (ECMWF; Fraley et al. (2010)). The time series extend from January 2011 to December 2016. The meteorological ensemble forecast used the reduced Gaussian transformation to the latitude-longitude system during the THORPEX Interactive Grand Global Ensemble (TIGGE) database retrieving by bilinear interpolation (e.g., Gaborit et al., 2013). The horizontal resolution was downscaled during retrieval from the 0.5° ECMWF grid resolution to a 0.1° grid resolution. This study resorts to the 12:00 UTC forecasts only, aggregated to a daily time step over a 7-day horizon. All data are aggregated at the catchment scale, averaging grid points located within each sub-catchments.

The HydrOlOgical Prediction Laboratory (HOOPLA; Thiboult et al. (2020)) provides the modular framework to perform calibration, simulation, and streamflow prediction using multiple (20) hydrologic models (Perrin, 2000; Seiller et al., 2012). The empirical two-parameter model CemaNeige (Valéry et al., 2014) simulates snow accumulation and melt. Five random hydrologic models from HOOPLA are exploited in this study. Their main characteristics are summarized in Table 2. All time series were split in two following the Split-Sample Test (SST) procedure of Klemeš (1986): 1986-2006 for calibration and 2013-2017 for validation. In both cases, three prior years were used for spin-up. January 2011-December 2016 is committed to hydrologic forecasting.

**Table 2.** Main characteristics of the hydrologic models (Seiller et al., 2012).

| Model | No. of optimized parameters | No. of reservoirs | Derived from |
|---|---|---|---|
| M01 | 6 | 3 | BUCKET (Thornthwaite and Mather, 1955) |
| M02 | 4 | 2 | GR4J (Perrin et al., 2003) |
| M03 | 9 | 3 | HBV (Bergström et al., 1973) |
| M04 | 7 | 3 | IHACRES (Jakeman et al., 1990) |
| M05 | 9 | 5 | SACRAMENTO (Burnash et al., 1973) |

Initial condition uncertainties within each H-EPS are accounted for by a 100-member Ensemble Kalman Filter (EnkF) that adjusts the model states distribution function given observational distributions. Meteorological uncertainties are quantified by providing the 50-member ECMWF ensemble forcing to the H-EPSs. Resulting ensemble streamflow forecasts thus consists of 5,000 members. This set-up is similar to the one described in more details by Thiboult et al. (2016).



## 3 Methodology

This study was conducted on the base of 1-7-day ensemble streamflow forecasts issued from five single-model H-EPSs and their
realizations. Both Affine kernel dressin and Nondominated sorting genetic algorithm II are utilized in this study as the statistical
post-processing or so-called ensemble interpretation method (Jewson, 2003)to transform the raw ensemble forecast into a
probability distribution. Rather than adopting the ensemble mean and the standard deviation and approximate the distribution
of the raw ensemble (Wilks, 2002), the principal insight of the methodology is that the probability distribution could be fitted
of the observation given the ensemble (Bröcker and Smith, 2008).

### 3.1 Affine kernel dressing

Affine kernel dressing (AKD) interprets the ensemble by approximating the distribution of the observation given the ensemble
forecasts. The ordering of the ensemble members is not taken into account (ensemble members are considered exchangeable).
Here, we denote the ensemble forecasts with m members over time by $X(t) = [x_1(t), x_2(t), \ldots, x_m(t)]$ and the observation by
$y(t)$. The mean and the variance of the raw ensemble forecasts are then:

$$\mu(X) = \frac{1}{m} \sum_i x_i \tag{1}$$

$$\upsilon(X) = \frac{1}{m} \sum_i [x_i - \mu(X)]^2 \tag{2}$$

In a general from, the probability density function of $p(y; X, \theta)$ defines the interpreted ensemble (i.e., kernel dressed ensemble) given the original ensemble with free parameter vector $\theta$:

$$p(y; X, \theta) = \frac{1}{bh} \sum_i K\left(\frac{y - ax_i - b}{h}\right) \tag{3}$$

for which the interpreted ensemble can be seen as a sum of probability functions (kernels) around each raw ensemble
member. $x_i$ represents the $i^{th}$ ensemble member and y is the corresponding observation. Hence, $ax_i + b$ identifies the center
of each kernel using the scale parameter $a$ and offset parameter $b$. $h$ is the positive bandwidth of each kernel. Note that various
distributions could be adopted as kernels (Silverman, 1986; Roulston and Smith, 2003; Bröcker and Smith, 2008). We opted
for the standard Gaussian density function with zero mean and unit variance for its computational convenience:

$$K(\cdot) = \frac{1}{\sqrt{2\pi}} exp\left(-\frac{1}{2}(\cdot)\right)^2 \tag{4}$$



The mean and the variance of the interpreted ensemble can be defined as:

$$\mu'(X) = b + a \cdot \frac{1}{m}\sum_i x_i = b + a \cdot \mu(X) \tag{5}$$

$$\upsilon'(X) = h^2 + a^2 \cdot \frac{1}{m}\sum_i [x_i - \mu(X)]^2 = h^2 + a^2 \cdot \upsilon(X) \tag{6}$$

The mapping parameters of $a$, $b$, and $h$ are determined from the raw ensemble. The updated mean $\mu'(X)$ of the kernel dressed ensemble is a function of the raw ensemble mean $\mu(X)$, scaled and shifted using a and b. The variance $\upsilon'(X)$ of the kernel dressed ensemble is a function of the initial ensemble variance $\upsilon(X)$, scaled and shifted using $a^2$ and $h^2$. Detailed derivations of these equations are given by Bröcker and Smith (2008).

AKD provides the following solutions for determining parameters of $a$, $b$, and $h$:

$$b = r_1 + r_2 \cdot \mu(X) \tag{7}$$

$$h^2 = h_S^2 \cdot [s_1 + s_1 \cdot a^2 \cdot \upsilon(X)] \tag{8}$$

$$h_S = 0.5 \cdot [4/(3m)]^{1/5} \tag{9}$$

Here, $h_S$ is calculated based on the Silverman's rule of thumb (Silverman, 1986). Once the optimal free parameter vector $\theta = [\alpha, r_1, r_2, s_1, s_2]$ is obtained, the interpreted ensemble can be set to:

$$p(y; X, \theta) = \frac{1}{bh}\sum_i K\left(\frac{y - ax_i - b}{h}\right) = \frac{1}{bh}\sum_i K\left(\frac{y - z_i}{h}\right) \tag{10}$$

$$Z_i = ax_i + r_2 \cdot \mu(X) + r_1 \tag{11}$$

$$h^2 = h_S^2 \cdot [s_1 + s_2 \cdot \upsilon(X)] \tag{12}$$

Where, $Z_i$ is the resulting kernel dressed ensemble, based on the raw ensemble $X$ and fitted parameters $a$, $r_1$, and $r_2$. Bröcker and Smith (2008) stressed that this affine ensemble transformation works on the whole ensemble rather than on each individual

member. Finally, the mean and variance of the interpreted ensemble shown in Eq. (5) and Eq. (6) can be further defined as:

$$\mu'(X) = b + a \cdot \mu(X) = r_1 + (a + r_2) \cdot \mu(X) \tag{13}$$





$$v'(X) = h^2 + a^2 \cdot v(X) = h_S^2 \cdot s_1 + a^2 \cdot \left(h_S^2 \cdot s_2 + 1\right) \cdot v(X) \tag{14}$$

### 3.2 Nondominated sorting genetic algorithm II

Multiobjective optimization problems are common and typically lead to a set of optimal options (Pareto solution set) for users
to choose from. Exploiting a genetic algorithm to find all Pareto solutions out from the entire solution space have been proposed
and improved since the publication of the vector-evaluated genetic algorithm (VEGA) around 1985 (Schaffer, 1985).

There exist two main standpoints for dealing with multiobjective optimization problems: (1) define a new objective function
as the weighted sum of all desired objective functions (e.g., MBGA, RWGA) or (2) determine the Pareto set or its representative
subsets for a selected group of objective functions (e.g., SPEA, SPEA-II, NSGA, NSGA-II). The first approach is more trivial
as it reduces to a single-objective optimization problem. Yet, the needed weighting strategy is difficult to set accurately as a
minor difference in weights may lead to quite different solutions. On the other hand, Pareto-ranking approaches have been
devised in order to avoid the problem of converging towards solutions that only behave well for one specific objective function.
Users still have to select objective functions that are pertinent to the problem and that are not heavily correlated to one another.
Readers may refer to the review of Konak et al. (2006) for more details.

Similar ideas can be utilized in this study as the goal is to achieve a "good forecast". Various efficiency criteria are needed
when we verify whether an H-EPS is competent issuing accurate and reliable forecasts. Usually, $bias$ is the first idea that
crosses our mind. $Bias$, also known as systematic error, refers to the correspondence between the average forecast and the
average observation, which is different from accuracy. For example, systematic $bias$ exists in the streamflow forecasts that are
consistently too high or too low .

Meanwhile, hydrologists also rely heavily on the Nash-Sutcliffe efficiency criterion ($NSE$, Nash and Sutcliffe (1970)) for
measuring how well forecasts reproduce the observed time series. Transforming the time series beforehand allows specializing
it (i.e., NSEinv, NSEsqrt) for specific needs (e.g., Seiller et al., 2017). $NSE$ is dimensionless and varies on the interval of
$[-\infty, 1]$. A perfect model output would have an $NSE$ of one.

$NSE$ and $bias$ are utilized here as objective functions, which is to say that it is seeked minimize $bias$ and maximize $NSE$
simultaneously. A fast and elitist multiobjective genetic algorithm, the Nondominated sorting genetic algorithm II (NSGA- II;
Deb et al. (2002)) is adopted for searching for the Pareto solution set. NSGA-II offers three specific advantages over previous
genetic algorithms: 1) there is no need to specify extra parameters such as the niche count for the fitness sharing procedure; 2)
it reduces complexity over alternative GA implementations; 3) elite individuals are well maintained and hence the effectiveness
of the multiobjective genetic algorithm is largely improved.

In this study, the population is denoted by $X(t) = [x_1(t), x_2(t), \ldots, x_m(t)]$, a forecast ensemble of $m$ members over time $t$,
and the observation is denoted by $y(t)$. Specific steps for NSGA-II are briefly introduced here:

1) Layer the whole population by using the fast nondominated sorting approach: $i$ is initially set to 1, while $z_i$ represents the
$i^{th}$ solution among the $m$ ones. Compare the domination and nondomination relationship between the individuals $z_i$ and $z_j$ for
all the $j = 1, 2, \ldots, m$ and $i \neq j$. $z_i$ is the nondominated solution as long as no $z_j$ dominates it. This process is repeated until





all the nondominated solutions are found and composed the first nondominated front of the population. Note that the selected individuals of the first front can be neglected when searching for subsequent fronts (i.e., marked as $k_{rank}$).

2) Find the crowding distance for each individual in each front. This step ensures the diversity of the population. For example, for the first front, sort the values of the objective functions in an ascending order. The boundary solutions (i.e., maximum and minimum solutions) are then the value at infinity. The crowding distance for other individuals can be assign as:

$$k_{distance} = \sum_{k=1}^{m} \left( \left| \int_{n}^{j+1} - \int_{n}^{j-1} \right| \right) \tag{15}$$

where $k_{distance}$ represents the value for the $k^{th}$ individual and $f_n^{j+1}$ and $f_n^{j-1}$ are the values of the $n^{th}$ objective function at $j+1$ and $j-1$, separately. Thereafter, the crowding-comparison operator can be utilized based on $k_{rank}$ and $k_{distance}$. Individual $z_i$ will be assumed superior than $z_j$ if $k_{rank}^i < k_{rank}^j$ or $k_{distance}^i > k_{distance}^j$, when their Pareto front ranks are equal.

3) Elitism strategy is introduced in the main loop. Offspring population $Q_t$ is firstly generated from parent population $P_t$ after mutation and gene cross-over. Then the abovementioned nondominated sorting and crowding distance assignment are conducted on the composed population $R_t$ that contain both $Q_t$ and $P_t$ with the size of $2m$. The fist-rate nondominated solutions will be assign to the new parent population $P_{t+1}$. Outputs after the whole evolutionary search are the un-repeated nondomination solutions and a weight matrix can also be extracted from the solutions. Specifically, in this study, the population

size is set to 50, the number of objective functions equals 2, the boundary is from 0 to 1, the mutation probability and crossing-over rate are 0.1 and 0.7, and the maximum evolution runs are 430 times.

### 3.3    Verifying metrics

The performance of the post-processed forecast distributions, mostly in terms of accuracy and reliability, is assessed using scoring rules. Except for bias and $NSE$ described above, seven other verifying scores are applied to both the raw and post-

processed forecast distributions.

The Mean Continuous Ranked Probability score ($MCRPS$, Matheson and Winkler, 1976, Hersbach, 2000, Gneiting and Raftery, 2007) verifies the accuracy and reliability of the probabilistic forecast. In practice, the $MCRPS$ is the average value of $CRPS$ over the whole time series $T$:

$$MCRPS = \frac{1}{T} \sum_{t=1}^{T} \int_{-\infty}^{+\infty} \left( P_t^{fcst}(y) - H \left( y_t \geq y_t^{obs} \right) \right)^2 dy \tag{16}$$

where $y$ is the predictand and $y_t^{obs}$, the corresponding observations. $P_t^{fcst}(y)$ is the cumulative distribution function of the forecasts at time step $t$. The Heaviside function $H$ equals 0 (or 1) when $y_t < y_t^{obs}$ (or, $y_t \geq y_t^{obs}$). The optimal $MCRPS$ value is 0.





As for the deterministic metrics, we adopt the mean absolute error ($MAE$) and mean square error ($MSE$, e.g., Brochero et al., 2013). More accurate forecasts lead to lower $MAE$ and $MSE$.

The Kling-Gupta efficiency ($KGE$; Gupta et al., 2009) also allows for a comprehensive performance assessment of the deterministic forecasts. $KGE'$, a slightly modified version of $KGE$ (Kling et al., 2012), avoids any cross-correlation between the bias and the variability ratios. It is defined as:

$$KGE' = \sqrt{(r-1)^2 + (\beta - 1)^2 + (\gamma - 1)^2} \tag{17}$$

$$\beta = \frac{\mu_y}{\mu_o} \tag{18}$$

$$\gamma = \frac{CV_y}{CV_o} = \frac{\sigma_y / u_y}{\sigma_o / u_o} \tag{19}$$

The correlation coefficient $r$ represents the linear association between the deterministic forecast and the observations. $\mu_y$ ($\mu_o$) and $\sigma_y$ ($\sigma_o$) are the mean and the standard deviations of the forecasts (here, the ensemble mean) and observation, respectively. $CV$ is the dimensionless coefficient of variation. The Taylor diagram (Taylor, 2001) provides a brief convenient summary in terms of $r$, $RMSE$, and ratio of variances.

The $Reliability diagram$ (Stanski et al., 1989) is a graphical representation of the reliability of an ensemble forecast. It contrasts the observed frequency against the probability of ensemble forecasts over all quantiles of interest. The proximity from the diagonal line indicate how close the forecast probabilities are associated to the observed frequencies for selected quantiles. The 45° diagonal line thus represents perfect reliability, i.e., when the ensemble forecast probabilities equals the observation ones. When the plotted curve lies above the 45° line, the predictive ensemble is over-dispersed. It is otherwise

under-dispersed. In addition, a flat curve reveals that the forecast has no resolution (i.e., climatology).

   The Spread Skill plot ($SSP$, Fortin et al., 2014) assesses the ensemble spread and identifies an ensemble forecast with poor predictive skill and large dispersion that would be positively assessed by a reliability diagram. Fortin et al. (2014) stresses that the ensemble spread should match the $RMSE$ of the ensemble mean when the predictive ensemble is reliable. Thus, the $SSP$ complements the spread component with an accuracy aspect.


### 3.4   Experimental setup

Establishing and analyzing both AKD and NSGA-II predictive models to interpret single-model hydrologic ensemble forecasts for uncertainty analysis can be summarized in three steps:

   (1) Determine the length of the training period. The target ensemble for interpretation has a horizon that extends from day 1 to

7. It is a well-known fact that the skill of hydrologic forecasts fades away with increasing lead time. The 4-day-ahead ensemble





forecasts issued from each single-model H-EPSs and their corresponding observations are chosen as a training dataset, since located in the middle of the forecast horizon. The test dataset thus consists of the remaining forecasts: day 1-3 and 5-7.

(2) Affine mapping between the ensemble and observation over the training dataset. The observation time series are used to identify the free parameter vector $\theta = [\alpha, r_1, r_2, s_1, s_2]$, minimizing the $MCRPS$ to obtain the kernel-dressed ensemble. Note

that AKD acts on the entire ensemble rather than on each individual member.

(3) Evaluate the Pareto fronts (i.e., nondominated solutions that minimize/maximize the $bias$ and the $NSE$) and the weight matrix, applying NSGA-II over the training dataset (Sloughter et al., 2007). A 30-day moving window is selected so it contains enough training samples with coherent consistency and satisfies operational requirements.

A general flowchart of the streamflow input, AKD and NSGA-II frameworks, and expected outputs is illustrated in Figure

275 3.

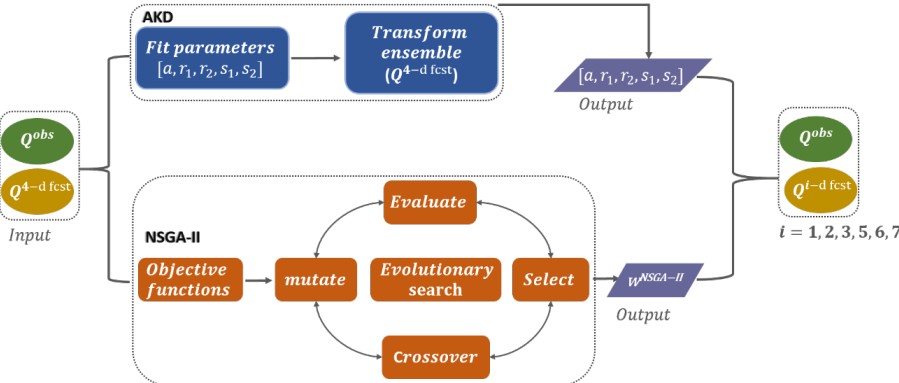

**Figure 3.** Flowchart of the experimental setup.

## 4   Results and discussions

### 4.1   Ensemble member exchangeability

The issue of member interchangeability is central to this study, since for AKD each raw ensemble will be considered as a whole (i.e., indistinguishable members) whereas for NSGA-II a weight matrix is sought, which implies that different weights are assigned to each candidate members.

Interchangeability is here assed visually, simultaneously looking at the individual RMSE values of all 5,000 members, 7 daily forecast horizons, and 5 H-EPSs. Figure 4 displays (typical) values for day 500 and Baskatong sub-catchment – a video

covering the full time series is available as a supplemental material to this paper). For each H-EPS – forecast horizon boxes, horizontal lines consist of 100 EnKF members and vertical lines, of 50 meteorological members. Mosaics with redder colors reveal higher values of the $RMSE$. The decreasing predictive skill of the H-EPSs with lead time is hence revealed by an ever increasing redder mosaic.

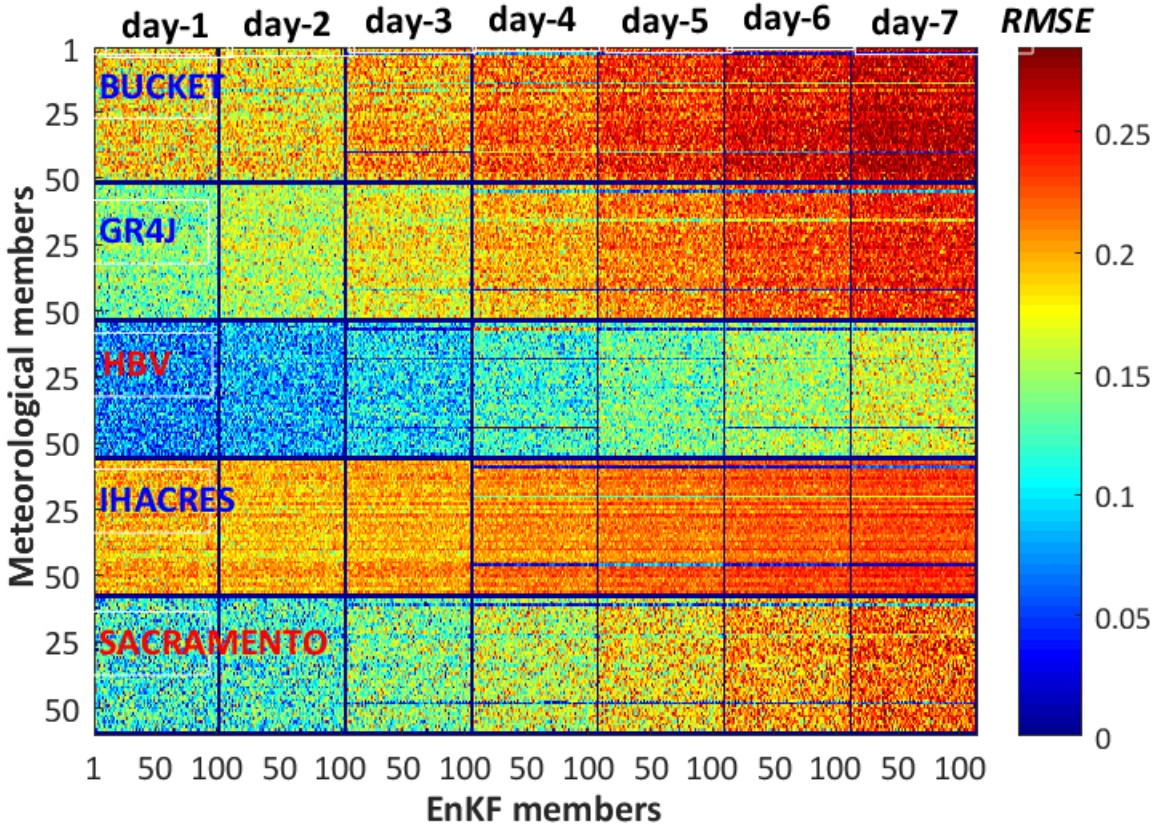

**Figure 4.** Illustration of the $RMSE$ values (mm/day) of the individual members of the forecast issued by the 5 H-EPSs for the Baskatong sub-catchment, on day 500. There are 7 daily forecasts horizons. Each box consists of 5,000 members, from 100 EnKF members (horizontal lines) and 50 meteorological members (vertical lines).

What is drawn in Figure 4 as ECMWF (50) members are not weather forecasts but rather the hydrologic forecasts that were built upon them. The idea behind Figure 4 (and accompanying video) is to visually asses if the initial interchangeability of the weather forecasts holds for the hydrologic forecasts (horizontal lines). The interchangeability of the probabilistic data assimilation scheme is assessed at the same time (vertical lines). One can notice in Figure 4 colorful horizontal lines within each box start to appear from day 3 and on, revealing a distinguishable character with longer lead times. At the same time, no obvious vertical lines are present in the same figure. These results suggest that the hydrologic forecasts produced in this study are fully interchangeable with respect to EnKF, but less so with respect to the weather, the latter being non-linearly transformed by the hydrologic models. This opens up the possibility of assigning weights to the hydrologic forecasts associated to the ECMWF members.

For practical reasons, as the 100-member data assimilation ensemble was deemed fully interchangeable, this component is randomly reduced to 50 members from now on in this document. This procedure simplifies the implementation of the AKD and NSGA-II post-processing computations, which results are presented next.





## 4.2 Uncertainty analysis

The NSGA-II Pareto front drawn in Figure 5 (model M01 over the Baskatong catchment) is quite typical. In this multiobjective evolutionary search, 35 (nondominated) Pareto solutions are identified. Which is to say, the optimal $NSE$ is inevitably accompanied with the highest bias (e.g., $NSE = 0.84594$, $bias = 0.034055$), or vice versa. Figure 6 confirms NSGA-II convergence.

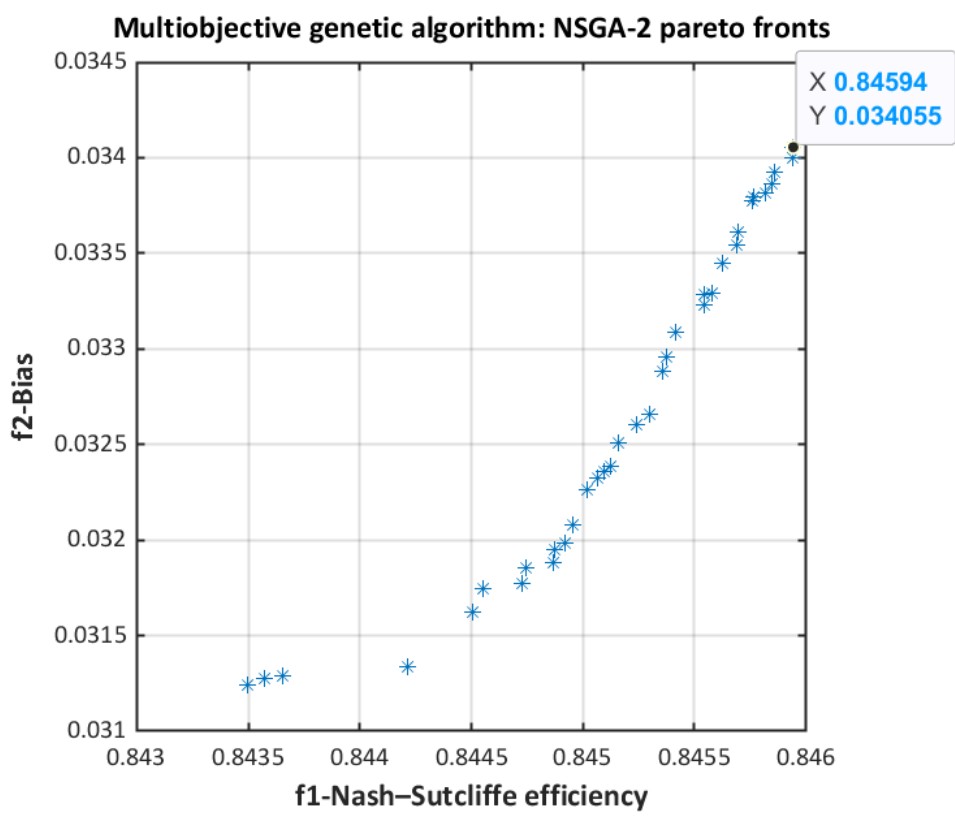

**Figure 5.** NSGA-II Pareto fronts of model M01 over Baskatong catchment. Horizontal and vertical axis are $NSE$ and $bias$, separately.



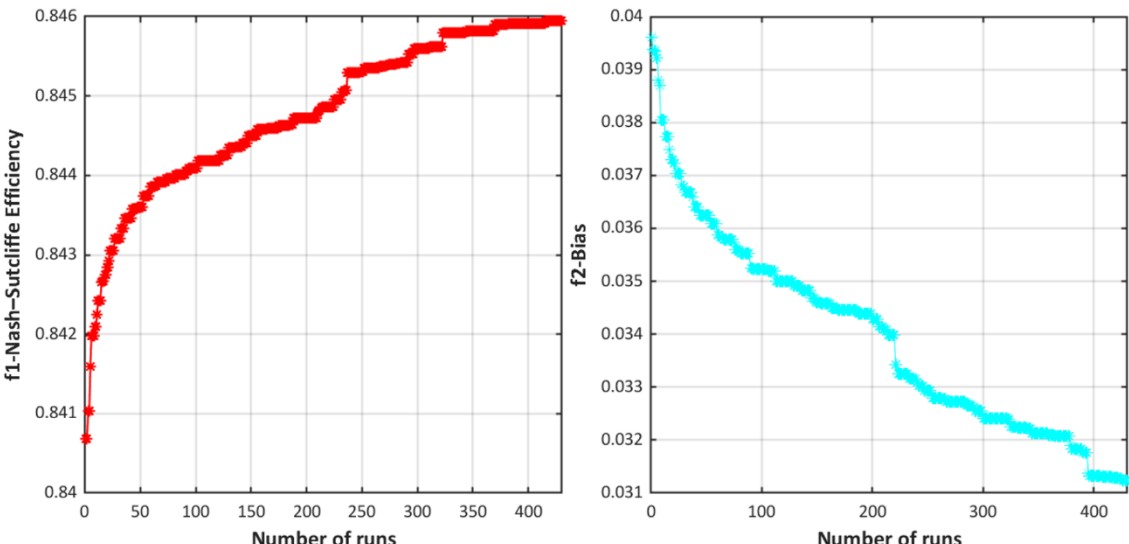

**Figure 6.** NSGA-II dynamical performance plots for both objective functions versus the number of evaluations, for model M01 over Baskatong catchment.

However, deterministic uncertainty analysis is not sufficient to compare the skill of these two post-processing methods. The
315 accuracy and reliability are verified in terms of probabilities as well. As mentioned above, the predictive models are trained on
4-day ahead ensemble forecasts issued from each model, and corresponding observations. The 1-3 and 5-7-day ahead forecasts
are used as a testing dataset. The reliability of the raw, kernel dressed and NSGA-II predictive distributions with different lead-
times are displayed in the $reliability diagrams$ of Figure 7. Both post-processing methods improve over the raw ensemble,
especially the NSGA-II, as it achieves the best reliability. Over-dispersion exists mainly over the Baskatong catchments for
320 NSGA-II.

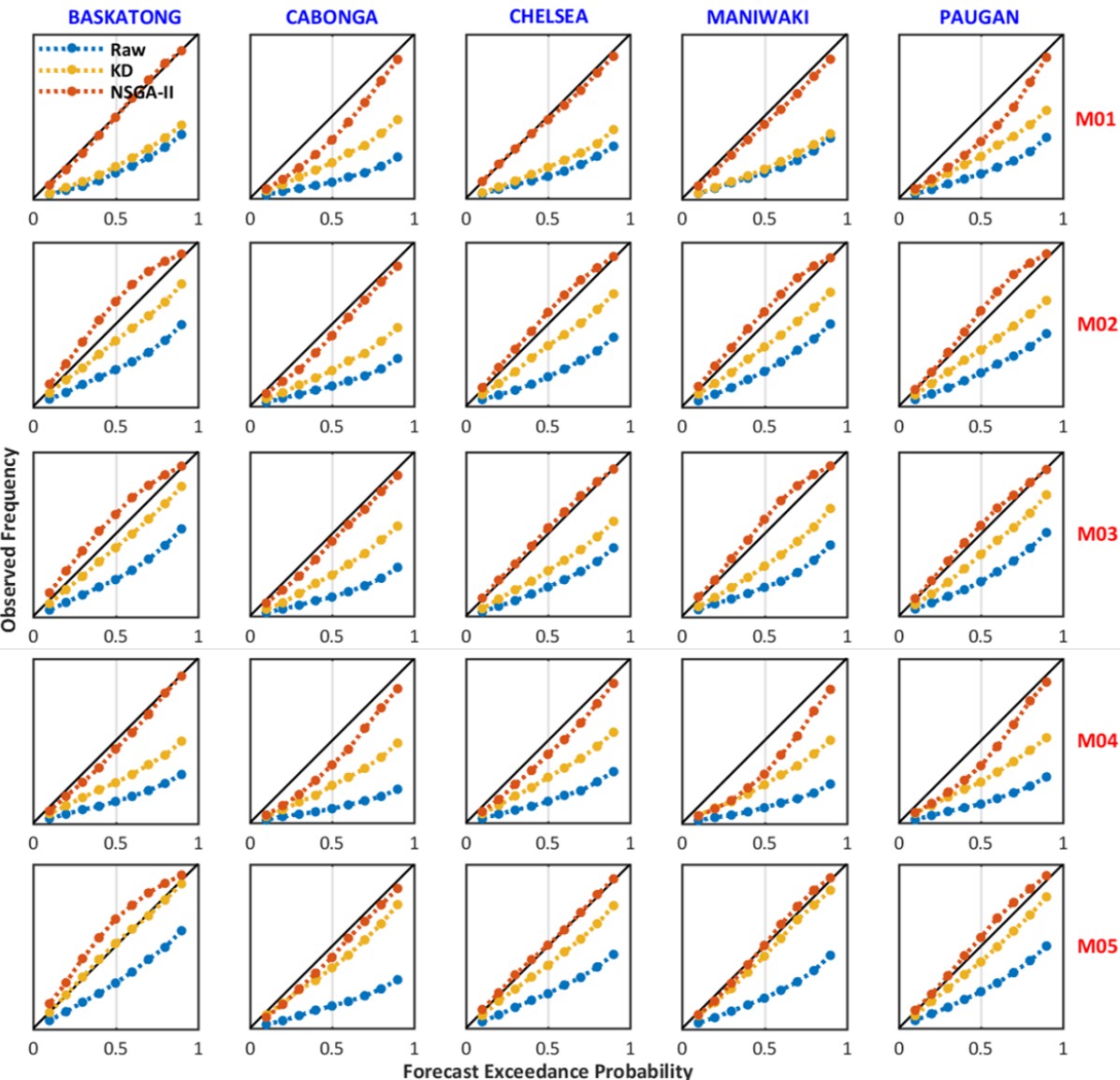

**Figure 7.** Forecasting reliability of the raw, AKD, and NSGA-II forecasts on the calibration data set (4-day ahead forecast) for five single-model H-EPSs over each individual catchment.

In the meanwhile, the relevant accuracy performances of the raw, AKD, and NSGA-II predictive models are summarized
325    using radar plots in Figure 8. We can notice that the kernel dressed ensemble fails in decreasing the forecast bias. However, it
adjusts the ensemble dispersion properly. As for the NSGA-II, the post-processed ensemble has an obvious improvement on
both bias and ensemble dispersion. Accordingly, it demonstrates a very reliable performance shown in the reliability diagram.







**Figure 8.** Accuracy performance assessment of the raw, AKD, and NSGA-II forecasts (4-day ahead) for five single-model H-EPSs over each sub-catchment of the Gatineau catchment.

The trained optimal free parameter vector $\theta = [\alpha, r_1, r_2, s_1, s_2]$ or weight estimates are obtained over the 4-day ahead ensemble forecasts. They are then applied to the validation data set. It comprises the 1-, 3-, 5-, and 7-day ahead raw forecasts issued from the associated H-EPSS. Figure 9 shows reliability diagrams for raw, kernel dressed, and NSGA-II forecasts for the validation data set over each individual catchment. Again, raw forecasts display a severe under-dispersion, revealing that error growth is not maintained well in a single-model H-EPS. However, the two statistical post-processing methods succeed in improving the forecast reliability, with the curves closer to the bisector lines. Note that over-dispersion appears with the AKD





transformed ensembles, especially at shorter lead times. The ensemble spread tends to a proper level as the lead time increases. In contrast, the predictive distributions of the kernel dressed ensemble are the most reliable for model M05 over almost all individual catchments.

**Figure 9.** Comparison of the reliability of the raw, kernel dressed and NSGA-II forecasts on the validation dataset (i.e., 1-3 and 5-7-day ahead forecasts) for five singe-model H-EPSs over all catchments.

One can notice that the performance among each single-model H-EPS only has a slightly difference and increases with the growing horizon. In general, the two statistical post-processor maintained the original accuracy level compared with the raw ensemble. As a complement, Figure 10 demonstrates the ensemble spread with different forecasting horizons on the $x$-axis, showing the changing performance trend. Clearly, both the kernel dressed ensemble and NSGA-II predictive forecasts have increased dispersion for all models over all catchments and result in more reliable predictive distributions.





Figure 10 also provides an intuitive reference of the accuracy performance of the raw, AKD and NSGA-II interpreted ensemble forecasts in terms of the $MAE$, $MCRPS$, and the ensemble dispersion for different forecasting horizons, showing the

evolution of forecasting performance. Clearly, both the kernel dressed ensemble and NSGA-II forecasts have increased dispersion compared to raw forecasts for all models and over all catchments. This results in more reliable predictive distributions, as shown in Figure 9.

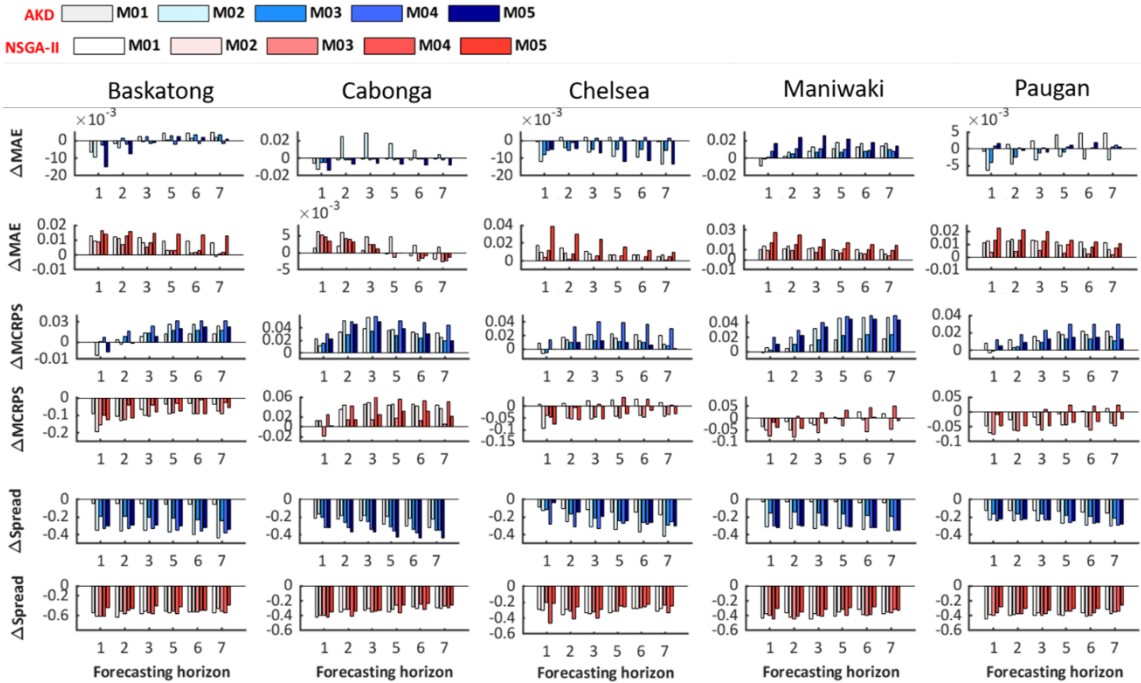

**Figure 10.** Comparison of the $MAE$, $MCRPS$, and ensemble dispersion of the raw, AKD, and NSGA-II forecasts (i.e., 1-3 and 5-7-day

ahead forecasts) for five singe-model H-EPSs over all catchments. The x-axis for each sub-plot represents different horizons.

## 5 Conclusions

For all hydrologic processes, different sources of uncertainties originate from the natural dynamical system's intrinsic chaotic mechanisms, imprecise knowledge of initial conditions and assumptions undelying the hydrologic modelling. They are inevitable. Pappenberger et al. (2005) proposed the term "uncertainty cascade model" to represent the uncertainty propaga-

tion chain. Researchers proposed various approaches for better identifying and representing the different sources of uncertainty along the forecasting chain. They are mainly associated with the observations, the initial conditions, and the atmospheric/climatic/hydrologic models.

Hydrologic post-processing of streamflow forecasts plays an important role for correcting the overall representation of uncertainties in the final streamflow forecasts. Both the kernel ensemble dressing and the evolutionary multiobjective optimization

approaches are tested in this study to estimate the probability density directly from the data (i.e., daily ensemble streamflow





forecast) over five single-model hydrologic ensemble prediction systems (H-EPSs). The Affine kernel dressing (AKD) method provides an affine mapping between the entire ensemble forecasts and the observations without any assumption of the underlying distributions. The Pareto fronts generated with NSGA-II relaxes the parametric assumptions regarding the shape of the predictive distributions and offers trade-offs between different objectives in a multi-score framework.

The single-model H-EPSs explored in this study account for both forcing uncertainty and initial conditions uncertainty by using the ensemble weather forecasts (ECMWF) and data assimilation (EnKF). Hydrologic post-processing with AKD and NSGA-II rely on very different assumptions and methodology. However, they both transform the raw ensembles into probability distributions. Results show that the post-processed forecasts achieve stronger predictive skill and better reliability than raw forecasts. In particular, the NSGA-II post-processed forecasts achieve the most reliable performances, since this

method improves both bias and ensemble dispersion. However, over-dispersion may exist occasionally over the Baskatong catchment for NSGA-II. Kernel dressed ensemble succeed in adjusting the ensemble dispersion properly, but bias increases. All in all, both AKD and NSGA-II offer efficient post-processing skill.

## Appendix A

**Table A1.** Hydroclimatic characteristics of five sub-catchments of the Gatineau River.

| Name | Lat. | Lon. | Catchment Area ($km^2$) | Reservoir Area ($km^2$) | Tmax (°C) | Tmin(°C) | Mean annual Q ($mm$) | Mean annual P ($mm$) |
|---|---|---|---|---|---|---|---|---|
| Cabonga | 47.21 | -76.59 | 2,665 | 426.4 | 8.7 | -3.2 | 1.35 | 977 |
| Baskatong | 47.21 | -75.95 | 13,057 | 315.7 | 8.3 | -3.8 | 1.49 | 1,031 |
| Maniwaki | 46.53 | -76.25 | 4,145 | 0 | 9.9 | -2.1 | 1.24 | 961 |
| Paugan | 46.07 | -76.13 | 2,790 | 35.2 | 10.4 | -1.3 | 1.29 | 972 |
| Chelsea | 45.70 | -76.01 | 1,142 | 10.4 | 10.9 | -0.5 | 1.27 | 957 |

**Table A2.** Main characteristics of the hydrologic models (Seiller et al., 2012).

| Model | No. of optimized parameters | No. of reservoirs | Derived from |
|---|---|---|---|
| M01 | 6 | 3 | BUCKET (Thornthwaite and Mather, 1955) |
| M02 | 4 | 2 | GR4J (Perrin et al., 2003) |
| M03 | 9 | 3 | HBV (Bergström et al., 1973) |
| M04 | 7 | 3 | IHACRES (Jakeman et al., 1990) |
| M05 | 9 | 5 | SACRAMENTO (Burnash et al., 1973) |

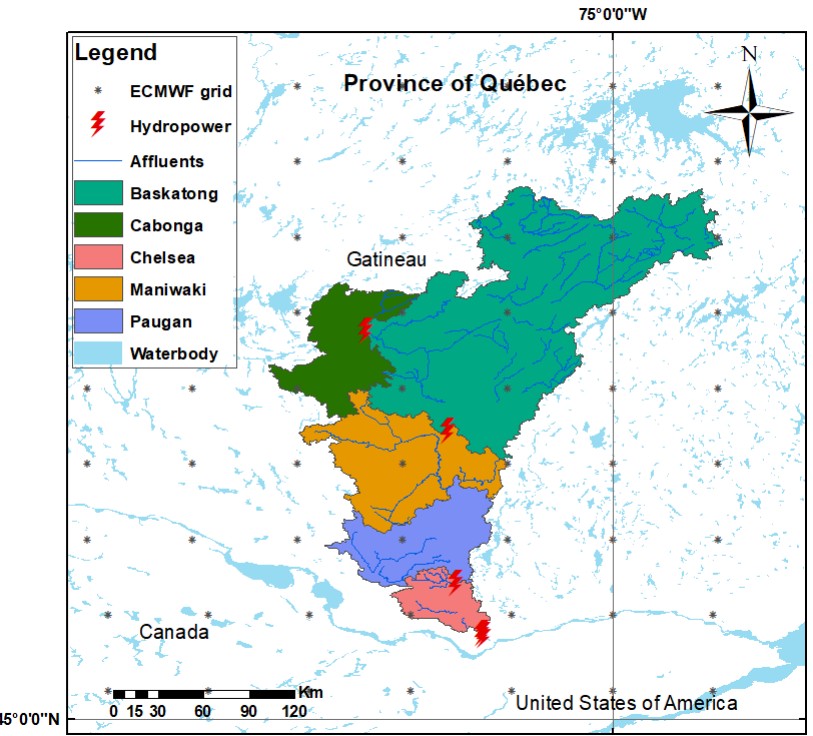

**Figure A1.** The five sub-catchments of the Gatineau River. The red thunder marks locate the dams while the original ECMWF grid points,
before downscaling, are marked using black stars.





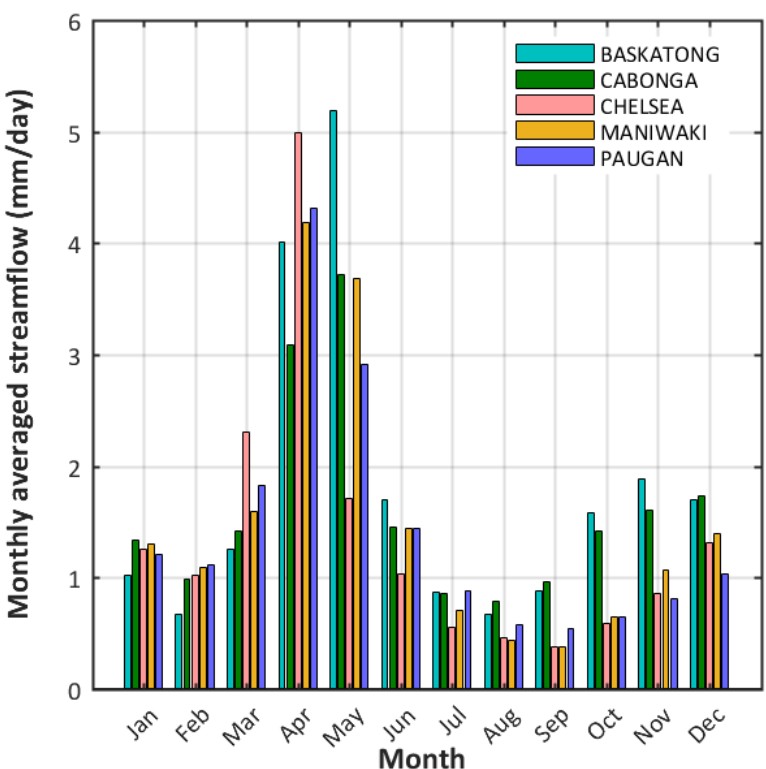

**Figure A2.** Monthly averaged hydrographs.

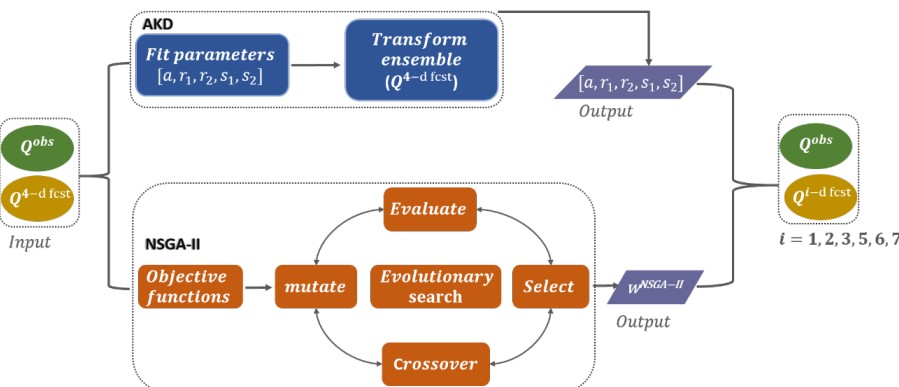

**Figure A3.** Flowchart of the experimental setup.

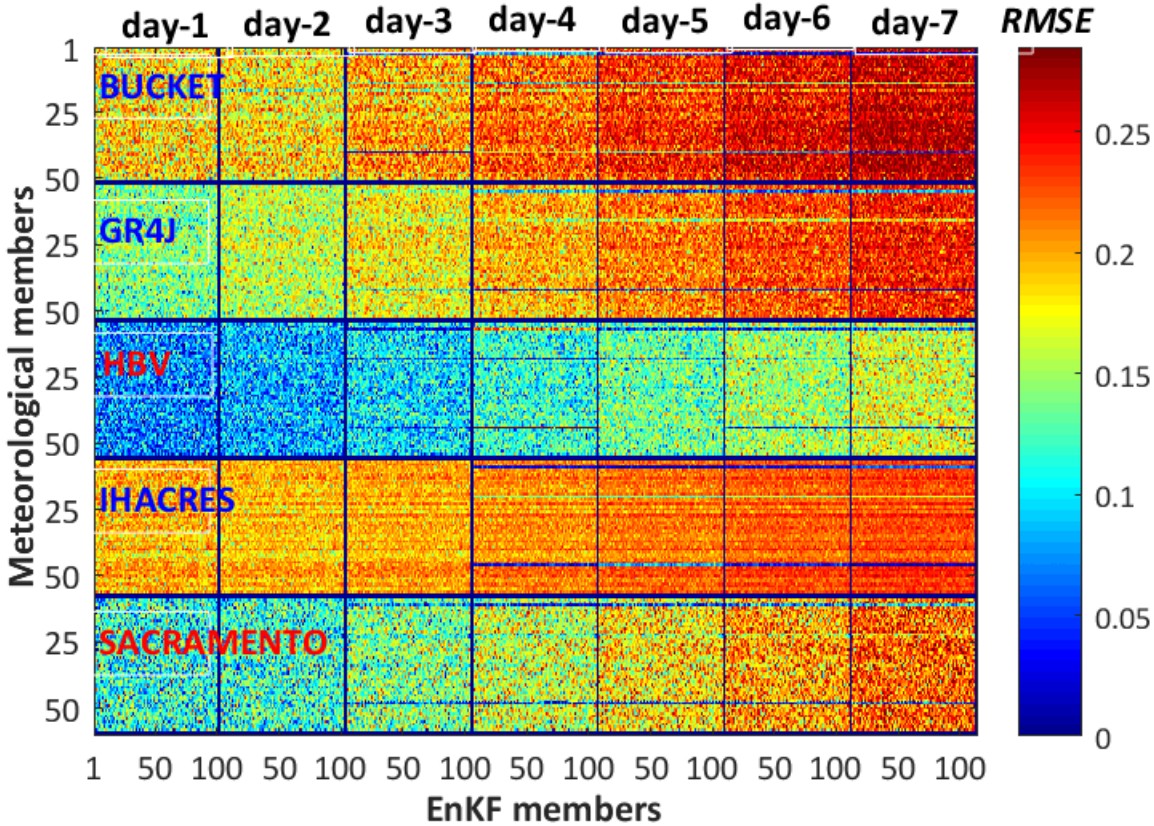


**Figure A4.** Illustration of the $RMSE$ values (mm/day) of the individual members of the forecast issued by the 5 H-EPSs for the Baskatong sub-catchment, on day 500. There are 7 daily forecasts horizons. Each box consists of 5,000 members, from 100 EnKF members (horizontal lines) and 50 meteorological members (vertical lines).





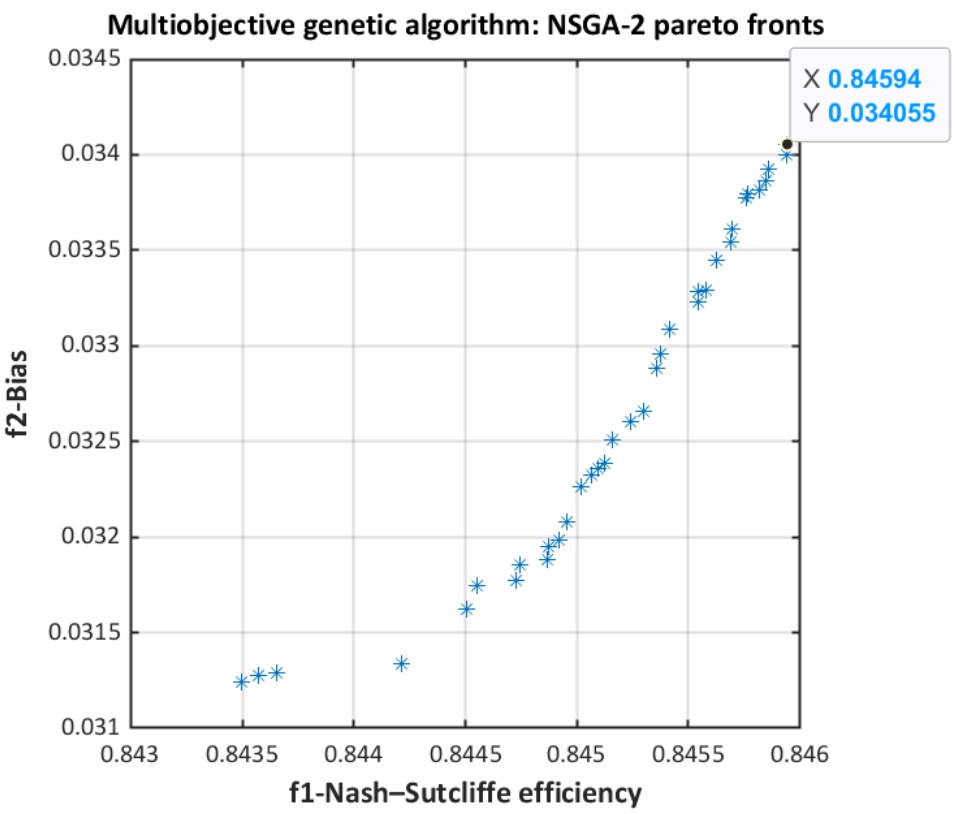

**Figure A5.** NSGA-II Pareto fronts of model M01 over Baskatong catchment. Horizontal and vertical axis are $NSE$ and $bias$, separately.

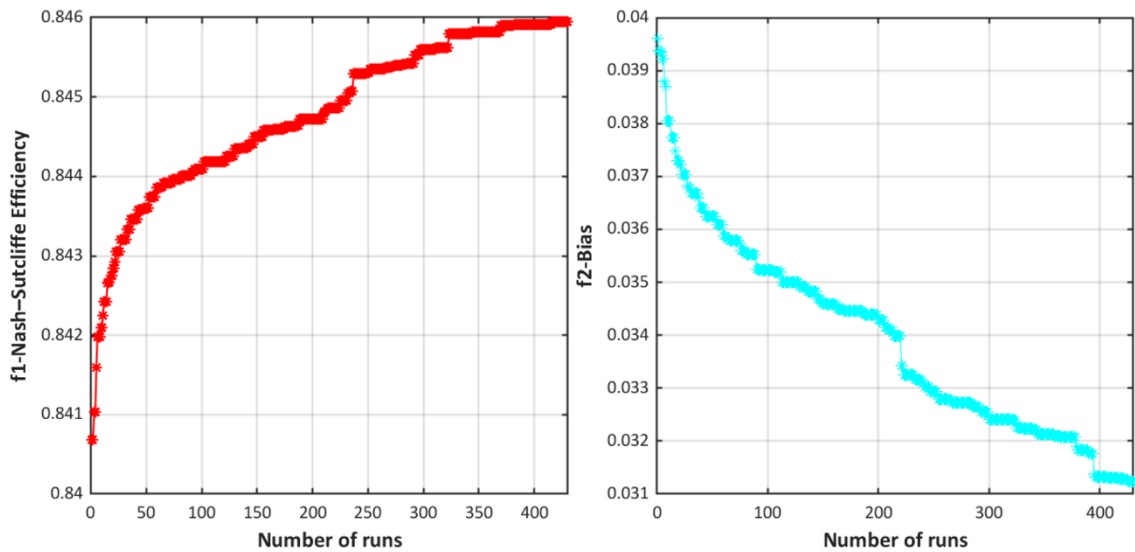



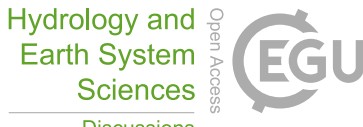

**Figure A6.** NSGA-II dynamical performance plots for both objective functions versus the number of evaluations, for model M01 over Baskatong catchment.

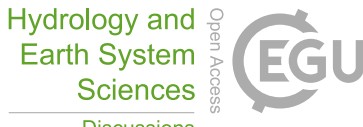

**Figure A7.** Forecasting reliability of the raw, AKD, and NSGA-II forecasts on the calibration data set (4-day ahead forecast) for five single-model H-EPSs over each individual catchment.







**Figure A8.** Accuracy performance assessment of the raw, AKD, and NSGA-II forecasts (4-day ahead) for five single-model H-EPSs over each sub-catchment of the Gatineau catchment.




**Figure A9.** Comparison of the reliability of the raw, kernel dressed and NSGA-II forecasts on the validation dataset (i.e., 1-3 and 5-7-day ahead forecasts) for five singe-model H-EPSs over all catchments.

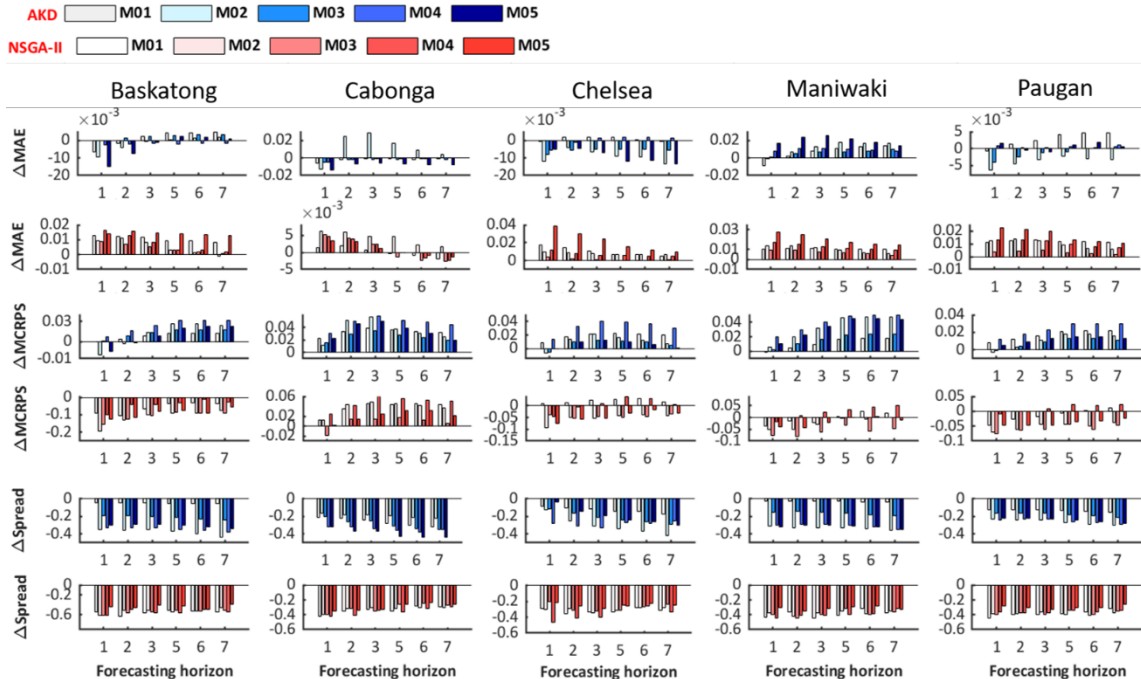

**Figure A10.** Comparison of the $MAE$, $MCRPS$, and ensemble dispersion of the raw, AKD, and NSGA-II forecasts (i.e., 1-3 and 5-7-day ahead forecasts) for five singe-model H-EPSs over all catchments. The x-axis for each sub-plot represents different horizons.

*Author contributions.*

Jing Xu, François Anctil, and Marie-Amélie Boucher designed the theoretical formalism. Jing Xu performed the analytic calculations. Both François Anctil, and Marie-Amélie Boucher supervised the project and contributed to the final version of the manuscript. We would like to thank Emixi Valdez, who provided the data that were used for this project. Funding for this work was provided by FloodNet. Finally, we would also like to thank the ECMWF for maintaining the TIGGE data portal and providing easy access to archived meteorological ensemble forecasts.

*Competing interests.*

I, Jing Xu, as the corresponding author, do hereby confirm that this manuscript has no potential conflicts of interest. My institution and I did not receive any payment or services from a third party (government, commercial, private foundation, etc.) for any aspect of the submitted wor. There is also no relationships or activities that readers could perceive to have influenced, or that give the appearance of potentially influencing for this submitted work.





*Acknowledgements.* This work was supported by the Natural Science and Engineering Research Council of Canada (NSERC) Canadian FloodNet (Grant number: NETGP 451456). The authors wish to thank the ECMWF for maintaining the TIGGE data portal and providing easy access to archived meteorological ensemble forecasts. HOOPLA is an open-source MATLAB toolbox available through GitHub.





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
