# Peer review of "Exploring hydrologic post-processing of ensemble streamflow forecasts based on Affine kernel dressing and Nondominated sorting genetic algorithm II"

_Hydrology and Earth System Sciences, 2020_

## Referee Comment (RC1) · Anonymous Referee #1 · 4 Oct 2020

This paper explores the post-processing of ensemble forecasts using two methods: Affine kernel dressing (AKD) and Non-dominated sorting genetic algorithm II (NSGA-II). The paper concludes on, in general, a better performance of the NSGA-II method. The topic is relevant and interesting to the forecasting community. However, the paper is sometimes too concise and needs some additional clarifications. The presentation and interpretation of the figures is too brief, and many aspects seem to remain undiscussed or unexplored. Overall, the paper is well written, but, in some places, it reads a bit strangely, and I would suggest a review of the use of the English language.

[Figure]

General questions and remarks:

- The aim of the paper should be more clearly stated already (and earlier) in the Introduction. My impression is that we discover the aim of the study while reading the methods and results (for instance, line 262). I also struggled to find out what the novelty of the paper is, with regards to other existing post-processing techniques in the literature. What is the additional (scientific or operational) value of the paper?

- Concerning the Introduction, I found it very difficult to follow the argumentation, since I could not see the direct links between paragraphs, and, most importantly, why the authors were raising, and long discussing, the issue of "sources of uncertainty": if a statistical post-processor is going to be applied, what difference does it make if one, previously, in the raw ensemble, quantified all sources of uncertainty, or, for instance, all but one source of uncertainty? Wouldn't the post-processor work equally well if we had 50 ensemble members from each hydrological model instead of 50x50 members?

- Also in the Introduction, overall, I think the key concepts are not introduced very clearly and just loosely thrown in the sentences. For a reader not used to the techniques, it becomes uncomprehensive. For instance, the whole paragraph on lines 49-65 reads very confusing to me. We read about "bias-corrected ensemble member", "normally distributed data", "predictive weights", or "other dressing parameters", without much explanation about what these terms mean.

- Then from line 66 onwards, it is not clear why it is novel to apply NSGA-II and compare it to a kernel-based dressing method. What are the advantages of using NSGA-II? Line 70: what "different conceptualizations" are we talking about? Line 74: what do you mean by "credibility"?

- I think the authors should completely re-write the Introduction, and think about better presenting the literature, the novel aspect of the paper and the questions the paper wants to answer (i.e., its aim). Some review of the literature presented in the "methods" section 3.2 (page 9, lines 179-204) should go to the Introduction to better explain the

reader why using NSGA-II could be considered a novel aspect in this paper.

- The paper investigates post-processing of ensemble forecasts based on 5 hydrological models and 5 sub-catchments in Canada. However, there is nothing in the paper that discusses the differences in performance among models and sub-catchments? What drives a better/worse performance of the post-processors used in the study? I missed some reflexions about this issue, which would certainly increase the value of the paper. Without this reflexion, and without aggregated (averages) results, I do not understand very well the usefulness of carrying out the study over 5 models and 5 sub-catchments. What does this diversity of applications bring to the analysis?

- I found the distinction between training and validation datasets and criteria very confusing . For instance, we present MCRPS as a validation criteria (section 3.3), but it is then said it is used in calibration (line 269). It is also not clear to me why we do not have a calibration for each lead time. What is the impact of using one unique lead time for calibration?

- Much of the justification for the selection of the study area comes from its operational role in reservoir management. However, the post-processing application presented in the paper is based on a "non-operational" context: the parameters of the post-processor are calibrated over the entire data available (not over a split sample) for a given lead time (4 days) and validated over different lead times. Operationally, though, a forecaster would have to calibrate the post-processor over a long series of past pairs of forecasts and observations, and apply it to a different set of real-time forecast (for which the observations are not yet available). What are the implications of the method proposed for an operational service? Would the operational service be fine with a post-processing that is optimized for a 4-day lead time? Is that the lead-time that most count for the service when forecasting over these catchments? Maybe some lines of discussion would be interesting in the final section of the paper.

Specific questions and remarks:

- lines 23-24: these sentences are not very clear to me.

- line 38: what are the three main sources mentioned?

- line 47-48: what are the implications of autocorrelation in the post-processing? Besides, aren't meteorological forecasts also auto-correlated? Why is it specifically a problem to hydrological forecasts?

- Fig. 2: I understand these are daily streamflow (it is written: mm/day) averaged over each month, and not monthly streamflows. Is that so? The caption should state the period over which the averages were obtained.

- Table 1: I do not understand the data on reservoir area: why it is important to this paper? Furthermore, I do not understand all these physical and climatic data provided: if the results are not going to be interpreted according to the characteristics of the catchments, why are these characteristics presented in the table? In what do they influence the results?

- Line 122: why have you chosen 5 models and why not work with the 20 models? If this is a matter of computational time, could you explain it to the reader? How long it takes to post-process one single model H-EPS?

- Line 136: I am used to forecast post-processing, but not with the term "ensemble interpretation method" or "interpreted ensemble (line 157). I would be happy with more explanations here.

- Line 147: correct English

- Line 169: what is this rule of thumb? Please, clarify.

- Equations, overall: it seems to me that not all terms are always defined, explained after the equations where they are presented. $r_1$, $r_2$, $s_1$, $s_2$, etc. $z_i$ is lower case in equation 10 but upper case in equation 11. a is alpha (line 169)? Please, check the equations and the way terms are presented.

[Figure]

- Line 175: "Eq. (6) can be further defined", should maybe be replaced to "can be re-written"

- Line 205: X(t) was already defined in line 143. Please, check.

- Line 191, 192: I tend not to agree with the authors. I think "accuracy" is what is first of all searched when issuing a forecast at a given day for a short lead time such as 7 days. This is specially the case for flood events, for instance. Please, explain your arguments.

- Lines 195-196: not very clear to me. Hydrologists may rely on NSE, but for simulations (long time series), not necessarily for forecasters. Please, clarify.

- Line 200: I do not understand "elitist". Please, clarify.

- Line 215: the concept of crowding distance was not clear to me. Please, clarify.

- line 235: was the MCRPS calculated using empirical distributions or a fitted theoretical distribution? Please, clarify.

- line 238: why do you need both, MAE and MSE?

- line 248-249: I do not understand why the Taylor diagram is mentioned here. Did you use it? How? Can you explain it?

- Figure 3: where do we find "w" in the text (output of NSGA-II in the figure)?

- lines 287-288: it is not unexpected that forecast performance decreases with lead time. I do no understand why it is "revealed" here. Please, clarify.

- lines 293-294: check for the English language.

- line 324: delete "In the meanwhile,"

- line 335: I understand that "error growth" is usually depicted as an increase in spread with lead time and decrease in accuracy. Why should it be maintained for a single model H-EPS if the post-processor was calibrated for 4 days of lead time only and

applied to other lead times? Please, clarify.

- Figure 9 is not explained in the text (notably the number of lines in each graph). Also, why AKD seems to work well with M05?

- line 343-344: not clear; please, revise it.

- line 345: figure 10 shows much more than spread. Please, clarify when presenting (fully) the figure.

- Figure 10 is very difficult to read. It is not clear (B&W print) which graph is AKD, which is NSGA-II. We can barely see what is inside the figure. I think it needs to be re-designed.

- Overall, terminology could be uniformed (ex., use of AKD)

- It is a pity that the paper does not have a discussion section. I would suggest the authors to introduce one, commenting further the results obtained, comparing post-processing performance among catchments (i.e., geographic location) and hydrological models in a summarized way. This piece of work is missing in the paper and would better justify the use of several catchments and models in the analysis.

―――――――――――――――――

---

## Referee Comment (RC2) · Anonymous Referee #2 · 14 Oct 2020

Dear authors,

Thank you for presenting this interesting work. With application of both EnkF for hydrologic initial condition uncertainty, and hydrologic post-processing, this is a potentially valuable case study on hydrologic ensemble prediction. The paper is overall clearly written, particularly the Data and Methodology sections.

My main questions and concerns are the following:

The objective(s) of the research is(are) in my view not clearly stated, nor the intended

contribution to the literature. Could the authors describe these?

This perhaps also makes the literature review rather general, not zooming-in to identify a gap or under-represented aspects/applications of ensemble prediction, or a particular forecast challenge in the case study catchment.

As I understand, post-processing of the meteorological ensemble forecasts was not done. Could the authors comment in the paper on the performance of the meteorological ensemble forecasts and state the reason for not also applying meteorological post-processing?

Based on Figure 4, presenting one forecast, I do not understand how it can be concluded that the members generated from the meteorological eps as forcing are not fully interchangeable, which is the basis of applying weights with NSGA-II. Perhaps that the video (I am sorry I could not find it, this is probably my omission) shows this, but this is not explicitly stated in the paper.

I do not understand why the authors choose to calibrate the post-processors on only one forecast horizon (day-4) and validate on the other horizons (1-3, 5-7). Because of a generally present decrease of skill with increasing forecast horizon, usually a post-processor is calibrated for each forecast horizon separately. It also seems that the analysis period for which re-forecasts have been prepared has not been split in a calibration and validation period (or a leave-one-out approach). There may be good reasons for choosing this approach, e.g. stemming from catchment or application characteristics versus limited data availability, or as a research objective, but I missed the explanation in the paper. Could the authors perhaps explain the chosen calibration/validation approach?

Lastly, I would kindly encourage the authors to expand the presentation and interpretation/discussion of the results. For example, why present the 5 sub-catchments, what should we learn from the results? What about the inflows to the reservoirs? Why present the 5 single h-eps, what should we learn from the results? Why not assess

the performance of the combined grand multi-model ensemble? How does the performance of the raw and post-processed forecasts compare with the performance of a reference forecast such as climatology or persistence (forecast skill)?

Detailed comments:

Introduction: Could you add explanation why AKD and NSGA-II have been chosen for this research? (line 51)

Introduction, data description, and/or Results section: Could you comment on observational uncertainty?

Line 92: Does the analysis of forecast performance take into account these different flood generating processes, and related seasonality? Would be interesting.

Line 99: In this section, kindly add some information on catchment response time to rainfall/snow melt, and travel time (routing), to inform us about potential forecast lead times without meteorological forecasts as forcing.

Line 104: Could you mention why inflow to the reservoirs is not measured (for some reservoirs), and how the inflow time series have been constructed?

Line 105/106: Could you briefly describe the observational network, and methods used to create sub-basin average precipitation? (to inform observational uncertainty, and perhaps a reason for not going for meteorological forecast post-processing)

Line 168: I think we are missing here, what the parameters are optimised on. From the later paragraph on Experimental set-up it seems that parameterisation of AKD was done by minimising MCRPS.

Lines 224-226: Kindly explain why these parameter values were chosen, and if a sensitivity analysis was done? Was the maximum evolution runs a result of a stopping criterion? If so, please mention this.

Lines 272-273: Please introduce the use of a moving window in Section 3.2, and expand explanation. Also the mentioning here of operational requirements is interesting and further explanation and discussion would be welcome.

Figure 5: The differences in bias and NSE over the range of the Pareto front are small. Please discuss. What weights are in the weight matrices of these solutions?

Line 338: This is interesting. Could you discuss what could be the reason? Something specific about Model M05?

Editorials

Lines 25-27. Deterministic systems do not asses/quantify uncertainty, so the superiority question, I think, did not concern uncertainty quantification, that difference is simply a given. The superiority question concerned more the value when using the forecasts in decision making, and ensemble mean versus deterministic forecast performance.

Figure 1: Please indicate in the map more clearly the main river reach and flow direction.

Lines 82-84: Not clear from this sentence if in Section 4 the results are analysed for each model individually first (not taking into account model structure uncertainty), and then are considered and processed as a grand multi-model ensemble, which does take into account model structure uncertainty.

Lines 137-139: Consider to move up to Introduction for literature review, or down in the sections below. In these few introductory sentences to the methodology I would focus on announcing what was the general approach followed to reach the research objectives. After having introduced the overall methodology, going into the details of the two post-processing methods as of section 3.1 makes sense.

Figure 3: Qobs is not output, so can be left out on the right. It is also not indicated that the output or final results concerns post-processed (interpreted) Qfcsts. The flowchart ending in only one set of post-processed forecasts is confusing, because up to now I was under the impression that AKD and NSGA-II would be used independently to

post-process and hence each method to result in a set of post-processed forecasts, after which the performance of each method will be analysed and compared.

Line 256: Spread Skill plots are announced, but later not presented.

Figure 8: Presenting results in spider plots is a nice idea, but with the scores selected this does not work well, because some scores indicate a better performance with lower value (RMSE) while others the other way around (NSE), and some have a scale only to 1 (NSE) while others are not limited. This makes interpretation of the plots rather difficult.

Lines 357-362: General/Literature - I suggest to delete or move to Introduction

Lines 366-369: Consider to move to Introduction or Methodology

---

## Author Comment (AC1) · 11 Nov 2020

[1]JingXu [1]FrançoisAnctil [2]Marie-AmélieBoucher

[1]Department of Civil and Water Engineering, Université Laval, 1065 avenue de la Médecine, Québec, Québec, Canada;

[2]Department of Civil and Building Engineering, Université de Sherbrooke, 2500 Boul.

[Figure]

de l'Université, Sherbrooke, Québec, Canada

Jing Xu (jing.xu.1@ulaval.ca)

**Reply on 'Review remarks on the paper "Exploring hydrologic post-processing of ensemble stream flow forecasts based on Affine kernel dressing and Nondominated sorting genetic algorithm II"'**

November 11, 2020

Dear Prof. Solomatine and reviewers:

Many thanks for your review comments that we received with respect to our paper. Those valuable comments have significantly enhanced our paper. We have carefully considered and addressed the reviewers' comments and suggestions, which will lead to significant revisions in many parts of the paper. Particularly, we rewrote the introduction section attached at the end of this view letter. Below we hereby provide our point by point responses to each of the reviewer's comments.

**1 General questions and remarks:**

*General question* 1 : The aim of the paper should be more clearly stated already (and earlier) in the Introduction.My impression is that we discover the aim of the study while reading the methods and results (for instance, line 262). I also struggled to find out what the novelty of the paper is, with regards to other existing post-processing techniques in the literature. What is the additional (scientific or operational) value of the paper?

*Response* : Many thanks for your valuable comments. We rewrote the introduction for better clarifying our research aim. Particularly, the novelty of this paper is to emphasize that in the practice, not only quantifying comprehensively, but also communicating the predictive uncertainties in probabilistic forecasts effectively will become an more essential topic progressively. And compared to the conventional post-processing methods, such as Affine kernel dressing (AKD), how the multi objective genetic algorithm (i.e., here, NSGA-II) can open up the opportunities to improve the forecast quality in harmony with the forecasting aims and the specific needs of end-users.

*General question* 2 : Concerning the Introduction, I found it very difficult to follow the argumentation, since I could not see the direct links between paragraphs, and, most importantly, why the authors were raising, and long discussing, the issue of "sources of uncertainty": if a statistical post-processor is going to be applied, what difference does it make if one, previously, in the raw ensemble, quantified all sources of uncertainty, or, for instance, all but one source of uncertainty? Wouldn't the post-processor work equally well if we had 50 ensemble members from each hydrological model instead of 50x50 members?

*Response* : Operational forecasters are open to ensemble forecasting methods and

products for assessing the flood in a probabilistic way. Their main concerns are how to comprehensively quantify the predictive uncertainties from different sources as well as how to use the uncertainty information for better decision-making. We rewrote the introduction to build stronger and more logical between paragraphs. In addition to clarifying the different sources of uncertainty in the hydrometeorological forecast chain, we explored the possibility of using NSGA-II for better fitting the end-user's specific needs.

*General question 3* : Also in the Introduction, overall, I think the key concepts are not introduced very clearly and just loosely thrown in the sentences. For a reader not used to the techniques, it becomes uncomprehensive. For instance, the whole paragraph on lines 49-65 reads very confusing to me. We read about "bias-corrected ensemble member", "normally distributed data", "predictive weights", or "other dressing parameters", without much explanation about what these terms mean.

*Response* : Thank you for your comments. We rewrote the introduction to give a better explanation of these terms.

*General question 4* :Then from line 66 onwards, it is not clear why it is novel to apply NSGA-II and compare it to a kernel-based dressing method. What are the advantages of using NSGA-II? Line 70: what "different conceptualizations" are we talking about? Line 74: what do you mean by "credibility"?

*Response* : Thanks! "different conceptualizations" refers that the mechanisms of these two statistical post-processing methods (i.e., kernel-based dressing method and NSGA-II) are different. While the term of "credibility" means "reliability". The two techniques share one similarity from another perspective, which is they can estimate
the probability density directly from the data (i.e., ensemble forecast) without assuming any particular underlying distribution. The advantages of using NSGA-II is to offer the flexibility to improve the forecast quality in harmony with the forecasting aims and the specific needs of end-users

*General question 5* : I think the authors should completely re-write the Introduction, and think about better presenting the literature, the novel aspect of the paper and the questions the paper wants to answer (i.e., its aim). Some review of the literature presented in the "methods" section 3.2 (page 9, lines 179-204) should go to the Introduction to better explain the reader why using NSGA-II could be considered a novel aspect in this paper.

*Response* : Thanks. We rewrote the introduction to emphasize the novel aspect of this study.

*General question 6* : The paper investigates post-processing of ensemble forecasts based on 5 hydrological models and 5 sub-catchments in Canada. However, there is nothing in the paper that discusses the differences in performance among models and sub-catchments? What drives a better/worse performance of the post-processors used in the study? I missed some reflexions about this issue, which would certainly increase the value of the paper. Without this reflexion, and without aggregated (averages) results, I do not understand very well the usefulness of carrying out the study over 5 models and 5 sub-catchments. What does this diversity of applications bring to the analysis?

*Response* : Many thanks for you suggestions. We will add more analysis and discussion for comparing the forecast performance among different models and subcatchments. The emphasis of this study is to highlight that NSGA-II not only improves the forecast performance compared to conventional post-processing methods but also enhance the predictive uncertainty communication by setting multiple specific objective functions from scratch.

*General question* 7 : I found the distinction between training and validation datasets and criteria very confusing. For instance, we present $MCRPS$ as a validation criteria (section 3.3), but it is then said it is used in calibration (line 269). It is also not clear to me why we do not have a calibration for each lead time. What is the impact of using one unique lead time for calibration?

*Response* : Thanks for you comments. We will modify the criteria for calibration and keep the $MCRPS$ as the verifying score. Besides, the skill of flood forecasts fades away with increasing lead time. The target ensemble has a horizon that extends from day 1 to 7. The 4-day-ahead ensemble forecasts issued from each single-model H-EPSs and their corresponding observations are chosen as a training dataset, since it locates in the middle of the forecast horizon as a compromise.

*General question* 8 : Much of the justification for the selection of the study area comes from its operational role in reservoir management. However, the post-processing application presented in the paper is based on a "non-operational" context: the parameters of the post-processor are calibrated over the entire data available (not over a split sample) for a given lead time (4 days) and validated over different lead times. Operationally, though, a forecaster would have to calibrate the post-processor over a long series of past pairs of forecasts and observations, and apply it to a different set of real-time forecast (for which the observations are not yet available). What are the implications of the method proposed for an operational service? Would the operational service be fine with a post-processing that is optimized for a 4-day lead time? Is that

the lead-time that most count for the service when forecasting over these catchments? Maybe some lines of discussion would be interesting in the final section of the paper.

*Response* : Thanks for you valuable suggestions. We will add more discussion and reflexions in the final section to explore what potential benefits will post-processing techniques will bring to the operational services.

**2 Specific questions and remarks:**

*Specific question* 1 : lines 23-24: these sentences are not very clear to me.

*Response* : Thanks. We will rewrote the abstract as well to make it more clear.

*Specific question* 2 : line 38: what are the three main sources mentioned?

*Response* : These different sources of uncertainty are related to deficiencies in the: (1) meteorological forcing; (2) mis-specified hydrologic initial and boundary conditions; (3) inherent hydrologic model structure errors, and biased estimated parameters.

*Specific question* 3 : line 47-48: what are the implications of autocorrelation in the post-processing? Besides, aren't meteorological forecasts also auto-correlated? Why is it specifically a problem to hydrological forecasts?

*Response* : Yes, the autocorrelation is the problem for both meteorological forecasts

and hydrological forecasts. We deleted this description in the updated version of introduction.

$Specific\ question\ 4$ : Fig. 2: I understand these are daily streamflow (it is written: mm/day) averaged over each month, and not monthly streamflows. Is that so? The caption should state the period over which the averages were obtained.

$Response$ : Yes, these are daily streamflow (mm/day) averaged over each month. We will modify the caption. Thanks.

$Specific\ question\ 5$ : Table 1: I do not understand the data on reservoir area: why it is important to this paper? Furthermore, I do not understand all these physical and climatic data provided: if the results are not going to be interpreted according to the characteristics of the catchments, why are these characteristics presented in the table? In what do they influence the results?

$Response$ : Thanks. We will modify this table to keep the useful characteristics especially for this study.

$Specific\ question\ 6$ : Line 122: why have you chosen 5 models and why not work with the 20 models? If this is a matter of computational time, could you explain it to the reader? How long it takes to post-process one single model H-EPS?

$Response$ : All of these 5 models are lumped models. They are representative of the 20 models. It is more of a layout concern rather than computation time.

*Specific question* 7 : Line 136: I am used to forecast post-processing, but not with the term "ensemble interpretation method" or "interpreted ensemble (line 157). I would be happy with more explanations here.

*Response* : Thanks. We will add more useful explanations here.

*Specific question* 8 : Line 147: correct English

*Response* : We will rephrase this content.

*Specific question* 9 : Line 169: what is this rule of thumb? Please, clarify.

*Response* : Thanks. We will add more explanation about this parameter $h_S$ (Silverman, 1986) in the methodology section.

*Specific question* 10 : Equations, overall: it seems to me that not all terms are always defined, explained after the equations where they are presented. r1, r2, s1, s2, etc. zi is lower case in equation 10 but upper case in equation 11. a is alpha (line 169)? Please, check the equations and the way terms are presented.

*Response* : Thanks. We will check all those equations and define all related terms.

*Specific question* 11 : Line 175: "Eq. (6) can be further defined", should maybe be replaced to "can be re-written"

*Response* : Thanks. We will rephrase it as "can be re-written".

*Specific question* 12 : Line 205: $X_t$ was already defined in line 143. Please, check.

*Response* : Thanks. Yes, it is. We will remove the description of $X_t$ in Line 205.

*Specific question* 13 : Line 191, 192: I tend not to agree with the authors. I think "accuracy" is what is first of all searched when issuing a forecast at a given day for a short lead time such as 7 days. This is specially the case for flood events, for instance. Please, explain your arguments.

*Response* : Thanks. Yes, probabilistic forecasts must be, first of all, accurate. We will rephrase this paragraph.

*Specific question* 14 : Lines 195-196: not very clear to me. Hydrologists may rely on $NSE$, but for simulations (long time series), not necessarily for forecasters. Please, clarify.

*Response* : Thanks. We will add more clarification here.

*Specific question* 15 : Line 200: I do not understand "elitist". Please, clarify.

*Response* : The Nondominated sorting genetic algorithm II (NSGA-II; Deb et al. (2002)) is admitted as a fast and elitist multiobjective genetic algorithm, adopted for searching for the Pareto solution set. I will add more description about the "elitism" of NGSA-II in section 3.2.

*Specific question* 16 : Line 215: the concept of crowding distance was not clear to me. Please, clarify.

*Response* : Thanks. I will add more description about the "crowding distance" of NGSA-II in section 3.2.

*Specific question* 17 : line 235: was the $MCRPS$ calculated using empirical distributions or a fitted theoretical distribution? Please, clarify.

*Response* : Thanks. We will clarify this in the section 3.3.

*Specific question* 18 : line 238: why do you need both, $MAE$ and $MSE$?

*Response* : Thanks. We will only keep $MSE$ later.

*Specific question* 19 : line 248-249: I do not understand why the Taylor diagram is mentioned here. Did you use it? How? Can you explain it?

*Response* : Thanks. We will remove this short description of the Taylor diagram.

*Specific question* 20 : Figure 3: where do we find "$w$" in the text (output of NSGA-II in the figure)?

*Response* :Thanks. We will redraw this flowchart and provide more details.

*Specific question* 21 : lines 287-288: it is not unexpected that forecast performance decreases with lead time. I do no understand why it is "revealed" here. Please, clarify.

*Response* :Thanks. We will rephrase this sentence.

*Specific question* 22 : lines 293-294: check for the English language.

*Response* : Thanks. We will check the English here.

*Specific question* 23 : line 324: delete "In the meanwhile,"

*Response* : Thanks. We will delete "In the meanwhile,".

*Specific question* 24 : line 335: I understand that "error growth" is usually depicted as an increase in spread with lead time and decrease in accuracy. Why should it be maintained for a single model H-EPS if the post-processor was calibrated for 4 days of lead time only and applied to other lead times? Please, clarify.

*Response* : Thanks. The target ensemble has a horizon that extends from day 1 to 7. The 4-day-ahead ensemble forecasts issued from each single-model H-EPSs and their corresponding observations are chosen as a training dataset, since it locates in the middle of the forecast horizon as a compromise. We will add more clarification here.

*Specific question* 25 : Figure 9 is not explained in the text (notably the number of lines
in each graph). Also, why AKD seems to work well with M05?

*Response* : We will add more explanations in text about Figure 9.

*Specific question* 26 : line 343-344: not clear; please, revise it.

*Response* : Thanks. We will rephrase this paragraph.

*Specific question* 27 : line 345: figure 10 shows much more than spread. Please, clarify when presenting (fully) the figure.

*Response* : Thanks. We will modify the description and analysis for Figure 10.

*Specific question* 28 : Figure 10 is very difficult to read. It is not clear (BW print) which graph is AKD, which is NSGA-II. We can barely see what is inside the figure. I think it needs to be re-designed.

*Response* : We will enlarge the figure size and add more analysis corresponding to Figure 10.

*Specific question* 29 : Overall, terminology could be uniformed (ex., use of AKD).

*Response* : Thanks. We will uniform the terminology in the whole paper.

*Specific question* 30 : It is a pity that the paper does not have a discussion section. I

would suggest the authors to introduce one, commenting further the results obtained, comparing post-processing performance among catchments (i.e., geographic location) and hydrological models in a summarized way. This piece of work is missing in the paper and would better justify the use of several catchments and models in the analysis.

*Response* : Thanks for this suggestion. We will add further discussion section about: (1) comparing post-processing performance among catchments hydrological models in a summarized way; (2) highlight the novelty and potential benefits that post-processing techniques may bring to the operational services.

**3  Introduction**

Hydrologic forecasting is crucial for flood warning and mitigation (e.g., Shim and Fontane, 2002; Cheng and Chau, 2004), water supply operation and reservoir management (e.g., Datta and Burges, 1984; Coulibaly et al., 2000; Boucher et al., 2011), navigation, and other related activities. Sufficient risk awareness, enhanced disaster preparedness in the flood mitigation measures, and strengthened early warning systems are crucial in reducing the weather-related event losses. Hydrologic models are typically driven by dynamic meteorological models in order to issue forecasts over a medium range horizon of 2 to 15 days (Cloke and Pappenberger, 2009). This kind of coupled hydrometeorologic forecasting systems are admitted as effective tools to issue longer lead times. Inherent in the coupled hydrometeorologic forecasting systems, some predictive uncertainties are then inevitable given the limits of knowledge and available information (Ajami et al., 2007). In fact, those uncertainties occur all along the different steps of the hydrometeorological modeling chain (e.g., Liu and Gupta,

2007; Beven and Binley, 2014). These different sources of uncertainty are related to deficiencies in the meteorological forcing, mis-specified hydrologic initial and boundary conditions, inherent hydrologic model structure errors, and biased estimated parameters (e.g., Vrugt and Robinson, 2007; Ajami et al., 2007; Salamon and Feyen, 2010; Thiboult et al., 2016). Among most cases, a single deterministic forecasts turns out to be way more insufficient.

Many substantive theories have been proposed in order to quantify and reduce the different sources of cascading forecast uncertainties and to add good values to flood forecasting and warning. Among them, the superiority of ensemble forecasting systems in quantifying the propagation of predictive uncertainties (over deterministic systems) is now well established (e.g., Cloke and Pappenberger, 2009; Palmer, 2002; Seo et al., 2006; Velázquez et al., 2009; Abaza et al., 2013; Wetterhall et al., 2013; Madadgar et al., 2014). Numerous challenges have been well tackled, for example: (1) meteorological ensemble prediction systems (M-EPSs) (e.g., Palmer, 1993; Houtekamer et al., 1996; Toth and Kalnay, 1997) are refined and operated worldwide by national agencies such as the European Centre for Medium-Range Weather Forecasts (ECMWF), the National Center for Environmental Prediction (NCEP), the Meteorological Service of Canada (MSC), and more; (2) the forecast accuracy is highly improved by adopting higher resolution data collection and assimilation. Sequential data assimilation techniques, such as the particle filter (e.g., Moradkhani et al., 2012; Thirel et al., 2013) and the ensemble Kalman filter (e.g., Evensen , 1994; Reichle et al., 2002; Moradkhani et al, 2005; McMillan et al., 2013) provide an ensemble of possible re-initializations of the initial conditions, expressed in the hydrologic model as state variables, such as soil moisture, groundwater level and so on; (3) forecasting skills of the coupled hydrometeorologic forecasting systems are also improved by tracking predictive errors using the full uncertainty analysis. Multimodel schemes were proposed to increase performance and decipher structural uncertainty (e.g., Duan et al., 2007; Fisher et al., 2008; Weigel et al., 2008; Najafi et al., 2011; Velázquez et al., 2011; Marty et al., 2015; Mockler et al., 2016). Thiboult et al. (2016) compared many H-EPS, accounting for the three

main sources of uncertainties located along the hydrometeorological modeling chain. They pointed out that EnKF probabilistic data assimilation provided most of the dispersion for the early forecasting horizons but failed in maintaining its effectiveness with increasing lead times. A multimodel scheme allowed sharper and more reliable ensemble predictions over a longer forecast horizon; (4) statistical hydrologic post-processing component is added in the H-EPS for rectifying biases and dispersion errors (i.e., too narrow/too large) are numerous, as reviewed by Li et al. (2017). It is noteworthy that many hydrologic variables, such as discharge, follow a skewed distribution (i.e., low probability associated to the highest streamflow values), which complicates the task. Usually, in a hydrologic ensemble prediction system (H-EPS) framework (e.g., Schaake et al., 2007; Cloke and Pappenberger, 2009; Velázquez et al., 2009; Boucher et al., 2012; Abaza et al., 2017), the post-processing procedure over the atmospheric input ensemble is often referred as pre-processing, while post-processing aims at improving the hydrologic ensemble forecasting outputs.

However, another challenge still remains: how to improve the human interpretation of probabilistic forecasts and the communication of integrated ensemble forecast products to end-users (e.g., operational hydrologists, water managers, local conservation authorities, stakeholders and other relevant decision makers). This step is considered to be the key part of facilitating the implementation of H-EPS in real-time operational forecasting effectively. Buizza et al. (2007) emphasized that both functional and technical qualities are supposed to be assessed for evaluating the overall forecast value of a hydrometeorologic forecasts. Ramos et al. (2010) further noted that the best way to communicate probabilistic forecast and interpret its usefulness should be in harmony with the goals of the forecasting system and the specific needs of end-users. She also demonstrated the main achievements from two studies obtained from a Member States workshop (Thielen et al., 2005) role-play game and another survey to explore the users' risk perception of forecasting uncertainties and how they dealt with uncertain forecasts for decision-making. The results revealed that there is still space for enhancing the forecasters' knowledge and experience on bridge the community gap between

predictive uncertainties quantification and effective decision-making.

Hence, in practice, which forecast quality impacts a given decision the most? Different end-users share their unique requirements: Crochemore et al. (2017) produced the seasonal streamflow forecasting by conditioning climatology with precipitations indices (SPI3). Forecast reliability, sharpness (i.e., spread), overall performance and low-flow event detection were verified to assess the conditioning impact. In some cases, the reliability and sharpness could be improved simultaneously while more often, there was a trade-off between them. Another IMPREX project conduct an optimization for the reservoir-based hydropower production to explore the relationship between the forecast quality and economic values. They found that an over-estimation comes along with more penalization. In the operational filed, not only quantifying, but also communicating the predictive uncertainties in probabilistic forecasts will become an more essential topic progressively.

[revised manuscript text omitted]

---

## Author Comment (AC2) · 11 Nov 2020

[1]JingXu [1]FrançoisAnctil [2]Marie-AmélieBoucher

[1]Department of Civil and Water Engineering, Université Laval, 1065 avenue de la Médecine, Québec, Québec, Canada;

[2]Department of Civil and Building Engineering, Université de Sherbrooke, 2500 Boul.

[Figure]

de l'Université, Sherbrooke, Québec, Canada

Jing Xu (jing.xu.1@ulaval.ca)

[Figure]
Interactive
comment

**Reply on 'Referee comment on "Exploring hydrologic post-processing of ensemble stream flow forecasts based on Affine kernel dressing and Nondominated sorting genetic algorithm II"'**

November 11, 2020

Dear Prof. Solomatine and reviewers:

Many thanks for your review comments that we received with respect to our paper. Those valuable comments have significantly enhanced our paper. We have carefully considered and addressed the reviewers' comments and suggestions, which will lead to significant revisions in many parts of the paper. Particularly, we rewrote the introduction section attached at the end of this view letter. Below we hereby provide our point by point responses to each of the reviewer's comments.

**1  General questions and remarks:**

*General question* 1 : The objective(s) of the research is(are) in my view not clearly stated, nor the intended contribution to the literature. Could the authors describe these?

*Response* : Many thanks for your comments. We rewrote the introduction for better clarifying our research objective. Particularly, the novelty of this paper is to emphasize that in the practice, not only quantifying comprehensively, but also communicating the predictive uncertainties in probabilistic forecasts effectively will become an more essential topic progressively. And compared to the conventional post-processing methods, such as Affine kernel dressing (AKD), how the multi objective genetic algorithm (i.e., here, NSGA-II) can open up the opportunities to improve the forecast quality in harmony with the forecasting aims and the specific needs of end-users.

*General question* 2 : This perhaps also makes the literature review rather general, not zooming-in to identify a gap or under-represented aspects / applications of ensemble prediction, or a particular forecast challenge in the case study catchment.

*Response* : Thanks for your comments. We re-designed the introduction structure and added more literature review from the operational perspective. Operational forecasters are open to ensemble forecasting methods and products for assessing the flood in a probabilistic way. The main challenges for them are how to comprehensively quantify the predictive uncertainties from different sources as well as how to use the uncertainty information for better decision-making. We rewrote the introduction to build stronger and more logical between paragraphs. In addition to clarifying the different sources of uncertainty in the hydrometeorological forecast chain, we explored the possibility of using NSGA-II for better fitting the end-user's specific needs.

*General question* 3 : As I understand, post-processing of the meteorological ensemble forecasts was not done. Could the authors comment in the paper on the performance of the meteorological ensemble forecasts and state the reason for not also applying meteorological post-processing?

*Response* : Thanks. We rewrote the introduction to give a better explanation: "It is noteworthy that many hydrologic variables, such as discharge, follow a skewed distribution (i.e., low probability associated to the highest streamflow values), which complicates the task. Usually,in a hydrologic ensemble prediction system (H-EPS) framework (e.g., Schaake et al., 2007; Cloke and Pappenberger, 2009;Velázquez et al., 2009; Boucher et al., 2012; Abaza et al., 2017), the post-processing procedure over the atmospheric input ensemble is often referred as pre-processing, while post-processing aims at improving the hydrologic ensemble forecasting outputs."

*General question* 4 : Based on Figure 4, presenting one forecast, I do not understand how it can be concluded that the members generated from the meteorological eps as forcing are not fully interchangeable, which is the basis of applying weights with NSGA-II. Perhaps that the video (I am sorry I could not find it, this is probably my omission) shows this, but this is not explicitly stated in the paper.

*Response* : Thanks! We will prepare and upload the animation (too large) that can reveal the phenomenon of not fully interchangeable of the meteorological forcing members for your consideration.

*General question* 5 : I do not understand why the authors choose to calibrate the post-processors on only one forecast horizon (day-4) and validate on the other horizons

(1-3, 5-7). Because of a generally present decrease of skill with increasing forecast horizon, usually a post-processor is calibrated for each forecast horizon separately. It also seems that the analysis period for which re-forecasts have been prepared has not been split in a calibration and validation period (or a leave-one-out approach). There may be good reasons for choosing this approach, e.g. stemming from catchment or application characteristics versus limited data availability, or as a research objective, but I missed the explanation in the paper. Could the authors perhaps explain the chosen calibration/validation approach?

*Response* : Thanks for you comments. The skill of flood forecasts fades away with increasing lead time. The target ensemble has a horizon that extends from day 1 to 7. The 4-day-ahead ensemble forecasts issued from each single-model H-EPSs and their corresponding observations are chosen as a training dataset, since it locates in the middle of the forecast horizon as a compromise. We will explain further the division of the calibration and validation period in the section 3.4.

*General question* 6 : Lastly, I would kindly encourage the authors to expand the presentation and interpretation / discussion of the results. For example, why present the 5 sub-catchments, what should we learn from the results? What about the inflows to the reservoirs? Why present the 5 single h-eps, what should we learn from the results? Why not assess the performance of the combined grand multi-model ensemble? How does the performance of the raw and post-processed forecasts compare with the performance of a reference forecast such as climatology or persistence (forecast skill)?

*Response* : Thanks for this suggestion. We will add further discussion about: (1) comparing post-processing performance among catchments hydrological models in a summarized way; (2) highlight the novelty and potential benefits that post-processing techniques may bring to the operational services. And we have already published another article mainly focus on exploring "the hydrological post-processing of streamflow forecasts issued from multimodel ensemble prediction systems". Please check: $Xu,$ $J., Anctil, F. and Boucher, M.A., 2019 \ Hydrological \ post-processing \ of \ streamflow$ $forecasts \ issued \ from \ multimodel \ ensemble \ prediction \ systems, \ J. \ Hydrol., \ 578,$ $p.124002, \ https://doi.org/10.1016/j.jhydrol. \ 2019. \ 124002.$ if this may interest you.

**2 Detailed comments:**

$Detailed \ comments \ 1$ : Introduction: Could you add explanation why AKD and NSGA-II have been chosen for this research? (line 51)

$Response$ : Thank you for your comments. We rewrote the introduction and further explained the reason for selecting these two post-processing techniques.

$Detailed \ comments \ 2$ : Introduction, data description, and / or Results section: Could you comment on observational uncertainty?

$Response$ : Thanks. We plan to add further description about the observational uncertainty in the section 2.

$Detailed \ comments \ 3$ : Line 92: Does the analysis of forecast performance take into account these different flood generating processes, and related seasonality? Would be interesting.

$Response$ : Seasonality diversity analysis was not the the focus of our previous script.

But I will add further comparison and analysis in the result section (Section 4).

*Detailed comments* 4 : Line 99: In this section, kindly add some information on catchment response time to rainfall/snow melt, and travel time (routing), to inform us about potential forecast lead times without meteorological forecasts as forcing.

*Response* : Many thanks for you suggestions. We will add more detailed information on catchment response tie to rainfall/snow melt, and travel time.

*Detailed comments* 5 : Line 104: Could you mention why inflow to the reservoirs is not measured (for some reservoirs), and how the inflow time series have been constructed?

*Response* : Thanks. We will add further clarification about how the inflow times series have been constructed in Section 2.

*Detailed comments* 6 : Line 105/106: Could you briefly describe the observational network, and methods used to create sub-basin average precipitation? (to inform observational uncertainty, and perhaps a reason for not going for meteorological forecast post-processing).

*Response* : Thanks. We will add more detailed description about the observational network to inform the observation uncertainty.

*Detailed comments* 7 : Line 168: I think we are missing here, what the parameters are optimised on. From the later paragraph on Experimental set-up it seems that parameterization of $AKD$ was done by minimising $MCRPS$.

[Figure]

*Response* : Yes, the parameterization of $AKD$ was done by minimizing $MCRPS$. We will further clarify these relevant parameters in Section 3.1.

*Detailed comments* 8 : Lines 224-226: Kindly explain why these parameter values were chosen, and if a sensitivity analysis was done? Was the maximum evolution runs a result of a stopping criterion? If so, please mention this.

*Response* : Thank you for your comments. We will add further description about NSGA-II method in Section 3.2.

*Detailed comments* 9 : Lines 272-273: Please introduce the use of a moving window in Section 3.2, and expand explanation. Also the mentioning here of operational requirements is interesting and further explanation and discussion would be welcome.

*Response* : Thanks. We will give further explanation about why we chose using "moving window" in the experiments from the operational perspective.

*Detailed comments* 10 : Figure 5: The differences in bias and $NSE$ over the range of the Pareto front are small. Please discuss. What weights are in the weight matrices of these solutions?

*Response* : Thanks. We will give further analysis about the weights obtained from the Pareto solutions.

*Detailed comments* 11 : Line 338: This is interesting. Could you discuss what could

be the reason? Something specific about Model M05?

*Response* : Thanks. We will add more analysis about the potentail reason why the predictive distributions of the kernel dressed ensemble are the most reliable for model M05 over almost all individual catchments.

**3  Editorials:**

*Editorials* 1 : Lines 25-27. Deterministic systems do not asses/quantify uncertainty, so the superiority question, I think, did not concern uncertainty quantification, that difference is simply a given. The superiority question concerned more the value when using the forecasts in decision making, and ensemble mean versus deterministic forecast performance.

*Response* : Many thanks for this comment. Yes, the main challenge for the operational forecasters are how to comprehensively quantify the predictive uncertainties from different sources as well as how to use the uncertainty information for better decision-making. The essential topic here is to bridge the gap between the "theory" (i.e., accuracy, reliability, etc.) and the "practice" (i.e., decision-driven trade-offs). We would like to emphasize this point in this paper.

*Editorials* 2 : Figure 1: Please indicate in the map more clearly the main river reach and flow direction.

*Response* : Thanks. We will refine the map in Figure 1.

*Editorials* 3 : Lines 82-84: Not clear from this sentence if in Section 4 the results are analysed for each model individually first (not taking into account model structure uncertainty), and then are considered and processed as a grand multi-model ensemble, which does take into account model structure uncertainty.

*Response* : Thanks. We will add further explanation here about how we consider the model structure uncertainty.

*Editorials* 4 : Lines 137-139: Consider to move up to Introduction for literature review, or down in the sections below. In these few introductory sentences to the methodology I would focus on announcing what was the general approach followed to reach the research objectives. After having introduced the overall methodology, going into the details of the two post-processing methods as of section 3.1 makes sense.

*Response* : Thank you for your suggestion. We plan to re-design the paper structure carefully.

*Editorials* 5 : Figure 3: Qobs is not output, so can be left out on the right. It is also not indicated that the output or final results concerns post-processed (interpreted) Qfcsts. The flowchart ending in only one set of post-processed forecasts is confusing, because up to now I was under the impression that AKD and NSGA-II would be used independently post-process and hence each method to result in a set of post-processed forecasts, after which the performance of each method will be analysed and compared.

*Response* : Thank you for proposing your concerns about the flowchart. We will refine

this flowchart.

*Editorials* 6 : Line 256: Spread Skill plots are announced, but later not presented.

*Response* : Thanks. The Spread Skill plot ($SSP$) were referd as "spread" in the corresponding results (i.e., Figure 8 and 10). We will rephrase the text to make it easier to track.

*Editorials* 7 : Figure 8: Presenting results in spider plots is a nice idea, but with the scores selected this does not work well, because some scores indicate a better performance with lower value ($RMSE$) while others the other way around ($NSE$), and some have a scale only to 1 ($NSE$) while others are not limited. This makes interpretation of the plots rather difficult.

*Response* : Thank you for your comment. We will re-plot those figures to fix the scale issue.

*Editorials* 8 : Lines 357-362: General/Literature - I suggest to delete or move to Introduction.

*Response* : Thanks. We will delete the general literature review here.

*Editorials* 9 : Lines 366-369: Consider to move to Introduction or Methodology.

*Response* : Thanks. We will move these descriptions here to the methodology section.

[Figure]

**4 Introduction**

Hydrologic forecasting is crucial for flood warning and mitigation (e.g., Shim and Fontane, 2002; Cheng and Chau, 2004), water supply operation and reservoir management (e.g., Datta and Burges, 1984; Coulibaly et al., 2000; Boucher et al., 2011), navigation, and other related activities. Sufficient risk awareness, enhanced disaster preparedness in the flood mitigation measures, and strengthened early warning systems are crucial in reducing the weather-related event losses. Hydrologic models are typically driven by dynamic meteorological models in order to issue forecasts over a medium range horizon of 2 to 15 days (Cloke and Pappenberger, 2009). This kind of coupled hydrometeorologic forecasting systems are admitted as effective tools to issue longer lead times. Inherent in the coupled hydrometeorologic forecasting systems, some predictive uncertainties are then inevitable given the limits of knowledge and available information (Ajami et al., 2007). In fact, those uncertainties occur all along the different steps of the hydrometeorological modeling chain (e.g., Liu and Gupta, 2007; Beven and Binley, 2014). These different sources of uncertainty are related to deficiencies in the meteorological forcing, mis-specified hydrologic initial and boundary conditions, inherent hydrologic model structure errors, and biased estimated parameters (e.g., Vrugt and Robinson, 2007; Ajami et al., 2007; Salamon and Feyen, 2010; Thiboult et al., 2016). Among most cases, a single deterministic forecasts turns out to be way more insufficient.

Many substantive theories have been proposed in order to quantify and reduce the different sources of cascading forecast uncertainties and to add good values to flood forecasting and warning. Among them, the superiority of ensemble forecasting systems in quantifying the propagation of predictive uncertainties (over deterministic systems) is now well established (e.g., Cloke and Pappenberger, 2009; Palmer, 2002; Seo et al., 2006; Velázquez et al., 2009; Abaza et al., 2013; Wetterhall et al., 2013; Madadgar et al., 2014). Numerous challenges have been well tackled, for example: (1) meteorological ensemble prediction systems (M-EPSs) (e.g., Palmer, 1993; Houtekamer et al., 1996; Toth and Kalnay, 1997) are refined and operated worldwide by national agencies such as the European Centre for Medium-Range Weather Forecasts (ECMWF), the National Center for Environmental Prediction (NCEP), the Meteorological Service of Canada (MSC), and more; (2) the forecast accuracy is highly improved by adopting higher resolution data collection and assimilation. Sequential data assimilation techniques, such as the particle filter (e.g., Moradkhani et al., 2012; Thirel et al., 2013) and the ensemble Kalman filter (e.g., Evensen , 1994; Reichle et al., 2002; Moradkhani et al, 2005; McMillan et al., 2013) provide an ensemble of possible re-initializations of the initial conditions, expressed in the hydrologic model as state variables, such as soil moisture, groundwater level and so on; (3) forecasting skills of the coupled hydrometeorologic forecasting systems are also improved by tracking predictive errors using the full uncertainty analysis. Multimodel schemes were proposed to increase performance and decipher structural uncertainty (e.g., Duan et al., 2007; Fisher et al., 2008; Weigel et al., 2008; Najafi et al., 2011; Velázquez et al., 2011; Marty et al., 2015; Mockler et al., 2016). Thiboult et al. (2016) compared many H-EPS, accounting for the three main sources of uncertainties located along the hydrometeorological modeling chain. They pointed out that EnKF probabilistic data assimilation provided most of the dispersion for the early forecasting horizons but failed in maintaining its effectiveness with increasing lead times. A multimodel scheme allowed sharper and more reliable ensemble predictions over a longer forecast horizon; (4) statistical hydrologic post-processing component is added in the H-EPS for rectifying biases and dispersion errors (i.e., too narrow/too large) are numerous, as reviewed by Li et al. (2017). It is noteworthy that many hydrologic variables, such as discharge, follow a skewed distribution (i.e., low probability associated to the highest streamflow values), which complicates the task. Usually, in a hydrologic ensemble prediction system (H-EPS) framework (e.g., Schaake et al., 2007; Cloke and Pappenberger, 2009; Velázquez et al., 2009; Boucher et al., 2012; Abaza et al., 2017), the post-processing procedure over the atmospheric input ensemble is often referred as pre-processing, while post-processing aims at improving

the hydrologic ensemble forecasting outputs.

However, another challenge still remains: how to improve the human interpretation of probabilistic forecasts and the communication of integrated ensemble forecast products to end-users (e.g., operational hydrologists, water managers, local conservation authorities, stakeholders and other relevant decision makers). This step is considered to be the key part of facilitating the implementation of H-EPS in real-time operational forecasting effectively. Buizza et al. (2007) emphasized that both functional and technical qualities are supposed to be assessed for evaluating the overall forecast value of a hydrometeorologic forecasts. Ramos et al. (2010) further noted that the best way to communicate probabilistic forecast and interpret its usefulness should be in harmony with the goals of the forecasting system and the specific needs of end-users. She also demonstrated the main achievements from two studies obtained from a Member States workshop (Thielen et al., 2005) role-play game and another survey to explore the users' risk perception of forecasting uncertainties and how they dealt with uncertain forecasts for decision-making. The results revealed that there is still space for enhancing the forecasters' knowledge and experience on bridge the community gap between predictive uncertainties quantification and effective decision-making.

Hence, in practice, which forecast quality impacts a given decision the most? Different end-users share their unique requirements: Crochemore et al. (2017) produced the seasonal streamflow forecasting by conditioning climatology with precipitations indices (SPI3). Forecast reliability, sharpness (i.e., spread), overall performance and low-flow event detection were verified to assess the conditioning impact. In some cases, the reliability and sharpness could be improved simultaneously while more often, there was a trade-off between them. Another IMPREX project conduct an optimization for the reservoir-based hydropower production to explore the relationship between the forecast quality and economic values. They found that an over-estimation comes along with more penalization. In the operational filed, not only quantifying, but also communicating the predictive uncertainties in probabilistic forecasts will become an more

essential topic progressively.

[revised manuscript text omitted]

1993.

Ramos, M.H., Mathevet, T., Thielen, J. and Pappenberger, F.: Communicating uncertainty in hydro‐meteorological forecasts: mission impossible? Meteorol. Appl., 17(2), pp.223-235, https://doi.org/10.1002/met.202, 2010.

Pappenberger, F., Beven, K.J., Hunter, N.M., Bates, P.D., Gouweleeuw, B.T., Thielen, J., and de Roo, A.P.J.: Cascading model uncertainty from medium range weather forecasts (10 days) through a rainfall-runoff model to flood inundation predictions within the Euro- pean Flood Forecasting System (EFFS), Hydrol. Earth. Syst. Sc., 9, 381–393, https://hal.archives-ouvertes.fr/hal-00304846, 2005.

Perrin, C.: Vers une amélioration d'un modèle global pluie-débit, PhD diss., Institut National Polytechnique de Grenoble-INPG, Grenoble, 287 pp, 2000.

Perrin, C., Michel, C., and Andréassian, V.: Improvement of a parsimonious model for stream-flow simulation, J. Hydrol., 279, 275-289, https://doi.org/10.1016/S0022-1694(03)00225-7, 2003.

Reichle, R., McLaughlin, D.B., and Entekhabi, D.: Hydrologic data assimilation with the ensemble Kalman filter, Mon. Weather Rev., 130, 103-114, https://doi.org/10.1175/1520-0493(2002)130<0103:HDAWTE>2.0.CO;2, 2002.

Roulston, M.S. and Smith, L.A.: Combining dynamical and statistical ensembles. Tellus. A., 55, 16-30, https://doi.org/10.3402/tellusa.v55i1.12082, 2003.

Salamon, P. and Feyen, L.: Disentangling uncertainties in distributed hydrological modeling us-ing multiplicative error models and sequential data assimilation, Water Resour. Res., 46(12), 1-20, https://doi.org/10.1029/2009WR009022, 2010.

Schaffer, J.: Multiple Objective Optimization with Vector Evaluated Genetic Algorithms, Pro-ceedings of the First International Conference on Genetic Algortithms, Lawrence Erlbaum Associates. Inc., 93-100, July 1985.

Seiller, G., Roy, R., and Anctil, F.: Influence of three common calibration metrics on the diagnosis of climate change impacts on water resources, J. Hydrol., 547, 280-295, https://doi.org/10.1016/j.jhydrol.2017.02.004, 2017.

Seiller, G., Anctil, F., and Perrin, C.: Multimodel evaluation of twenty lumped hydrologi-cal models under contrasted climate conditions, Hydrol. Earth. Syst. Sc., 16, 1171-1189, https://doi.org/DOI : 10.5194/hess-1116-1171-2012, 2012.

[revised manuscript text omitted]

---

## Editor Comment (EC1) · Dimitri Solomatine (Editor) · 19 Nov 2020

In my view both referees did really an excellent job, giving insightful comments, and constructive suggestions. In their replies, the authors have shown the eagerness to revise the manuscript accordingly, and indicating that pieces were already rewritten. They are experienced researchers, and it is obvious they have a clear plan for revision. I wish them success.

238, 2020.

---

## Author Response (AR1)

**Reply referee #1 on "Exploring hydrologic post-processing of ensemble streamflow forecasts based on Affine kernel dressing and Nondominated sorting genetic algorithm II"**

Jing Xu[1], François Anctil[1], and Marie-Amélie Boucher[2]

[1]Department of Civil and Water Engineering, Université Laval, 1065 avenue de la Médecine, Québec, Québec, Canada;
[2]Department of Civil and Building Engineering, Université de Sherbrooke, 2500 Boul. de l'Université, Sherbrooke, Québec, Canada

**Correspondence:** Jing Xu (jing.xu.1@ulaval.ca)

Dear Prof. Solomatine and reviewers:

Thank you for your review comments that we received with respect to our paper. Those valuable comments have significantively enhanced our paper. We have carefully considered and addressed the reviewers' comments and suggestions, which to significant revisions in many parts of the paper. Particularly, we re-wrote the ***Introduction*** section attached at the end of this letter. Below we provide a point by point responses to each of the reviewer's comments.

**1 General questions and remarks:**

*General question* 1 : The aim of the paper should be more clearly stated already (and earlier) in the Introduction.My impression is that we discover the aim of the study while reading the methods and results (for instance, line 262). I also struggled to find out what the novelty of the paper is, with regards to other existing post-processing techniques in the literature. What is the additional (scientific or operational) value of the paper?

*Response* :

Thank you for your valuable comments. We re-wrote the ***Introduction*** to clarify our research aim. Particularly, the novelty of this paper is to emphasize that in the practice, not only quantifying comprehensively, but also communicating the predictive uncertainties in probabilistic forecasts effectively will become an essential topic. Compared to conventional post-processing methods, such as Affine kernel dressing (AKD), the multi objective genetic algorithm (i.e., here, NSGA-II) can open up the opportunities to improve the forecast quality in line with the forecasting aims and the specific needs of end-users.

*General question* 2 : Concerning the Introduction, I found it very difficult to follow the argumentation, since I could not see the direct links between paragraphs, and, most importantly, why the authors were raising, and long discussing, the issue of "sources of uncertainty": if a statistical post-processor is going to be applied, what difference does it make if one, previously, in the raw ensemble, quantified all sources of uncertainty, or, for instance, all but one source of uncertainty? Wouldn't the

post-processor work equally well if we had 50 ensemble members from each hydrological model instead of 50x50 members?

*Response* :

Thank you for your comments and questions. Operational forecasters are open to ensemble forecasting methods and products for assessing the flood in a probabilistic way. Their main concerns are how to comprehensively quantify the predictive uncertainties from different sources as well as how to use the uncertainty information for better decision-making. We re-wrote the **_Introduction_** section to build stronger and more logical links between paragraphs. In addition to clarifying the different sources of uncertainty in the hydrometeorological forecast chain, we explored the possibility of using NSGA-II for better fitting the end-user's specific needs.

*General question* 3 : Also in the Introduction, overall, I think the key concepts are not introduced very clearly and just loosely thrown in the sentences. For a reader not used to the techniques, it becomes uncomprehensive. For instance, the whole paragraph on lines 49-65 reads very confusing to me. We read about "bias-corrected ensemble member", "normally distributed data", "predictive weights", or "other dressing parameters", without much explanation about what these terms mean.

*Response* :

Thank you for your comments. We re-wrote the **_Introduction_** section to give a better explanation of these terms.

**_"Line 71: The study is a contribution to probe this topic by exploring hydrological post-processing of ensemble streamflow forecasts based on Affine kernel dressing and Non-dominated sorting genetic algorithm II. The mechanisms of these two statistical post-processing methods are completely different. However, they share one similarity from another perspective, which is they can estimate the probability density directly from the data (i.e., ensemble forecast) without assuming any particular underlying distribution. As a more conventional method, Silverman (1986) firstly proposed the kernel density smoothing method to estimate the distribution from the data by centering a kernel function K that determines the shape of a probability distribution (i.e., kernel) fitted around every data point (i.e., ensemble members). The smooth kernel estimate is then the sum of those kernels. As for the choice of bandwidth h of each dressing kernel, Silverman's rule of thumb finds an optimal bandwidth textslh by assuming that the data is normally distributed. Improvements to the original idea were soon to follow. For instance, the improved Sheather Jones (ISJ) algorithm is more suitable and robust with respect to multimodality (Wand and Jones, 1994). Roulston and Smith (2003) rely on the series of "best forecasts" (i.e., best-member dressing) to compute the kernel bandwidth h. Wang and Bishop (2005) as well as Fortin et al. (2006) further improved the best member method. The later advocated that the more extreme ensemble members are more likely to be the best member of raw under-dispersive forecasts, while the central members tend to be more "precise" for over-dispersive ensemble. They proposed the idea that different predictive weights should be set over each ensemble member, given each member's rank within the ensemble. Instead of standard dressing kernels that act on individual ensemble members, Bröcker and Smith (2008) proposed the affine kernel dressing (AKD) by assuming_**

*an affine mapping between ensemble members and observation over the entire ensemble. They approximate the distribution of the observation given the ensemble."*

*General question* 4 : Then from line 66 onwards, it is not clear why it is novel to apply NSGA-II and compare it to a kernel-based dressing method. What are the advantages of using NSGA-II? Line 70: what "different conceptualizations" are we talking about? Line 74: what do you mean by "credibility"?

*Response* :

The description of "different conceptualizations" refers to the differences in the mechanisms of these two statistical post-processing methods (i.e., kernel-based dressing method and NSGA-II). The term "credibility" means "reliability". The two techniques share one similarity from another perspective, which is they can estimate the probability density directly from the data (i.e., ensemble forecast) without assuming any particular underlying distribution. The advantages of using NSGA-II is to offer the flexibility to improve the forecast quality in harmony with the forecasting aims and the specific needs of end-users.

*"Line 89: NSGA-II opens up the opportunity of improving the forecast quality in harmony with the forecasting aims and the specific needs of end-users. Given the single-model H-EPSs studied here, the hydrologic ensemble is generated by activating two forecasting tools: the ensemble weather forecasts and the EnKF. Henceforth, enhancing the H-EPS forecasting skill by assigning different credibility to ensemble members becomes preferred than reducing the number of members. Multiple objective functions (i.e., here, verifying scores) for evaluating the forecasting performances of the H-EPS are selected to guide the optimization process. The expected output is a group of solutions, also known as Pareto fronts, that can give the trade-offs between different objectives. Other post-processing techniques, like the Non-dominated sorting genetic algorithm II (NSGA-II), are now common (e.g., Liong et al., 2001; De Vos and Rientjes, 2007; Confesor and Whittaker, 2007). Such techniques are conceptually linked to the multiobjective parameter calibration of hydrologic models using Pareto approaches. Indeed, formulating a model structure or representing the hydrologic processes using a unique global optimal parameter set proves to be very subjective. Multiple optimal parameter sets exist with satisfying behavior given the different conceptualizations, albeit not identical Beven and Binley (1992). For example, Brochero et al. (2013) utilized the Pareto fronts generated with NSGA-II for selecting the "best" ensemble from a hydrologic forecasting model with a pool of 800 streamflow predictors, in order to reduce the H-EPS complexity."*

*General question* 5 : I think the authors should completely re-write the Introduction, and think about better presenting the literature, the novel aspect of the paper and the questions the paper wants to answer (i.e., its aim). Some review of the literature presented in the "methods" section 3.2 (page 9, lines 179-204) should go to the Introduction to better explain the reader why using NSGA-II could be considered a novel aspect in this paper.

90    *Response* :

     Thank you for your suggestions. We re-wrote the ***Introduction*** section to emphasize the novel aspect of this study.

*General question* 6 : The paper investigates post-processing of ensemble forecasts based on 5 hydrological models and 5 sub-catchments in Canada. However, there is nothing in the paper that discusses the differences in performance among models and sub-catchments? What drives a better/worse performance of the post-processors used in the study? I missed some reflex-

95    ions about this issue, which would certainly increase the value of the paper. Without this reflexion, and without aggregated (averages) results, I do not understand very well the usefulness of carrying out the study over 5 models and 5 sub-catchments. What does this diversity of applications bring to the analysis?

     *Response* :

100        Thank you for you suggestions. The 5 hydrologic models exploited in this study are randomly selected from HydrOlOg-ical Prediction Laboratory (HOOPLA; Thiboult et al. (2020)) that provides a modular framework to perform calibration, simulation, and streamflow prediction using multiple hydrologic models (up to 20 models) (Perrin, 2000; Seiller et al., 2012). All of these 20 models are lumped models. We opted for only 5 models more because of a layout concern rather than computation time.

105        We would also like to emphasize that the novelty of this study is that NSGA-II could not only improves the forecast per-formance compared to conventional post-processing methods but also enhance the predictive uncertainty communication by setting multiple specific objective functions from scratch.

*General question* 7 : I found the distinction between training and validation datasets and criteria very confusing. For in-stance, we present *MCRPS* as a validation criteria (section 3.3), but it is then said it is used in calibration (line 269). It is also

110   not clear to me why we do not have a calibration for each lead time. What is the impact of using one unique lead time for calibration?

     *Response* :

     Thank you for your questions. We conduct the affine mapping between the ensemble and observation over the train-

115        ing dataset (i.e., 4-day-ahead forecast). The observation time series are used to identify the free parameter vector $\theta = [a, r_1, r_2, s_1, s_2]$, minimizing the *MCRPS* to obtain the kernel-dressed ensemble. Then we tested it on other lead times to assess the robustness of the AKD predictive model. For verifying the forecast performance and skill of the raw, AKD and NSGA-II predictive models, we used 5 more verifying scores (e.g., *KGE'*, *RMSE*, *NSE*, *spread* and *MAE* shown in Figure 8). *MCRPS* was also included for reference only.

120        The skill of flood forecasts fades away with increasing lead time. The target ensemble has a horizon that extends from day 1 to 7. The 4-day-ahead ensemble forecasts issued from each single-model H-EPSs and their corresponding observations

are chosen as a training dataset, since it locates in the middle of the forecast horizon as a compromise. Here, this specific procedure we selected is mainly for an example. We conduct the calibration on day-4 and then tested it on other lead times to assess the robustness of the predictive models. Now we know that it is quite robust, however others may try some alternatives, such as implementing the calibration/validation procedures separately for each days. For better clarification, we added more explanations about why we opted for this strategy in the ***3.4 Experimental setup*** section as well as some discussion about the robustness and potential flexibility of this application in the ***Conclusions*** section.

*"Line 308: (1) Determine the length of the training period. The target ensemble for interpretation has a horizon that extends from day 1 to 7. It is a well-known fact that the skill of hydrologic forecasts fades away with increasing lead time.* ***The 4-day-ahead ensemble forecasts issued from each single-model H-EPSs and their corresponding observations are chosen as a training dataset, since located in the middle of the forecast horizon. The validation dataset thus consists of the remaining forecasts: day 1-3 and 5-7 ahead raw forecasts issued from the associated H-EPSs. Here, this specific procedure we selected is mainly for an example. We conducted the calibration on day-4 and then tested it on other lead times to assess the robustness of the predictive models. Yet one may decide otherwise, such as implementing the calibration/validation procedures separately for each days. "***

*"Line 419: The single-model H-EPSs explored in this study account for both forcing uncertainty and initial conditions uncertainty by using the ensemble weather forecasts (ECMWF) and data assimilation (EnKF). Hydrologic post-processing with AKD and NSGA-II rely on very different assumptions and methodology. However, they both transform the raw ensembles into probability distributions. Results show that the post-processed forecasts achieve stronger predictive skill and better reliability than raw forecasts. In particular, the NSGA-II post-processed forecasts achieve the most reliable performances, since this method improves both bias and ensemble dispersion. However, over-dispersion may exist occasionally over the Baskatong catchment for NSGA-II. Kernel dressed ensemble succeed in adjusting the ensemble dispersion properly, but bias increases.* ***Note that here we calibrated the models on day 4 and then tested it on the other days to assess the robustness of the procedure. The results show that both AKD and NSGA-II predictive models could offer an efficient post-processing skill and the procedure is quite robust as well. Others may try some alternatives, such as implementing the models separately on other lead times."***

*General question* 8 : Much of the justification for the selection of the study area comes from its operational role in reservoir management. However, the post-processing application presented in the paper is based on a "non-operational" context: the parameters of the post-processor are calibrated over the entire data available (not over a split sample) for a given lead time (4 days) and validated over different lead times. Operationally, though, a forecaster would have to calibrate the post-processor over a long series of past pairs of forecasts and observations, and apply it to a different set of real-time forecast (for which the observations are not yet available). What are the implications of the method proposed for an operational service? Would the operational service be fine with a post-processing that is optimized for a 4-day lead time? Is that the lead-time that most count for the service when forecasting over these catchments? Maybe some lines of discussion would be interesting in the final

155    section of the paper.

*Response* :

Thank you for you valuable suggestions. We added more discussion in the final section to discuss robustness and potential flexibility of application and explore what potential benefits will post-processing techniques could bring to the
160    operational services.

**"Line 429: In the operational field, not only quantifying, but also communicating the predictive uncertainties in probabilistic forecasts will become an essential topic. As mentioned in the introduction, another challenge that remains is how we can bridge the communication gap between the forecasters' interpretation about probabilistic forecasts and the end-users, such as the operational hydrologists, local conservation authorities, and some other**
165    **relevant stakeholders. What factor may have the strongest impact on decision-making? The different end-users may have their unique preference and demand. For instance, the reliability and sharpness (i.e., spread) could be improved simultaneously or there could be a trade-off between them. Compared to conventional post-processing method, such as AKD, NSGA-II demonstrated its superior ability for improving the forecast performance. In parallel, the use of NSGA-II opens up the opportunities to enhance the forecast quality in line with the specific needs**
170    **of end-users, since it allows for setting multiple specific objective functions from scratch. This flexibility should be considered as a key part of facilitating the implementation of H-EPSs in real-time operational forecasting effectively."**

**2   Specific questions and remarks:**

*Specific question* 1 : lines 23-24: these sentences are not very clear to me.

175

*Response* :

Thank you. We re-wrote the ***Introduction*** for better clarification.

*Specific question* 2 : line 38: what are the three main sources mentioned?

180    *Response* :

These different sources of uncertainty are related to deficiencies in the: (1) meteorological forcing; (2) mis-specified hydrologic initial and boundary conditions; (3) inherent hydrologic model structure errors, and biased estimated parameters.

*Specific question* 3 : line 47-48: what are the implications of autocorrelation in the post-processing? Besides, aren't meteorological forecasts also auto-correlated? Why is it specifically a problem to hydrological forecasts?

*Response* :

Yes, the autocorrelation is a problem for both meteorological and hydrological forecasts. We deleted this description in the updated version of the ***Introduction***.

*Specific question* 4 : Fig. 2: I understand these are daily streamflow (it is written: mm/day) averaged over each month, and not monthly streamflows. Is that so? The caption should state the period over which the averages were obtained.

*Response* :

Yes, these are daily streamflow (mm/day) averaged over each month. We modified the caption.

[Figure]

**Figure 1.** *Hydrograph of daily streamflows (mm/day) averaged over each month during 33 years from 1985 to 2017.*

*Specific question* 5 : Table 1: I do not understand the data on reservoir area: why it is important to this paper? Furthermore, I do not understand all these physical and climatic data provided: if the results are not going to be interpreted according to the characteristics of the catchments, why are these characteristics presented in the table? In what do they influence the results?

200

*Response* :

Thank you. We modified this table to keep the useful characteristics (e.g., geographic coordinates, catchment area and mean annual streamflow) especially for this study.

**Table 1.** Hydroclimatic characteristics of five sub-catchments of the Gatineau River.

| Name | Lat. | Lon. | Catchment Area ($km^2$) | Mean annual Q ($mm$) |
|---|---|---|---|---|
| Cabonga | 47.21 | -76.59 | 2,665 | 1.35 |
| Baskatong | 47.21 | -75.95 | 13,057 | 1.49 |
| Maniwaki | 46.53 | -76.25 | 4,145 | 1.24 |
| Paugan | 46.07 | -76.13 | 2,790 | 1.29 |
| Chelsea | 45.70 | -76.01 | 1,142 | 1.27 |

205

*Specific question* 6 : Line 122: why have you chosen 5 models and why not work with the 20 models? If this is a matter of computational time, could you explain it to the reader? How long it takes to post-process one single model H-EPS?

*Response* :

210

The 5 hydrologic models exploited in this study are randomly selected from HydrOlOgical Prediction Laboratory (HOOPLA; Thiboult et al. (2020)) that provides a modular framework to perform calibration, simulation, and streamflow prediction using multiple hydrologic models (up to 20 models) (Perrin, 2000; Seiller et al., 2012). All of these 20 models are lumped models. It is more of a layout concern than a computation time concern.

*Specific question* 7 : Line 136: I am used to forecast post-processing, but not with the term "ensemble interpretation method" or "interpreted ensemble (line 157). I would be happy with more explanations here.

215

*Response* :

Thank you. The term "ensemble interpretation" is widely used in some previous researches, such as Jewson (2003), Gneiting et al. (2005) and Bröcker and Smith (2008).

220

*Specific question* 8 : Line 147: correct English

*Response* :

We corrected the misspelling.

**"Line 175: In a general form, the probability density function of $p(y; X, \theta)$ defines the interpreted ensemble (i.e., kernel dressed ensemble) given the original ensemble with free parameter vector $\theta$:"**

$Specific\ question\ 9$ : Line 169: what is this rule of thumb? Please, clarify.

$Response$ :

We added more explanation about parameter $h_S$ (Silverman, 1986) in the methodology section.

**"Line 195: Here, $h_S$ is Silverman's factor (Silverman, 1986). Technically, we can use some scores (e.g., mean square error, etc) to select the optimal bandwidth $h$ for a kernel density estimation, yet this would be difficult to estimate for general kernels. Hence, the first rule of thumb proposed by Silverman gives the optimal bandwidth $h$ which is the standard deviation of the distribution. And in this case, the kernel is also assumed to be Gaussian. The parameters $\theta = [a, r_1, r_2, s_1, s_2]$ are free parameters and usually $r_1 = 0$, $r_2 = 1$, $s_1 = 0$ and $r_2 = 1$ are rational initial selections (Bröcker and Smith, 2008). Once the optimal free parameter vector $\theta = [a, r_1, r_2, s_1, s_2]$ is obtained, the interpreted ensemble can be set to:"**

$Specific\ question\ 10$ : Equations, overall: it seems to me that not all terms are always defined, explained after the equations where they are presented. $r_1$, $r_2$, $s_1$, $s_2$, etc. $z_i$ is lower case in equation 10 but upper case in equation 11. $a$ is alpha (line 169)? Please, check the equations and the way terms are presented.

$Response$ :

Thank you. We corrected all those terms in the equations and define all related ones.

**"Line 198: The parameters $\theta = [a, r_1, r_2, s_1, s_2]$ are free parameters and usually $r_1 = 0$, $r_2 = 1$, $s_1 = 0$ and $r_2 = 1$ are rational initial selections (Bröcker and Smith, 2008)."**

$Specific\ question\ 11$ : Line 175: "Eq. (6) can be further defined", should maybe be replaced to "can be re-written"

$Response$ :

We rephrased it as: **"Line 206: can be re-written as".**

$Specific\ question\ 12$ : Line 205: $X_t$ was already defined in line 143. Please, check.

$Response$ :

Yes, it was. We have removed the description of $X_t$ in this paragraph.

$Specific\ question\ 13$ : Line 191, 192: I tend not to agree with the authors. I think "accuracy" is what is first of all searched when issuing a forecast at a given day for a short lead time such as 7 days. This is specially the case for flood events, for

255    instance. Please, explain your arguments.

*Response* :

Yes, probabilistic forecasts must be, first of all, accurate. We rephrased this paragraph as below:

***"Line 221: Similar ideas can be utilized in this study as the goal is to achieve a "good forecast". Various efficiency***
260    ***criteria are needed when we verify whether an H-EPS is competent issuing accurate and reliable forecasts. Accu-***
***racy might be the first idea that crosses our mind that indicates that there is a good match between the forecasts***
***and the observations. Since here we are focused on probabilistic streamflow forecast, the accuracy could be mea-***
***sured by computing the distances between the forecast densities with the observed ones (Wilks, 2011). Usually,***
***hydrologists could rely on the Nash-Sutcliffe efficiency criterion (NSE, Nash and Sutcliffe (1970)) for measuring***
265    ***how well forecasts can reproduce the observed time series. Transforming the time series beforehand allows spe-***
***cializing it (i.e., $NSE_{inv}$, $NSE_{sqrt}$) for specific needs (e.g., Seiller et al., 2017). NSE is dimensionless and varies***
***on the interval of $[-\infty, 1]$. A perfect model forecast output would have an NSE value that equals to one."***

***"Line 229: Meanwhile, bias, also known as systematic error, refers to the correspondence between the average***
***forecast and the average observation, which is different from accuracy. For example, systematic bias exists in the***
270    ***streamflow forecasts that are consistently too high or too low. Hence, NSE and bias are utilized here as objective***
***functions, which is to say that it is seeking to minimize the bias and maximize the NSE simultaneously. This brings***
***us a multi-objective optimization question to solve."***

*Specific question* 14 : Lines 195-196: not very clear to me. Hydrologists may rely on *NSE*, but for simulations (long time
series), not necessarily for forecasters. Please, clarify.

275

*Response* :

Thank you. Yes, the *NSE* is a conventional mean-squared-error verification measure. This measure shows great sensitiv-
ity to errors in the simulation or forecast of high stremflow events. And for the probabilistic forecasts we studied, *NSE*
turned out to be a very convenient measure for verifying the overall forecast performance. Hence, we selected *NSE* as
280    one objective function (i.e., to maximaize the *NSE* and the larger value implies a better forecast mean) here.

*Specific question* 15 : Line 200: I do not understand "elitist". Please, clarify.

*Response* :

Thank you. First of all, inserting the elitism in the multi-objective optimization algorithms is not compulsory. However,
285    it would have a strong influence if the algorithms could preserve the best individuals (i.e., elites) that were found during
the search process and then incorporated the elitism back in the evolutionary process (Groşaelin et al., 2003). The

Nondominated sorting genetic algorithm II (NSGA-II; Deb et al. (2002)) proposed this classic, fast and elitist multi-objective genetic algorithm, adopted for searching for the Pareto solution set.

$Specific\ question\ 16$ : Line 215: the concept of crowding distance was not clear to me. Please, clarify.

290

$Response$ :

Thank you. Deb et al. (2002) explained "the crowding distance" calculated in the NSGA-II algorithm as "it is a measure of how close an individual is to its neighbors. Large average crowding distance will result in better diversity in the population" and "the basic idea behind the crowing distance is finding the euclidian distance between each individual
295     in a front based on their $m$ objectives in the $m$ dimensional hyper space. The individuals in the boundary are always selected since they have infinite distance assignment."

We also clarified in the manuscript as: **"Line 248: Find the crowding distance for each individual in each front. Deb et al. (2002) pointed out the basic idea of the "crowding distance" calculated in the NSGA-II is "to find the $Euclidian Distance$ between each individual in a front based on their $m$ objectives in the $m$ dimensional hyper**
300     **space. The individuals in the boundary are always selected since they have infinite distance assignment. The large average crowding distance will result in better diversity in the population". This step ensures the diversity of the population. For example, for the first front, sort the values of the objective functions in an ascending order. The boundary solutions (i.e., maximum and minimum solutions) are then the value at infinity. The crowding distance for other individuals can be assigned as:"**

305 $Specific\ question\ 17$ : line 235: was the *MCRPS* calculated using empirical distributions or a fitted theoretical distribution? Please, clarify.

$Response$ :

Thank you. We clarified the description about the *MCRPS* score in the section 3.3.

310     **"Line 271: The overall accuracy and reliability of the probabilistic forecast can be evaluated using the Continuous ranked probability score (CRPS, Matheson and Winkler, 1976, Hersbach, Gneiting and Raftery, 2007). Hersbach (2000) decomposed the CRPS into two parts: reliability and resolution. In practice, The Mean continuous ranked probability score (MCRPS) is the average value of CRPS over the whole time series T and is calculated using empirical distributions. Besides, MCRPS is negatively oriented and the optimal MCRPS value is 0:"**

315 $Specific\ question\ 18$ : line 238: why do you need both, *MAE* and *MSE*?

$Response$ :

Thank you. We replaced the *MSE* with the of Root mean squared error (*RMSE*) here. Both *MAE* and *RMSE* could evaluates the average forecast error and have the range of $[0, +\infty]$. Besides, both of them are negatively oriented so that the smaller values imply a better forecast. These are the similarities. However, the *RMSE* score is more sensitive to large errors. In some cases that the variance corresponding to the frequency distribution in higher, the *RMSE* will experiment a larger increase while the *MAE* remains stable.

*"Line 279: As for the deterministic metrics, we adopt the Mean absolute error (MAE) and Root mean squared error (RMSE, e.g., Brochero et al., 2013) for verifying the average forecast error of the variable of interest. Both MAE and RMSE are negatively oriented and range from 0 to +∞. More accurate forecasts lead to lower MAE and RMSE. Note that the RMSE score tends to penalize the large errors more than MAE. In some cases that the variance corresponding to the frequency distribution in higher, the RMSE will have larger increase while the MAE remains stable."*

*Specific question* 19 : line 248-249: I do not understand why the Taylor diagram is mentioned here. Did you use it? How? Can you explain it?

*Response* :

Thank you. We removed this short description of the Taylor diagram.

*Specific question* 20 : Figure 3: where do we find "*w*" in the text (output of NSGA-II in the figure)?

*Response* :

Thank you. Actually, the description about the weight matrix is in the section 3.2 as: *"3) Elitism strategy is introduced in the main loop. Offspring population $Q_t$ is firstly generated from parent population $P_t$ after mutation and gene cross-over. Then the above-mentioned nondominated sorting and crowding distance assignment are conducted on the composed population $R_t$ that contain both $Q_t$ and $P_t$ with the size of $2m$. The first-rate nondominated solutions will be assigned to the new parent population $P_{t+1}$.* **"Line 263: Outputs after the whole evolutionary search are the un-repeated nondomination solutions and a weight matrix can also be extracted from the solutions."** *Specifically, in this study, the population size is set to 50, the number of objective functions equals to 2, the boundary is from 0 to 1, the mutation probability and crossing-over rate are 0.1 and 0.7, and the maximum evolution runs are 430 times."*

*Specific question* 21 : lines 287-288: it is not unexpected that forecast performance decreases with lead time. I do no understand why it is "revealed" here. Please, clarify.

*Response* :

Thank you. We rephrased this sentence as: *"Interchangeability is here assessed visually, simultaneously looking at the individual RMSE values of all 5,000 members, 7 daily forecast horizons, and 5 H-EPSs. Figure 4 displays (typical) values*

*for day 500 and Baskatong sub-catchment - a video covering the full time series is available as a supplemental material to this paper). For each H-EPS forecast horizon boxes, horizontal lines consist of 100 EnKF members and vertical lines, of 50 meteorological members.* **"Line 356: Mosaics with redder colors represent higher values of the RMSE. The decreasing predictive skill of the H-EPSs with lead time is hence shown by an increasingly red mosaic."**

355  $Specific\ question$ 22 : lines 293-294: check for the English language.

$Response$ :

Thank you. We corrected the English as: *"Figure 4 displays the hydrologic forecasts build upon the 50-member ECMWF ensemble forecasts. The basic idea behind Figure 4 (and its accompanying video) is to visually assess if the initial inter-*

360  *changeability of the weather forecasts holds for the hydrologic forecasts (i.e., horizontal lines). The interchangeability of the probabilistic data assimilation scheme is assessed in parallel (vertical lines).*

$Specific\ question$ 23 : line 324: delete "In the meanwhile,"

$Response$ :

365  Thank you. We deleted "In the meanwhile," in this paragraph.

$Specific\ question$ 24 : line 335: I understand that "error growth" is usually depicted as an increase in spread with lead time and decrease in accuracy. Why should it be maintained for a single model H-EPS if the post-processor was calibrated for 4 days of lead time only and applied to other lead times? Please, clarify.

370  $Response$ :

Thank you for your question. Here, this specific procedure we selected is to be taken as an example. We conducted the calibration on day-4 and then tested it on other lead times to assess the robustness of the predictive models. Now that we know that it is quite robust, others may try alternatives such as implementing the calibration/validation procedures separately for each days. We provided more detailed responses for the ***General question 7***.

375  $Specific\ question$ 25 : Figure 9 is not explained in the text (notably the number of lines in each graph). Also, why AKD seems to work well with M05?

$Response$ :

Thank you. We added more explanations in the text for Figure 9.

380  **"Line 385: The trained optimal free parameter vector $\theta = [\alpha, r_1, r_2, s_1, s_2]$ or weight estimates are obtained over the 4-day ahead ensemble forecasts. They are then applied to the validation data set. It comprises the 1-, 3-, 5-, and 7-day ahead raw forecasts issued from the associated H-EPSs. Figure 9 shows the reliability diagrams for raw,**

*kernel dressed, and NSGA-II forecasts for the validation data set over five individual catchment. Therefore, there are 15 lines shown in each sub diagram. Again, raw forecasts (i.e., blue lines) display a severe under-dispersion, revealing that error growth is not maintained well in a single-model H-EPS. In general, the other two statistical post-processing methods succeed in improving the forecast reliability, with the curves closer to the bisector lines. Especially, the NSGA-II (i.e., red curves) demonstrates its superior ability for maintaining the reliability with the lead time. The over-dispersion appears with most of the AKD transformed ensembles (i.e., yellow lines), especially at shorter lead times. The ensemble spread tends to a proper level as the lead time increases. Note that there is one special case that the predictive distributions of the kernel dressed ensemble are the most reliable for model M05 over almost all individual catchments."*

*Specific question* 26 : line 343-344: not clear; please, revise it.

*Response* :

Thank you. We deleted this sentence in this paragraph.

*Specific question* 27 : line 345: figure 10 shows much more than spread. Please, clarify when presenting (fully) the figure.

*Response* :

Thank you. We added more description and analysis for Figure 10.

*Specific question* 28 : Figure 10 is very difficult to read. It is not clear (BW print) which graph is AKD, which is NSGA-II. We can barely see what is inside the figure. I think it needs to be re-designed.

*Response* :

We enlarged the figure and added more analysis corresponding to Figure 10.

*Specific question* 29 : Overall, terminology could be uniformed (ex., use of AKD).

*Response* :

We uniformized the terminology throughout the whole manuscipt.

*Specific question* 30 : It is a pity that the paper does not have a discussion section. I would suggest the authors to introduce one, commenting further the results obtained, comparing post-processing performance among catchments (i.e., geographic location) and hydrological models in a summarized way. This piece of work is missing in the paper and would better justify the use of several catchments and models in the analysis.

*Response* :

415     Thank you for this suggestion. We added more discussion in the final section to highlight the novelty and potential benefits that post-processing techniques may bring to the operational services.

> *"Line 429: In the operational field, not only quantifying, but also communicating the predictive uncertainties in probabilistic forecasts will become an essential topic. As mentioned in the introduction, another challenge that remains is how we can bridge the communication gap between the forecasters' interpretation about probabilistic*
420     *forecasts and the end-users, such as the operational hydrologists, local conservation authorities, and some other relevant stakeholders. What factor may have the strongest impact on decision-making? The different end-users may have their unique preference and demand. For instance, the reliability and sharpness (i.e., spread) could be improved simultaneously or there could be a trade-off between them. Compared to conventional post-processing method, such as AKD, NSGA-II demonstrated its superior ability for improving the forecast performance. In par-*
425     *allel, the use of NSGA-II opens up the opportunities to enhance the forecast quality in line with the specific needs of end-users, since it allows for setting multiple specific objective functions from scratch. This flexibility should be considered as a key part of facilitating the implementation of H-EPSs in real-time operational forecasting effectively."*

**3   Introduction**

430     Hydrologic forecasting is crucial for flood warning and mitigation (e.g., Shim and Fontane, 2002; Cheng and Chau, 2004), water supply operation and reservoir management (e.g., Datta and Burges, 1984; Coulibaly et al., 2000; Boucher et al., 2011), navigation, and other related activities. Sufficient risk awareness, enhanced disaster preparedness in the flood mitigation measures, and strengthened early warning systems are crucial in reducing the weather-related event losses. Hydrologic models are typically driven by dynamic meteorological models in order to issue forecasts over a
435     medium range horizon of 2 to 15 days (Cloke and Pappenberger, 2009). This kind of coupled hydrometeorologic forecasting systems are admitted as effective tools to issue longer lead times. Inherent in the coupled hydrometeorologic forecasting systems, some predictive uncertainties are then inevitable given the limits of knowledge and available information (Ajami et al., 2007). In fact, those uncertainties occur all along the different steps of the hydrometeorological modeling chain (e.g., Liu and Gupta, 2007; Beven and Binley, 2014). These different sources of uncertainty are related to
440     deficiencies in the meteorological forcing, mis-specified hydrologic initial and boundary conditions, inherent hydrologic model structure errors, and biased estimated parameters (e.g., Vrugt and Robinson, 2007; Ajami et al., 2007; Salamon and Feyen, 2010; Thiboult et al., 2016). Among most cases, a single deterministic forecasts turns out to be way more insufficient.

    Many substantive theories have been proposed in order to quantify and reduce the different sources of cascading fore-
445     cast uncertainties and to add good values to flood forecasting and warning. Among them, the superiority of ensemble

forecasting systems in quantifying the propagation of predictive uncertainties (over deterministic systems) is now well established (e.g., Cloke and Pappenberger, 2009; Palmer, 2002; Seo et al., 2006; Velázquez et al., 2009; Abaza et al., 2013; Wetterhall et al., 2013; Madadgar et al., 2014). Numerous challenges have been well tackled, for example: (1) meteorological ensemble prediction systems (M-EPSs) (e.g., Palmer, 1993; Houtekamer et al., 1996; Toth and Kalnay, 1997) are refined and operated worldwide by national agencies such as the European Centre for Medium-Range Weather Forecasts (ECMWF), the National Center for Environmental Prediction (NCEP), the Meteorological Service of Canada (MSC), and more; (2) the forecast accuracy is highly improved by adopting higher resolution data collection and assimilation. Sequential data assimilation techniques, such as the particle filter (e.g., Moradkhani et al., 2012; Thirel et al., 2013) and the ensemble Kalman filter (e.g., Evensen , 1994; Reichle et al., 2002; Moradkhani et al, 2005; McMillan et al., 2013) provide an ensemble of possible re-initializations of the initial conditions, expressed in the hydrologic model as state variables, such as soil moisture, groundwater level and so on; (3) forecasting skills of the coupled hydrometeorologic forecasting systems are also improved by tracking predictive errors using the full uncertainty analysis. Multimodel schemes were proposed to increase performance and decipher structural uncertainty (e.g., Duan et al., 2007; Fisher et al., 2008; Weigel et al., 2008; Najafi et al., 2011; Velázquez et al., 2011; Marty et al., 2015; Mockler et al., 2016). Thiboult et al. (2016) compared many H-EPS, accounting for the three main sources of uncertainties located along the hydrometeorological modeling chain. They pointed out that EnKF probabilistic data assimilation provided most of the dispersion for the early forecasting horizons but failed in maintaining its effectiveness with increasing lead times. A multimodel scheme allowed sharper and more reliable ensemble predictions over a longer forecast horizon; (4) statistical hydrologic post-processing component is added in the H-EPS for rectifying biases and dispersion errors (i.e., too narrow/too large) are numerous, as reviewed by Li et al. (2017). It is noteworthy that many hydrologic variables, such as discharge, follow a skewed distribution (i.e., low probability associated to the highest streamflow values), which complicates the task. Usually, in a hydrologic ensemble prediction system (H-EPS) framework (e.g., Schaake et al., 2007; Cloke and Pappenberger, 2009; Velázquez et al., 2009; Boucher et al., 2012; Abaza et al., 2017), the post-processing procedure over the atmospheric input ensemble is often referred as pre-processing, while post-processing aims at improving the hydrologic ensemble forecasting outputs.

However, another challenge still remains: how to improve the human interpretation of probabilistic forecasts and the communication of integrated ensemble forecast products to end-users (e.g., operational hydrologists, water managers, local conservation authorities, stakeholders and other relevant decision makers). Buizza et al. (2007) emphasized that both functional and technical qualities are supposed to be assessed for evaluating the overall forecast value of a hydrometeorologic forecasts. Ramos et al. (2010) further note that the best way to communicate probabilistic forecast and interpret its usefulness should be in harmony with the goals of the forecasting system and the specific needs of end-users. She also demonstrated the main achievements from two studies obtained from a Member States workshop (Thielen et al., 2005) role-play game and another survey to explore the users' risk perception of forecasting uncertainties and how they dealt with uncertain forecasts for decision-making. The results revealed that there is still space for enhancing the fore-

480  casters' knowledge and experience on bridging the communication gap between predictive uncertainties quantification and effective decision-making.

Hence, in practice, which forecast quality impacts a given decision the most? Different end-users share their unique requirements: Crochemore et al. (2017) produced the seasonal streamflow forecasting by conditioning climatology with precipitations indices (SPI3). Forecast reliability, sharpness (i.e., the ensemble spread), overall performance and low-
485  flow event detection were verified to assess the conditioning impact. In some cases, the reliability and sharpness could be improved simultaneously while more often, there was a trade-off between them. Another IMPREX project conduct an optimization for the reservoir-based hydropower production to explore the relationship between the forecast quality and economic values. They found that an over-estimation comes along with more penalization.

The study is a contribution to probe this topic by exploring hydrological post-processing of ensemble streamflow fore-
490  casts based on Affine kernel dressing (AKD) and Non-dominated sorting genetic algorithm II (NSGA-II). The mechanisms of these two statistical post-processing methods are completely different. However, they share one similarity from another perspective, which is they can estimate the probability density directly from the data (i.e., ensemble forecast) without assuming any particular underlying distribution. As a more conventional method, Silverman (1986) firstly proposed the kernel density smoothing method to estimate the distribution from the data by centering a kernel function $K$
495  that determines the shape of a probability distribution (i.e., kernel) fitted around every data point (i.e., ensemble members). The smooth kernel estimate is then the sum of those kernels. As for the choice of bandwidth $h$ of each dressing kernel, Silverman's rule of thumb finds an optimal bandwidth $h$ by assuming that the data is normally distributed. Improvements to the original idea were soon to follow. For instance, the improved Sheather Jones (ISJ) algorithm is more suitable and robust with respect to multimodality (Wand and Jones, 1994). Roulston and Smith (2003) rely on the series
500  of "best forecasts" (i.e., best-member dressing) to compute the kernel bandwidth $h$. Wang and Bishop (2005) as well as Fortin et al. (2006) further improved the best member method. The later advocated that the more extreme ensemble members are more likely to be the best member of raw under-dispersive forecasts, while the central members tend to be more "precise" for over-dispersive ensemble. They proposed the idea that different predictive weights should be set over each ensemble member, given each member's rank within the ensemble. Instead of standard dressing kernels that
505  act on individual ensemble members, Bröcker and Smith (2008) proposed the AKD method by assuming an affine mapping between ensemble members and observation over the entire ensemble. They approximate the distribution of the observation given the ensemble.

Line 89: NSGA-II opens up the opportunity of improving the forecast quality in harmony with the forecasting aims and the specific needs of end-users. Given the single-model H-EPSs studied here, the hydrologic ensemble is generated
510  by activating two forecasting tools: the ensemble weather forecasts and the EnKF. Henceforth, enhancing the H-EPS forecasting skill by assigning different credibility to ensemble members becomes preferred than reducing the number of members. Multiple objective functions (i.e., here, verifying scores) for evaluating the forecasting performances of

[revised manuscript text omitted]

**Reply referee #2 on "Exploring hydrologic post-processing of ensemble streamflow forecasts based on Affine kernel dressing and Nondominated sorting genetic algorithm II"**

Jing Xu[1], François Anctil[1], and Marie-Amélie Boucher[2]

[1]Department of Civil and Water Engineering, Université Laval, 1065 avenue de la Médecine, Québec, Québec, Canada;
[2]Department of Civil and Building Engineering, Université de Sherbrooke, 2500 Boul. de l'Université, Sherbrooke, Québec, Canada

**Correspondence:** Jing Xu (jing.xu.1@ulaval.ca)

Dear Prof. Solomatine and reviewers:

Thank you for your review comments that we received with respect to our paper. Those valuable comments have significantly enhanced our paper. We have carefully considered and addressed the reviewers' comments and suggestions, which to significant revisions in many parts of the paper. Particularly, we rewrote the ***Introduction*** section attached at the end of this letter. Below we provide our point by point responses to each of the reviewer's comments.

**1   General questions and remarks:**

*General question* 1 : The objective(s) of the research is(are) in my view not clearly stated, nor the intended contribution to the literature. Could the authors describe these?

*Response* :

Thank you for your comments. We rewrote the ***Introduction*** to clarify our research objective. Particularly, the novelty of this paper is to emphasize practicality, not only quantifying comprehensively, but also communicating the predictive uncertainties in probabilistic forecasts effectively will become an essential topic. Compared to conventional post-processing methods, such as Affine kernel dressing (AKD), the multi objective genetic algorithm (i.e., here, NSGA-II) can open up the opportunities to improve the forecast quality in line with various unique needs of end-users.

*General question* 2 : This perhaps also makes the literature review rather general, not zooming-in to identify a gap or under-represented aspects / applications of ensemble prediction, or a particular forecast challenge in the case study catchment.

*Response* :

Thank you for your comments. We re-designed the ***Introduction*** structure and added more literature review from the operational perspective. Operational forecasters are open to ensemble forecasting methods and products for assessing

floods in a probabilistic way. The main challenges for them are how to comprehensively quantify the predictive uncertainties from different sources as well as how to use the uncertainty information for better decision-making. We rewrote the introduction to build stronger and more logical sequence of thoughts. In addition to clarifying the different sources of uncertainty in the hydrometeorological forecast chain, we explored the possibility of using NSGA-II for better fitting the end-user's specific needs.

*General question* 3 : As I understand, post-processing of the meteorological ensemble forecasts was not done. Could the authors comment in the paper on the performance of the meteorological ensemble forecasts and state the reason for not also applying meteorological post-processing?

*Response* :

Thank you. We rewrote the **Introduction** section to give a better explanation: ***"Line 48: It is noteworthy that many hydrologic variables, such as discharge, follow a skewed distribution (i.e., low probability associated to the highest streamflow values), which complicates the task. Usually, in a hydrologic ensemble prediction system (H-EPS) framework (e.g., Schaake et al., 2007; Cloke and Pappenberger, 2009; Velázquez et al., 2009; Boucher et al., 2012; Abaza et al., 2017), the post-processing procedure over the atmospheric input ensemble is often referred as pre-processing, while post-processing aims at improving the hydrologic ensemble forecasting outputs."***

*General question* 4 : Based on Figure 4, presenting one forecast, I do not understand how it can be concluded that the members generated from the meteorological eps as forcing are not fully interchangeable, which is the basis of applying weights with NSGA-II. Perhaps that the video (I am sorry I could not find it, this is probably my omission) shows this, but this is not explicitly stated in the paper.

*Response* :

Thank you. The size of the animation we mentioned in the manuscript is around 3.7G which is too large for uploading to the CO Editor Portal. Hence we prepared 12 screenshots (i.e., take a screenshot every two hundred days and there are 2,192 days in total). It could reveal the phenomenon of not fully interchangeable of the meteorological forcing members for your consideration.

*General question* 5 : I do not understand why the authors choose to calibrate the post-processors on only one forecast horizon (day-4) and validate on the other horizons (1-3, 5-7). Because of a generally present decrease of skill with increasing forecast horizon, usually a post-processor is calibrated for each forecast horizon separately. It also seems that the analysis period for which re-forecasts have been prepared has not been split in a calibration and validation period (or a leave-one-out approach). There may be good reasons for choosing this approach, e.g. stemming from catchment or application characteristics versus limited data availability, or as a research objective, but I missed the explanation in the paper. Could the authors perhaps explain the chosen calibration/validation approach?

*Response* :

Thank you for you comments. The skill of flood forecasts fades away with increasing lead time. The target ensemble has a horizon that extends from day 1 to 7. The 4-day-ahead ensemble forecasts issued from each single-model H-EPSs and their corresponding observations are chosen as a training dataset, since it is located in the middle of the forecast horizon as a compromise. Here, this specific procedure was selected mainly for an example. We conducted the calibration on day-4 and then tested it on other lead times to assess the robustness of the predictive models. Now that we know that the procedure is quite robust, others may try alternatives such as implementing the calibration/validation procedures separately for each days. For better clarification, we added more explanations about why we opted for this strategy in the **3.4 Experimental setup** section as well as some discussion about the robustness and potential flexibility of this application in the **Conclusions** section.

*"Line 308: (1) Determine the length of the training period. The target ensemble for interpretation has a horizon that extends from day 1 to 7. It is a well-known fact that the skill of hydrologic forecasts fades away with increasing lead time.* ***The 4-day-ahead ensemble forecasts issued from each single-model H-EPSs and their corresponding observations are chosen as a training dataset, since located in the middle of the forecast horizon. The validation dataset thus consists of the remaining forecasts: day 1-3 and 5-7 ahead raw forecasts issued from the associated H-EPSs. Here, this specific procedure we selected is mainly for an example. We conduct the calibration on day-4 and then tested it on other lead times to assess the robustness of the predictive models. Yet one may decide otherwise, such as implementing the calibration/validation procedures separately for each days. "***

*"Line 419: The single-model H-EPSs explored in this study account for both forcing uncertainty and initial conditions uncertainty by using the ensemble weather forecasts (ECMWF) and data assimilation (EnKF). Hydrologic post-processing with AKD and NSGA-II rely on very different assumptions and methodology. However, they both transform the raw ensembles into probability distributions. Results show that the post-processed forecasts achieve stronger predictive skill and better reliability than raw forecasts. In particular, the NSGA-II post-processed forecasts achieve the most reliable performances, since this method improves both bias and ensemble dispersion. However, over-dispersion may exist occasionally over the Baskatong catchment for NSGA-II. Kernel dressed ensemble succeed in adjusting the ensemble dispersion properly, but bias increases.* **Note that here we calibrated the models on day 4 and then tested it on the other days to assess the robustness of the procedure. The results show that both AKD and NSGA-II predictive models could offer an efficient post-processing skill and the procedure is quite robust as well. Others may try alternatives such as implementing the models separately on other lead days."**

*General question* 6 : Lastly, I would kindly encourage the authors to expand the presentation and interpretation / discussion of the results. For example, why present the 5 sub-catchments, what should we learn from the results? What about the inflows to the reservoirs? Why present the 5 single h-eps, what should we learn from the results? Why not assess the performance of the combined grand multi-model ensemble? How does the performance of the raw and post-processed forecasts compare with

the performance of a reference forecast such as climatology or persistence (forecast skill)?

*Response* :

> Thank you for your kind suggestions. Actually, the 5 hydrologic models exploited in this study are randomly selected from HydrOlOgical Prediction Laboratory (HOOPLA; Thiboult et al. (2020)) that provides a modular framework to perform calibration, simulation, and streamflow prediction using multiple hydrologic models (up to 20 models) (Perrin, 2000; Seiller et al., 2012). All of these 20 models are lumped models. So, it is more of a layout concern. We also added further discussion about the novelty and potential benefits that post-processing techniques may bring to the operational services.

> ***"Line 429: In the operational field, not only quantifying, but also communicating the predictive uncertainties in probabilistic forecasts will become an essential topic. As mentioned in the introduction, another challenge that remains is how we can bridge the communication gap between the forecasters' interpretation about probabilistic forecasts and the end-users, such as the operational hydrologists, local conservation authorities, and some other relevant stakeholders. What factor may have the strongest impact on decision-making? The different end-users may have their unique preference and demand. For instance, the reliability and sharpness (i.e., spread) could be improved simultaneously or there could be a trade-off between them. Compared to conventional post-processing method, such as AKD, NSGA-II demonstrated its superior ability for improving the forecast performance. In parallel, the use of NSGA-II opens up the opportunities to enhance the forecast quality in line with the specific needs of end-users, since it allows for setting multiple specific objective functions from scratch. This flexibility should be considered as a key part of facilitating the implementation of H-EPSs in real-time operational forecasting effectively."***

> Besides, we have another article mainly focus on exploring "the hydrological post-processing of streamflow forecasts issued from multimodel ensemble prediction systems" published already. Please check: $Xu, J., Anctil, F. and Boucher, M.A., 2019$ $Hydrological\ post-processing\ of\ streamflow\ forecasts\ issued\ from\ multimodel\ ensemble\ prediction$ $systems, J. Hydrol., 578, p.124002, https://doi.org/10.1016/j.jhydrol. 2019. 124002.$ if this may interest you.

**2   Detailed comments:**

*Detailed comments* 1 : Introduction: Could you add explanation why AKD and NSGA-II have been chosen for this research? (line 51)

*Response* :

> Thank you for your comments. We rewrote the ***Introduction*** section and further explained the reason why we selected these two post-processing techniques.

*"Line 71: The study is a contribution to probe this topic by exploring hydrological post-processing of ensemble streamflow forecasts based on Affine kernel dressing (AKD) and Non-dominated sorting genetic algorithm II (NSGA-II). The mechanisms of these two statistical post-processing methods are completely different. However, they share one similarity from another perspective, which is they can estimate the probability density directly from the data (i.e., ensemble forecast) without assuming any particular underlying distribution. As a more conventional method, Silverman (1986) firstly proposed the kernel density smoothing method to estimate the distribution from the data by centering a kernel function K that determines the shape of a probability distribution (i.e., kernel) fitted around every data point (i.e., ensemble members). The smooth kernel estimate is then the sum of those kernels. As for the choice of bandwidth h of each dressing kernel, Silverman's rule of thumb finds an optimal bandwidth h by assuming that the data is normally distributed. Improvements to the original idea were soon to follow. For instance, the improved Sheather Jones (ISJ) algorithm is more suitable and robust with respect to multimodality (Wand and Jones, 1994). Roulston and Smith (2003) rely on the series of "best forecasts" (i.e., best-member dressing) to compute the kernel bandwidth h. Wang and Bishop (2005) as well as Fortin et al. (2006) further improved the best member method. The later advocated that the more extreme ensemble members are more likely to be the best member of raw under-dispersive forecasts, while the central members tend to be more "precise" for over-dispersive ensemble. They proposed the idea that different predictive weights should be set over each ensemble member, given each member's rank within the ensemble. Instead of standard dressing kernels that act on individual ensemble members, Bröcker and Smith (2008) proposed the AKD method by assuming an affine mapping between ensemble members and observation over the entire ensemble. They approximate the distribution of the observation given the ensemble."*

*"Line 89: NSGA-II opens up the opportunity of improving the forecast quality in harmony with the forecasting aims and the specific needs of end-users. Given the single-model H-EPSs studied here, the hydrologic ensemble is generated by activating two forecasting tools: the ensemble weather forecast and the EnKF. Henceforth, enhancing the H-EPS forecasting skill by assigning different credibility to ensemble members becomes preferred than reducing the number of members. Multiple objective functions (i.e., here, verifying scores) for evaluating the forecasting performances of the H-EPS are selected to guide the optimization process. The expected output is a group of solutions, also known as Pareto fronts, that can give the trade-offs between different objectives. Other post-processing techniques, like the Non-dominated sorting genetic algorithm II (NSGA-II), are now common (e.g., Liong et al., 2001; De Vos and Rientjes, 2007; Confesor and Whittaker, 2007). Such techniques are conceptually linked to the multiobjective parameter calibration of hydrologic models using Pareto approaches. Indeed, formulating a model structure or representing the hydrologic processes using a unique global optimal parameter set proves to be very subjective. Multiple optimal parameter sets exist with satisfying behavior given the different conceptualizations, albeit not identical Beven and Binley (1992). For example, Brochero et al. (2013) utilized the*

 *Pareto fronts generated with NSGA-II for selecting the "best" ensemble from a hydrologic forecasting model with a pool of 800 streamflow predictors, in order to reduce the H-EPS complexity."*

*Detailed comments* 2 : Introduction, data description, and / or Results section: Could you comment on observational uncertainty?

*Response* :

Thank you. Observational uncertainty was indirectly taken into account in the data assimilation phase of the forecasts generation as the Ensemble Kalman Filter (EnKF) asks for the perturbation of the precipitation and temperature inputs and of the streamflow observations. More specifically, streamflow uncertainties are assumed to be proportional to the value, and are expressed in terms of percentage of the observed amounts. In practice, a hyperparameter of 10% was applied to the streamflow, following the detailed analysis of Thiboult and Anctil (2015) using the same HEPS over watersheds in the vicinity of the one studied here. As this issue does not interference with the post-processing process, we feel that there is no need to discuss it in length.

Nonetheless we added the following description at the end of section 2: *Line 157: The EnKF hyperparameters selection follows the work of Thiboult and Anctil (2015). Streamflow and precipitation uncertainties are assumed proportional; they are set to 10% and 50%, respectively. Temperature uncertainty is considered constant; it amounts to 2°C. A Gaussian describes the streamflow and temperature uncertainty and a gamma law represents the precipitation uncertainty.*

*Detailed comments* 3 : Line 92: Does the analysis of forecast performance take into account these different flood generating processes, and related seasonality? Would be interesting.

*Response* :

Thank you for your interesting questions. Actually, the seasonality diversity analysis was not the main focus of this research. Here, we would really like to focus on and emphasize that in the operational filed, not only quantifying correctly, but also communicating the predictive uncertainties in probabilistic forecasts in a proper way matters. And compared with conventional statistical post-processor, NSGA-II demonstrates its superior ability for improving the forecast performance as well as provides much flexibility to enhance the forecast quality in line with the specific needs of end-users.

*Detailed comments* 4 : Line 99: In this section, kindly add some information on catchment response time to rainfall/snow melt, and travel time (routing), to inform us about potential forecast lead times without meteorological forecasts as forcing.

*Response* :

Unfortunately, we do not hold that information. We understand that they would be helpful.

*Detailed comments* 5 : Line 104: Could you mention why inflow to the reservoirs is not measured (for some reservoirs), and how the inflow time series have been constructed?

*Response* :

Thank you. The hydroclimatic times series were provided by Hydro-Québec that operates the Gatineau River system for hydroelectricity production. They constitute the operational database of this public company that have been operating their own hydrological prediction system for many decades.

Climatic time series were created by applying *kriging* to the observations collected in the study area, while the streamflows are reconstructed from the measurements of the water level in the reservoirs as well as some standard streamflow observations whenever available. This methodology is of course imperfect, especially for quite large reservoirs.

We added the following text to the manuscript: **Line 126: All hydroclimatic time series to the project were made available by Hydro-Québec that carefully constructed them for their own hydropower operations.**

*Detailed comments* 6 : Line 105/106: Could you briefly describe the observational network, and methods used to create sub-basin average precipitation? (to inform observational uncertainty, and perhaps a reason for not going for meteorological forecast post-processing).

*Response* :

Thank you. We modified this table to keep the useful characteristics (e.g., geographic coordinates, catchment area and mean annual streamflow) especially for this study.

**Table 1.** Hydroclimatic characteristics of five sub-catchments of the Gatineau River.

| Name | Lat. | Lon. | Catchment Area $(km^2)$ | Mean annual Q $(mm)$ |
|---|---|---|---|---|
| Cabonga | 47.21 | -76.59 | 2,665 | 1.35 |
| Baskatong | 47.21 | -75.95 | 13,057 | 1.49 |
| Maniwaki | 46.53 | -76.25 | 4,145 | 1.24 |
| Paugan | 46.07 | -76.13 | 2,790 | 1.29 |
| Chelsea | 45.70 | -76.01 | 1,142 | 1.27 |

*Detailed comments* 7 : Line 168: I think we are missing here, what the parameters are optimised on. From the later paragraph on Experimental set-up it seems that parameterization of *AKD* was done by minimising *MCRPS*.

*Response* :

Yes, the parameterization of *AKD* was done by minimizing *MCRPS*. We added further clarifications about these relevant parameters in Section 3.1.

*"Line 195: Here, $h_S$ is Silverman's factor (Silverman, 1986). Technically, we can use some scores (e.g., mean square error, etc) to select the optimal bandwidth $h$ for a kernel density estimation, yet this would be difficult to estimate for general kernels. Hence, the first rule of thumb proposed by Silverman gives the optimal bandwidth $h$ which is the standard deviation of the distribution. And in this case, the kernel is also assumed to be Gaussian. The parameters $\theta = [a, r_1, r_2, s_1, s_2]$ are free parameters and usually $r_1 = 0$, $r_2 = 1$, $s_1 = 0$ and $r_2 = 1$ are rational initial selections (Bröcker and Smith, 2008). Once the optimal free parameter vector $\theta = [a, r_1, r_2, s_1, s_2]$ is obtained, the interpreted ensemble can be set to:"*

*"Line 315: (2) AKD mapping between the ensemble and observation over the training dataset. The observation time series are used to identify the free parameter vector $\theta = [a, r_1, r_2, s_1, s_2]$, minimizing the MCRPS to obtain the kernel-dressed ensemble. Note that AKD acts on the entire ensemble rather than on each individual member."*

*Detailed comments* 8 : Lines 224-226: Kindly explain why these parameter values were chosen, and if a sensitivity analysis was done? Was the maximum evolution runs a result of a stopping criterion? If so, please mention this.

*Response* :

Thank you for your comments. We added more description about NSGA-II method in Section 3.2. The objective functions we selected here is *NSE* and *bias* as an example, one may try alternatives. The population size of 50, the mutation probability of 0.1, crossing-over rate of 0.7 the maximum runs of 430 were obtained through several tests. From our own experience, it is better to start testing the population size, mutation probability and number of generations with some lower values (e.g., 50, 100 for population size, and 0.1, 0.2 for mutation probability, etc.). While the opposite is recommended for the cross-over rate.

*Detailed comments* 9 : Lines 272-273: Please introduce the use of a moving window in Section 3.2, and expand explanation. Also the mentioning here of operational requirements is interesting and further explanation and discussion would be welcome.

*Response* :

We rephrased the explanation about why we chose using "a 30-day moving window" in the procedure from an operational perspective.

*"Line 315: (3) Evaluate the Pareto fronts (i.e., nondominated solutions that minimize/maximize the bias and the NSE) and the weight matrix, applying NSGA-II over the training dataset. Sloughter et al. (2007) mentioned that the training period should be specific for each dataset or region. Here a 30-day moving window is selected so it contains enough training samples with coherent consistency. Especially, from the operational perspective, a monthly moving window is more coherent and efficient in the real world, with limited length for time series."*

*Detailed comments* 10 : Figure 5: The differences in *bias* and *NSE* over the range of the Pareto front are small. Please discuss. What weights are in the weight matrices of these solutions?

*Response* :

We added some analysis about the weights obtained from the Pareto solutions.

250 **"Line 355: The NSGA-II Pareto front drawn in Figure 5 (model M01 over the Baskatong catchment) is quite typical. In this multiobjective evolutionary search, 35 (nondominated) Pareto solutions are identified. No objective can be improved more without the sacrifice of another. The optimal NSE is inevitably accompanied with the highest bias (e.g., NES = 0.84594, bias = 0.034055), or vice versa. The solutions in the elbow region of the Pareto front are the compromise between both two objective functions. Pareto fronts with different numbers of solutions**

255 **can be attained daily via setting the sliding window. Therefore, rather than choosing only one fixed position in the front, we opted to pick the solution randomly for respecting and exploring the diversity within. Figure 6 confirms NSGA-II convergence."**

*Detailed comments* 11 : Line 338: This is interesting. Could you discuss what could be the reason? Something specific about Model M05?

260

*Response* :

Thank you. The 5 lumped models exploited in this study are randomly selected from HydrOlOgical Prediction Laboratory (HOOPLA; Thiboult et al. (2020)) which means they are quite equally competitive. We would like to treat it as one special case: **Line 390: In general, the other two statistical post-processing methods succeed in improving the fore-**

265 **cast reliability, with the curves closer to the bisector lines. Especially, the NSGA-II (i.e., red curves) demonstrates its superior ability for maintaining the reliability with the lead time. The over-dispersion appears with most of the AKD transformed ensembles (i.e., yellow lines), especially at shorter lead times. The ensemble spread tends to a proper level as the lead time increases. Note that there is one special case that the predictive distributions of the kernel dressed ensemble are the most reliable for model M05 over almost all individual catchments.**

270 **3 Editorials:**

*Editorials* 1 : Lines 25-27. Deterministic systems do not asses/quantify uncertainty, so the superiority question, I think, did not concern uncertainty quantification, that difference is simply a given. The superiority question concerned more the value when using the forecasts in decision making, and ensemble mean versus deterministic forecast performance.

275 *Response* :

Thank you for this comment. Yes, the main challenge for the operational forecasters are how to comprehensively quantify the predictive uncertainties from different sources as well as how to use the uncertainty information for better decision-making. The essential topic here is to bridge the gap between the "theory" (i.e., accuracy, reliability, etc.) and the "practice" (i.e., decision-driven trade-offs). We would like to emphasize this point in this paper.

*Editorials* 2 : Figure 1: Please indicate in the map more clearly the main river reach and flow direction.

*Response* :

Thank you. The map in Figure 1 was refined.

*Editorials* 3 : Lines 82-84: Not clear from this sentence if in Section 4 the results are analysed for each model individually first (not taking into account model structure uncertainty), and then are considered and processed as a grand multi-model ensemble, which does take into account model structure uncertainty.

*Response* :

Thank you. We have another article focused on exploring "the hydrological post-processing of streamflow forecasts issued from multimodel ensemble prediction systems" published: $Xu, J., Anctil, F. and Boucher, M.A., 2019$ $Hydrological\ post-processing\ of\ streamflow\ forecasts\ issued\ from\ multimodel\ ensemble\ prediction\ systems,$ $J.\ Hydrol., 578, p.124002, https://doi.org/10.1016/j.jhydrol.$ 2019. 124002. In this above-mentioned paper, we took all sources of uncertainties into account since we tested on the grand multi-model ensemble forecast. For the manuscript discussed here, we mainly focused on the single-model H-EPSs and that is the reason why we analysed each model individually.

*Editorials* 4 : Lines 137-139: Consider to move up to Introduction for literature review, or down in the sections below. In these few introductory sentences to the methodology I would focus on announcing what was the general approach followed to reach the research objectives. After having introduced the overall methodology, going into the details of the two post-processing methods as of section 3.1 makes sense.

*Response* :

Thank you. We moved some description down to section 3.1.

*Editorials* 5 : Figure 3: Qobs is not output, so can be left out on the right. It is also not indicated that the output or final results concerns post-processed (interpreted) Qfcsts. The flowchart ending in only one set of post-processed forecasts is confusing, because up to now I was under the impression that AKD and NSGA-II would be used independently post-process and hence each method to result in a set of post-processed forecasts, after which the performance of each method will be analysed and compared.

310 Thank you for your comments about the flowchart. We refined this flowchart.

[Figure]

**Figure 1.** Schematic of the experimental setup flowchart.

*Editorials* 6 : Line 256: Spread Skill plots are announced, but later not presented.

315

*Response* :

Thank you. The Spread Skill plot ($SSP$) were referd as "*spread*" in the corresponding results (i.e., Figure 8 and 10). We rephrased the text to make it easier to track.

*Editorials* 7 : Figure 8: Presenting results in spider plots is a nice idea, but with the scores selected this does not work well,

320 because some scores indicate a better performance with lower value (*RMSE*) while others the other way around (*NSE*), and some have a scale only to 1 (*NSE*) while others are not limited. This makes interpretation of the plots rather difficult.

*Response* :

Thank you for your comment. Yes, we chose this kind of radar plots for a layout concern. We enlarged the figure size .

325 *Editorials* 8 : Lines 357-362: General/Literature - I suggest to delete or move to Introduction.

*Response* :

Thank you. We deleted this paragraph.

*Editorials* 9 : Lines 366-369: Consider to move to Introduction or Methodology.

330

*Response* :

We rephrased the final section.

**4 Introduction**

Hydrologic forecasting is crucial for flood warning and mitigation (e.g., Shim and Fontane, 2002; Cheng and Chau,
335 2004), water supply operation and reservoir management (e.g., Datta and Burges, 1984; Coulibaly et al., 2000; Boucher
et al., 2011), navigation, and other related activities. Sufficient risk awareness, enhanced disaster preparedness in the
flood mitigation measures, and strengthened early warning systems are crucial in reducing the weather-related event
losses. Hydrologic models are typically driven by dynamic meteorological models in order to issue forecasts over a
medium range horizon of 2 to 15 days (Cloke and Pappenberger, 2009). This kind of coupled hydrometeorologic fore-
340 casting systems are admitted as effective tools to issue longer lead times. Inherent in the coupled hydrometeorologic
forecasting systems, some predictive uncertainties are then inevitable given the limits of knowledge and available infor-
mation (Ajami et al., 2007). In fact, those uncertainties occur all along the different steps of the hydrometeorological
modeling chain (e.g., Liu and Gupta, 2007; Beven and Binley, 2014). These different sources of uncertainty are related to
deficiencies in the meteorological forcing, mis-specified hydrologic initial and boundary conditions, inherent hydrologic
345 model structure errors, and biased estimated parameters (e.g., Vrugt and Robinson, 2007; Ajami et al., 2007; Salamon
and Feyen, 2010; Thiboult et al., 2016). Among most cases, a single deterministic forecasts turns out to be way more
insufficient.

Many substantive theories have been proposed in order to quantify and reduce the different sources of cascading fore-
cast uncertainties and to add good values to flood forecasting and warning. Among them, the superiority of ensemble
350 forecasting systems in quantifying the propagation of predictive uncertainties (over deterministic systems) is now well
established (e.g., Cloke and Pappenberger, 2009; Palmer, 2002; Seo et al., 2006; Velázquez et al., 2009; Abaza et al.,
2013; Wetterhall et al., 2013; Madadgar et al., 2014). Numerous challenges have been well tackled, for example: (1)
meteorological ensemble prediction systems (M-EPSs) (e.g., Palmer, 1993; Houtekamer et al., 1996; Toth and Kalnay,
1997) are refined and operated worldwide by national agencies such as the European Centre for Medium-Range Weather
355 Forecasts (ECMWF), the National Center for Environmental Prediction (NCEP), the Meteorological Service of Canada
(MSC), and more; (2) the forecast accuracy is highly improved by adopting higher resolution data collection and assim-
ilation. Sequential data assimilation techniques, such as the particle filter (e.g., Moradkhani et al., 2012; Thirel et al.,
2013) and the ensemble Kalman filter (e.g., Evensen , 1994; Reichle et al., 2002; Moradkhani et al, 2005; McMillan et
al., 2013) provide an ensemble of possible re-initializations of the initial conditions, expressed in the hydrologic model

as state variables, such as soil moisture, groundwater level and so on; (3) forecasting skills of the coupled hydrometeorologic forecasting systems are also improved by tracking predictive errors using the full uncertainty analysis. Multimodel schemes were proposed to increase performance and decipher structural uncertainty (e.g., Duan et al., 2007; Fisher et al., 2008; Weigel et al., 2008; Najafi et al., 2011; Velázquez et al., 2011; Marty et al., 2015; Mockler et al., 2016). Thiboult et al. (2016) compared many H-EPS, accounting for the three main sources of uncertainties located along the hydrometeorological modeling chain. They pointed out that EnKF probabilistic data assimilation provided most of the dispersion for the early forecasting horizons but failed in maintaining its effectiveness with increasing lead times. A multimodel scheme allowed sharper and more reliable ensemble predictions over a longer forecast horizon; (4) statistical hydrologic post-processing component is added in the H-EPS for rectifying biases and dispersion errors (i.e., too narrow/too large) are numerous, as reviewed by Li et al. (2017). It is noteworthy that many hydrologic variables, such as discharge, follow a skewed distribution (i.e., low probability associated to the highest streamflow values), which complicates the task. Usually, in a hydrologic ensemble prediction system (H-EPS) framework (e.g., Schaake et al., 2007; Cloke and Pappenberger, 2009; Velázquez et al., 2009; Boucher et al., 2012; Abaza et al., 2017), the post-processing procedure over the atmospheric input ensemble is often referred as pre-processing, while post-processing aims at improving the hydrologic ensemble forecasting outputs.

However, another challenge still remains: how to improve the human interpretation of probabilistic forecasts and the communication of integrated ensemble forecast products to end-users (e.g., operational hydrologists, water managers, local conservation authorities, stakeholders and other relevant decision makers). Buizza et al. (2007) emphasized that both functional and technical qualities are supposed to be assessed for evaluating the overall forecast value of a hydrometeorologic forecasts. Ramos et al. (2010) further note that the best way to communicate probabilistic forecast and interpret its usefulness should be in harmony with the goals of the forecasting system and the specific needs of end-users. She also demonstrated the main achievements from two studies obtained from a Member States workshop (Thielen et al., 2005) role-play game and another survey to explore the users' risk perception of forecasting uncertainties and how they dealt with uncertain forecasts for decision-making. The results revealed that there is still space for enhancing the forecasters' knowledge and experience on bridging the communication gap between predictive uncertainties quantification and effective decision-making.

Hence, in practice, which forecast quality impacts a given decision the most? Different end-users share their unique requirements: Crochemore et al. (2017) produced the seasonal streamflow forecasting by conditioning climatology with precipitations indices (SPI3). Forecast reliability, sharpness (i.e., the ensemble spread), overall performance and low-flow event detection were verified to assess the conditioning impact. In some cases, the reliability and sharpness could be improved simultaneously while more often, there was a trade-off between them. Another IMPREX project conduct an optimization for the reservoir-based hydropower production to explore the relationship between the forecast quality and economic values. They found that an over-estimation comes along with more penalization.

The study is a contribution to probe this topic by exploring hydrological post-processing of ensemble streamflow forecasts based on Affine kernel dressing (AKD) and Non-dominated sorting genetic algorithm II (NSGA-II). The mechanisms of these two statistical post-processing methods are completely different. However, they share one similarity from another perspective, which is they can estimate the probability density directly from the data (i.e., ensemble forecast) without assuming any particular underlying distribution. As a more conventional method, Silverman (1986) firstly proposed the kernel density smoothing method to estimate the distribution from the data by centering a kernel function $K$ that determines the shape of a probability distribution (i.e., kernel) fitted around every data point (i.e., ensemble members). The smooth kernel estimate is then the sum of those kernels. As for the choice of bandwidth $h$ of each dressing kernel, Silverman's rule of thumb finds an optimal bandwidth $h$ by assuming that the data is normally distributed. Improvements to the original idea were soon to follow. For instance, the improved Sheather Jones (ISJ) algorithm is more suitable and robust with respect to multimodality (Wand and Jones, 1994). Roulston and Smith (2003) rely on the series of "best forecasts" (i.e., best-member dressing) to compute the kernel bandwidth $h$. Wang and Bishop (2005) as well as Fortin et al. (2006) further improved the best member method. The later advocated that the more extreme ensemble members are more likely to be the best member of raw under-dispersive forecasts, while the central members tend to be more "precise" for over-dispersive ensemble. They proposed the idea that different predictive weights should be set over each ensemble member, given each member's rank within the ensemble. Instead of standard dressing kernels that act on individual ensemble members, Bröcker and Smith (2008) proposed the AKD method by assuming an affine mapping between ensemble members and observation over the entire ensemble. They approximate the distribution of the observation given the ensemble.

Line 89: NSGA-II opens up the opportunity of improving the forecast quality in harmony with the forecasting aims and the specific needs of end-users. Given the single-model H-EPSs studied here, the hydrologic ensemble is generated by activating two forecasting tools: the ensemble weather forecasts and the EnKF. Henceforth, enhancing the H-EPS forecasting skill by assigning different credibility to ensemble members becomes preferred than reducing the number of members. Multiple objective functions (i.e., here, verifying scores) for evaluating the forecasting performances of the H-EPS are selected to guide the optimization process. The expected output is a group of solutions, also known as Pareto fronts, that can give the trade-offs between different objectives. Other post-processing techniques, like the Non-dominated sorting genetic algorithm II (NSGA-II), are now common (e.g., Liong et al., 2001; De Vos and Rientjes, 2007; Confesor and Whittaker, 2007). Such techniques are conceptually linked to the multiobjective parameter calibration of hydrologic models using Pareto approaches. Indeed, formulating a model structure or representing the hydrologic processes using a unique global optimal parameter set proves to be very subjective. Multiple optimal parameter sets exist with satisfying behavior given the different conceptualizations, albeit not identical Beven and Binley (1992). For example, Brochero et al. (2013) utilized the Pareto fronts generated with NSGA-II for selecting the "best" ensemble from a hydrologic forecasting model with a pool of 800 streamflow predictors, in order to reduce the H-EPS complexity.

430

In this study, the daily streamflow ensemble forecasts issued from five single-model H-EPSs over the Gatineau River (Province of Québec, Canada) are post-processed. Details about the study area, hydrologic models, and hydrometeorologic data are described in Section 2. Section 3 explains the methodology and training strategy of AKD and NSGA-II methods, in parallel with the scoring rules that evaluate the performance of the forecasts. Specific concepts associated with those scores are also introduced in this section. Predictive distribution estimation based on the five single-model H-EPSs configurations, which lack accounting for the model structure uncertainty, is presented in Section 4. The comparison of both statistical post-processing methods in improving the forecasting quality as well as enhancing the uncertainty communication are discussed and analyzed as well. Conclusion follows in Section 5.

[revised manuscript text omitted]

---

## Author Response (AR2)

**Reply referee #1 on "Exploring hydrologic post-processing of ensemble streamflow forecasts based on Affine kernel dressing and Nondominated sorting genetic algorithm II"**

Jing Xu[1], François Anctil[1], and Marie-Amélie Boucher[2]

[1]Department of Civil and Water Engineering, Université Laval, 1065 avenue de la Médecine, Québec, Québec, Canada;
[2]Department of Civil and Building Engineering, Université de Sherbrooke, 2500 Boul. de l'Université, Sherbrooke, Québec, Canada

**Correspondence:** Jing Xu (jing.xu.1@ulaval.ca)

Dear Prof. Solomatine and reviewers:

Thank you for your review comments that we received with respect to our paper. Those valuable comments have significantly enhanced our paper. We have carefully considered and addressed the reviewers' comments and suggestions, which led to significant revisions in many parts of the manuscript. Below we provide a point by point responses to each of the reviewer's

5    comments. Please note that all the line numbers mentioned in our responses refer to the track-changes version of the manuscript.

**1 Summary:**

This study aims at improving the quality of probabilistic forecasts and facilitating the uncertainty communication by using two post-processing approaches within a hydrologic ensemble prediction system framework. The methods are clearly described and I find the results convincing. I think the article is ready for publication after some minor corrections.

**2 Minor comments:**

$Specific\ question\ 1$: L53: It might be useful to give a short description of what "post-processing" means, something like "a correction of predictive distributions based on a comparison with past observations".

15    $Response$:

Thank you for your suggestion. We added a short description about "post-processing" here for better clarification.

*"Line 54: By correcting the bias and adjusting the dispersion based on the comparison with past observations, statistical post-processing generally leads to a more accurate and reliable hydrologic ensemble forecast."*

$Specific\ question\ 2$: L68: A reference to "Another IMPREX project" is needed.

20

*Response* :

Thank you. We left this sentence out as the other reviewer suggested.

If you may be interested, the reference for this project is: $Cassagnole, M., Ramos, M.H., Zalachori, I, Thirel, G.,$ $Garçon, R., Gailhard, J., and Ouillon, T. : Impact \, of \, the \, quality \, of \, hydrological \, forecasts \, on \, the \, management$ $and \, revenue \, of \, hydroelectric \, reservoirs-a \, conceptual \, approach, Hydrol. Earth Syst. Sci., 25(2), 1033-1052,$ $https : //doi.org/10.5194/hess-25-1033-2021 \, 2021.$

*Specific question* 3 : L120: It might be useful to add the location of the gauging stations.

*Response* :

Thank you. The daily streamflow $(m^3/s)$ time series entering the reservoirs were constructed by the electricity producer using a water balance equation for reservoirs and the turbine flow for run-of-the-river dams (identified as red thunder marks in Figure 1) and made available to the study along with spatially averaged minimum and maximum air temperature $(°C)$ and precipitation $(mm)$ for each sub-basin.

*Specific question* 4 : L148: "Five random hydrologic models from HOOPLA are exploited in this study.": what do you mean by "random" ? I find it very surprising that no other reason explains the choice of using those models.

*Response* :

Thank you for your question. For simplicity, we picked 5 models that are representatives among the 20 lumped models that were available to the study. The 5 representative models exploited here were selected from HydrOlOgical Prediction Laboratory (HOOPLA; Thiboult et al. (2020)) as typical examples. HOOPLA was able to provide a modular framework to perform calibration, simulation, and streamflow prediction using multiple hydrologic models (up to 20 models) (Perrin, 2000; Seiller et al., 2012). We rephrased the relative description in the manuscript for better clarification.

*"Line 133: The HydrOlOgical Prediction Laboratory (HOOPLA; Thiboult et al. (2020)) provides a modular framework to perform calibration, simulation, and streamflow prediction using multiple hydrologic models (up to 20 lumped models) (Perrin, 2000; Seiller et al., 2012). The empirical two-parameter model CemaNeige (Valéry et al., 2014) simulates snow accumulation and melt. **In this study, five representative models were selected from HOOPLA as typical examples.** Their main characteristics are summarized in Table 2."*

*Specific question* 5 : L225: It find it difficult to understand how the *NSE* is computed: with probabilistic streamflow forecast or with "average forecast and the average observation"? In addition, what do you mean by: "observed ones"? Did you use the same uncertainty estimation for observed values as in the *EnKF* (L160)? If not, please explain why.

*Response* :

Thank you. We rephrased this sentence using only "observations". We also added one more equation to show how we calculated the $NSE$ score for better clarification. The $x_t$ and $y_t$ in the equation below represent the forecasted and observed values at time step $t$, respectively. We averaged the ensemble members for each time step.

*"Line 220: Since here we are focused on probabilistic streamflow forecast, the accuracy could be measured by computing the distances between the forecast densities with the **observations** (Wilks, 2011). Usually, hydrologists could rely on the Nash-Sutcliffe efficiency criterion (NSE, Nash and Sutcliffe (1970)) for measuring how well forecasts can reproduce the observed time series. Transforming the time series beforehand allows specializing it (i.e., $NSE_{inv}$, $NSE_{sqrt}$) for specific needs (e.g., Seiller et al., 2017). **NSE is attained by dividing the Mean square error (MSE) by the variance of the observations and then subtracting that ratio from 1.***

$$NES = 1 - \frac{MSE}{var(y)} = 1 - \frac{\sum_{t=1}^{T}(x_t - y_t)^2}{\sum_{t=1}^{T}(y_t - \bar{y})^2} \tag{1}$$

*where $x_t$ and $y_t$ are the forecasted and observed values at time step $t$, respectively. $\bar{y}$ and $var(y)$ represent the mean and variance of the observations. A perfect model forecast output would have an $NSE$ value that equals to one."*

No. The $EnKF$ uncertainty estimation is not our direct research subject. The $EnKF$ hyperparameters selection follows the work of Thiboult and Anctil (2015). Streamflow and precipitation uncertainties are assumed proportional; they are set to 10% and 50%, respectively. Temperature uncertainty is considered constant that it amounts to $2°C$. A Gaussian describes the streamflow and temperature uncertainty and a gamma law represents the precipitation uncertainty.

*Specific question* 6 : L390: I find it a bit difficult to understand the link between the post-processing settings and the impact of some probabilistic scores. In particular, reliability is clearly improved, at the cost of dispersion, but I am not sure to understand how and why the weights obtained with NSGA-II can lead to those results. I would find it useful that you discuss this point in more depth.

*Response* :

Thank you for your question. We tested the post-processing performance of evolutionary multi-objective optimization (with NSGA-II) in this study. After the whole multiobjective evolutionary search, the un-repeated nondomination Pareto solutions can be identified. The solutions in the elbow region of the Pareto front are the compromise between both two objective functions. For example, in the study, the optimal $NSE$ is inevitably accompanied with the highest $bias$ (e.g., $NES$= 0.846, $bias$=0.034), or vice versa. The solutions can be obtained daily by setting the sliding window. Specifically speaking, the NSGA-II post-processors were trained using only the past 30 days and day-4 forecast data and then re-trained every next day. Then the weight matrix can also be extracted from the solutions daily.

Hence, if $y$ is assumed here as the target variable to streamflow forecast and $Y_{obs}^t=[y^1, y^2,\ldots, y^T]$ groups the time variant observations over the training period $t = 1, \ldots, T$. Also, let $X_i^t=[x_1^t, x_2^t,\ldots,x_K^t]$ represents $K$ ensemble forecast members issued from the single-model H-EPSs daily. The weights $w_k$ reflect how well the $k^{th}$ prediction fits the training data at each time step. By assigning the weights to each candidate ensemble member, the bias and dispersion could be adjusted based on the comparison with past observations. This leads to a more reliable and skillful hydrologic ensemble forecast. For the detailed calculation steps, please refer to section 3.2 (Nondominated sorting genetic algorithm II) and section 3.4 (Experimental setup).

*Specific question* 7 : L413: While I find the study convincing, I would appreciate a bit more of discussion, especially regarding the limits of the H-EPS and the various ways to improve them. It might be surprising that a H-EPS based on weather forecasts and data assimilation (EnKF) still needs some post-processing methods to be reliable. It might indicate that more work is needed to more appropriately define the data assimilation settings and inflate the ensemble within the data assimilation step. Or that a multimodel approach is required ?

*Response* :

Thank you for your comment. Yes, a comprehensive uncertainty analysis will be needed to track all sources of uncertainties in the hydro-meteorological forecasting chain. While, in practice, the forecasting performance of data assimilation fades away quickly as the lead time progresses. In addition, operational forecasts users may not be able to perfectly utilize all the forecasting tools (i.e., meteorological ensemble forcing, data assimilation, and multimodel) jointly. For the manuscript discussed here, we would like mainly focus on the single-model H-EPSs and did the uncertainty analysis for each model individually.

As for the other strategy of multimodel approach, We actually have another article focused on exploring "the hydrological post-processing of streamflow forecasts issued from multimodel ensemble prediction systems" published: *Xu, J., Anctil, F. and Boucher, M.A.*, 2019 *Hydrological post−processing of streamflow forecasts issued from multimodel ensemble prediction systems*, *J. Hydrol.*, 578, p.124002, *https* : //doi.org/10.1016/j.jhydrol.2019. 124002. In this above-mentioned paper, we took all sources of uncertainties into account since we tested on the grand multi-model ensemble forecast.

**References**

110    Movahedinia, F.: Assessing hydro-climatic uncertainties on hydropower generation, Université Laval, Québec city, 7 pp, https://corpus.ulaval.ca/jspui/handle/20.500.11794/25294, 2014.

Nash, J.E. and Sutcliffe, I.: River flow forecasting through conceptual models. Part 1-A discussion of principles. J. Hydrol., 10(3), 282-290, https://doi.org/10.1016/0022-1694(70)90255-6, 1970.

Perrin, C.: Vers une amélioration d'un modèle global pluie-débit, PhD diss., Institut National Polytechnique de Grenoble-INPG, Grenoble,

115    287 pp, 2000.

Seiller, G., Roy, R., and Anctil, F.: Influence of three common calibration metrics on the diagnosis of climate change impacts on water resources, J. Hydrol., 547, 280-295, https://doi.org/10.1016/j.jhydrol.2017.02.004, 2017.

Seiller, G., Anctil, F., and Perrin, C.: Multimodel evaluation of twenty lumped hydrological models under contrasted climate conditions, Hydrol. Earth. Syst. Sc., 16, 1171-1189, https://doi.org/DOI : 10.5194/hess-1116-1171-2012, 2012.

120    Thiboult, A., Seiller, G., Poncelet, C., and Anctil, F.: The HOOPLA toolbox: a HydrOlOgical Prediction LAboratory to explore ensemble rainfall-runoff modeling, arXiv [preprint], Hydrol. Earth Syst. Sci. Discuss., https://doi.org/10.5194/hess-2020-6, 28 January 2020.

Thiboult, A, Anctil, F.: On the difficulty to optimally implement the Ensemble Kalman filter: An experiment based on many hydrological models and catchments, J. Hydrol., 529, 1147-1160, https://doi.org/10.1016/j.jhydrol.2015.09.036, 2015.

Valéry, A., Andréassian, V., and Perrin, C.: As simple as possible but not simpler': What is useful in a temperature-based snow-accounting

125    routine? Part 2 - Sensitivity analysis of the Cemaneige snow accounting routine on 380 catchments, J. Hydrol., 517: 1176-1187, https://doi.org/10.1016/j.jhydrol.2014.04.058, 2014.

Wilks, D.S.: On the Reliability of the Rank Histogram, Mon. Weather. Rev., 139, 311-316, https://doi.org/10.1175/2010MWR3446.1, 2011.

**Reply referee #2 on "Exploring hydrologic post-processing of ensemble streamflow forecasts based on Affine kernel dressing and Nondominated sorting genetic algorithm II"**

Jing Xu[1], François Anctil[1], and Marie-Amélie Boucher[2]

[1]Department of Civil and Water Engineering, Université Laval, 1065 avenue de la Médecine, Québec, Québec, Canada;
[2]Department of Civil and Building Engineering, Université de Sherbrooke, 2500 Boul. de l'Université, Sherbrooke, Québec, Canada

**Correspondence:** Jing Xu (jing.xu.1@ulaval.ca)

Dear Prof. Solomatine and reviewers:

Thank you for your review comments that we received with respect to our paper. Those valuable comments have significantly enhanced our paper. We have carefully considered and addressed the reviewers' comments and suggestions, which led to significant revisions in many parts of the manuscript. Below we provide our point by point responses to each of the reviewer's comments. Please note that all the line numbers mentioned in our responses refer to the track-changes version of the manuscript.

**1 General Comments:**

*General question* 1 : The body of the experiments and results is in my understanding a comparison of performance of two post-processing methods AKD and NSGA-II multi-objective optimisation. In the Introduction and Conclusion sections now a reflection has been added on user requirements of ensemble forecasts and benefits of being able to communicate trade-offs in which skill aspect(s) to increase with a hydrologic post-processor. The multi-objective optimisation has that benefit through presenting of pareto fronts and discussing these with end users. The current write-up of the Introduction and Conclusions, however, leaves the reflection on end user perspective still quite disconnected from the experimental set-up and results (and the reflection on end user perspective is not mentioned at all in the Abstract). A suggestion, for consideration, to further clarify the connection between objective and experimental set-up and results is to take as objective something like: "In this paper the performance of evolutionary multi-objective optimisation (with NSGA-II) as hydrological ensemble post-processor is tested and compared with a conventional state-of-the-art post-processor, AKD, because of the benefit of NSGA-II method in communicating trade-offs with end users on which performance aspects to improve." This is rather a long sentence and can be split-up in several sentences if that makes the message more clear. In the Conclusion sections the authors can then reflect on this objective, and objective and conclusions can be taken up in the abstract as well.

*Response* :

Thank you for your suggestions. We added more clarification in both ***Abstract*** and ***Conclusion*** sections.

*"Line 1: Forecast uncertainties are unfortunately inevitable when conducting the deterministic analysis of a dynamical system. The cascade of uncertainty originates from different components of the forecasting chain, such as the chaotic nature of the atmosphere, various initial conditions and boundaries, inappropriate conceptual hydrologic modeling, and the inconsistent stationarity assumption in a changing environment. Ensemble forecasting proves to be a powerful tool to represent error growth in the dynamical system and to capture the uncertainties associated with different sources. **In practice, the proper interpretation of the predictive uncertainties and model outputs will also have a crucial impact on risk-based decision. In this study, the performance of evolutionary multi-objective optimization (i.e., Non-dominated sorting genetic algorithm II, NSGA-II) as a hydrological ensemble post-processor was tested and compared with a conventional state-of-the-art post-processor, the Affine kernel dressing (AKD). Those two methods are theoretically/technically distinct, yet however share the same feature that both of them relax the parametric assumption of the underlying distribution of the data (the streamflow ensemble forecast). Both NSGA-II and AKD post-processors showed efficiency and effectiveness in eliminating forecast biases and maintaining a proper dispersion with increasing forecasting horizons. In addition, NSGA-II method demonstrated superiority in communicating trade-offs with end-users on which performance aspects to improve."***

*"Line 422: In this paper, the performance of NSGA-II method is compared with a conventional post-processing method, the AKD. NSGA-II demonstrated its superior ability in improving the forecast performance as well as communicating trade-offs with end-users on which performance aspects to improve most. As selected objective functions here, neither $NSE$ nor $bias$ could be improved more without negatively impacting the other. The use of NSGA-II opens up opportunities to enhance the forecast quality in line with the specific needs of end-users, since it allows for setting multiple specific objective functions from scratch. This flexibility should be considered as a key element for facilitating the implementation of H-EPSs in real-time operational forecasting."*

*General question* 2 : The choice to train the post-processors on day-4 lead time and test/validate on days 1-3 and 5-7 has been clearly described, and the option to train for each lead time separately instead has been mentioned. The main point of discussion, however, is that by the approach taken, the observational data set for training and validation is the same. Question to be discussed in the paper by the authors is whether and how a validation of the methods to a period unseen would affect the results. Unless there is a miss-understanding on my side, because I do not quite understand the 30-day moving window mentioned as training period in lines 320-321. In other sections years 1985-2017 are referred to as the focus of this study (e.g. line 132), and 2011-2016 as "committed to forecasting" (line 150). It could be that the longer time period was used in earlier studies for calibrating and validating the hydrological models used, and then perhaps the years 2011-2016 have been used to emulate re-forecasting using the trained post-processing methods, and then perhaps the training of the post-processor is done only on the past 30-days (emulating an operational setting over the period 2011-2016 and re-training every next day within that period). After which the performance metrics reported in figures 7 onwards report the performance over 2011-2016. While these elements are all stated somewhere in the paper, I would request the authors to further clarify. This can be at

the lines referred to in my comment here, and in addition perhaps by adding analysis periods to the captions of the result figures.

*Response* :

Thank you for your comments and suggestions. After the whole multiobjective evolutionary search in the NSGA-II method, the un-repeated nondomination Pareto solutions can be identified. The solutions can be obtained daily by setting the 30-day sliding window. Specifically speaking, the NSGA-II post-processors were trained using only the past 30 days and day-4 forecast data and then re-trained every next day.

In addtion, we reformulated the relevant description of the dataset we used. The analysis periods were also added to each corresponding figure captions for better clarification.

*"Line 133: The HydrOlOgical Prediction Laboratory (HOOPLA; Thiboult et al. (2020)) provides a modular framework to perform calibration, simulation, and streamflow prediction using multiple hydrologic models (up to 20 lumped models) (Perrin, 2000; Seiller et al., 2012). The empirical two-parameter model CemaNeige (Valéry et al., 2014) simulates snow accumulation and melt. In this study, five representative models were selected from HOOPLA as typical examples. Their main characteristics are summarized in Table 2.*

*The original observational time series extend from January 1950 to December 2017. While in terms of the input of HOOPLA, the observational period was limited to 33 years (1985-2017) to avoid the increased bias and variability caused by missing values within the record. The meteorological ensemble forecasts were retrieved from the European Center for Medium-Range Weather Forecasts (ECMWF; Fraley et al. (2010)). The time series extend from January 2011 to December 2016. The meteorological ensemble forecast used the reduced Gaussian transformation to the latitude-longitude system during the THORPEX Interactive Grand Global Ensemble (TIGGE) database retrieving by bilinear interpolation (e.g., Gaborit et al., 2013). The horizontal resolution was downscaled during retrieval from the $0.5°$ ECMWF grid resolution to a $0.1°$ grid resolution. This study resorts to the 12:00 UTC forecasts only, aggregated to a daily time step over a 7-day horizon. All data are aggregated at the catchment scale, averaging grid points located within each sub-catchments. All time series were split in two following the Split-Sample Test (SST) procedure of Klemeš (1986): 1986-2006 for calibration and 2013-2017 for validation. In both cases, three prior years were used for spin-up. January 2011-December 2016 is committed to hydrologic forecasting."*

*General question* 3 : If there is any overlap between training/calibration and validation period in hydrological model development and/or re-forecast analysis and/or post-processor training, the potential impact of that overlap on the results and interpretation should be discussed.

*Response* :

Thank you for your comment. No, there is no overlap between training and validation period. The dataset used here can be considered having two components: the observations/forecasts that last from January 2011 to December 2016, and

the target ensemble for interpretation with a forecasting horizon that extends from day 1 to 7. In this study, a common calibration/validation procedure was conducted on the second component of the dataset. We added more clarification in section 3.4 Experimental setup as:

*"Line 308: (1) Determine the training period. Subject to the dataset used in this study, it can be considered having two components: the observations/forecasts that last from January 2011 to December 2016, and the target ensemble for interpretation with a forecasting horizon that extends from day 1 to 7. Here, a common calibration/validation procedure was conducted on the second component of the dataset. We conducted the calibration on day-4 forecast and then tested it on other lead times to assess the robustness of the predictive models. The skill of hydrologic forecasts fades away with increasing lead time. The 4-day-ahead ensemble forecasts issued from each single-model H-EPSs and their corresponding observations are chosen as a training dataset, since located in the middle of the forecast horizon. The validation dataset then consists of the remaining forecasts: day 1-3 and 5-7 ahead raw forecasts issued from the associated H-EPSs. The procedure was selected as a specific example.* Yet one may decide otherwise, such as implementing the calibration/validation procedures separately for each day."*

**2 Detailed comments and editorials:**

*Detailed comments* 1 : l26-27: suggest to leave out colloquial sentence about insufficient single deterministic forecast.

*Response* :

Thank you for your comments. We removed this sentence.

*Detailed comments* 2 : l34: delete "national" (because ECMWF is not a national organisation)

*Response* :

Thank you. We deleted it.

*Detailed comments* 3 : l46-47: Edit, sentence is not correct.

*Response* :

Thank you. We rephrased the sentence.

*"Line 48: the statistical hydrologic post-processors, which have been added in the H-EPS for rectifying biases and dispersion errors (i.e., too narrow/too large), are numerous as reviewed by Li et al. (2017)."*

*Detailed comments* 4 : l57: Edit: "value of a hydrological forecasts"

120 *Response* :

Thank you. We edited this sentence.

*"Line 59: Buizza et al. (2007) emphasized that both functional and technical qualities are supposed to be assessed for evaluating the overall forecast **value of hydrological forecasts."***

*Detailed comments* 5 : Line 59: "She also demonstrated.." ? Colloquial. Provide specific name/reference.

125 *Response* :

Thank you. We added the reference.

***"Line 61: Ramos et al. (2010) reported similar achievements from two studies obtained from a Member States workshop (Thielen et al., 2005) role-play game and another survey to explore the users' risk perception of forecasting uncertainties and how they dealt with uncertain forecasts for decision-making."***

130 *Detailed comments* 6 : l64-69: this is an important paragraph leading-up to the objective of the paper, but it contains some unclear sentences. Please reformulate to clarify. (see suggestion in General Comment above)

*Response* :

Thank you. We reformulated this paragraph for better clarification.

135 ***"Line 71: Here, two hydrological post-processors, namely the Affine kernel dressing (AKD) and the evolutionary multi-objective optimization (Non-dominated sorting genetic algorithm II, NSGA-II), were explored. Compared to conventional post-processing method, such as AKD, NSGA-II opens up the opportunity of improving the forecast quality in harmony with the forecasting aims and the specific needs of end-users. Multiple objective functions (i.e., here, verifying scores) for evaluating the forecasting performances of the H-EPSs are selected to guide the optimiza-***
140 ***tion process."***

*Detailed comments* 7 : l68: "..another IMPREX product conduct.." ? (not a correct sentence, but also IMPREX not introduced before I think. Leave out if possible)

*Response* :

145 Thank you for your suggestion. We left this sentence out.

*Detailed comments* 8 : l71: should be "This study is a contribution.."

*Response* :

Thank you. We deleted this sentence and reformulated this paragraph for better clarification.

*"Line 71: Here, two hydrological post-processors, namely the Affine kernel dressing (AKD) and the evolutionary multi-objective optimization (Non-dominated sorting genetic algorithm II, NSGA-II), were explored. Compared to conventional post-processing method, such as AKD, NSGA-II opens up the opportunity of improving the forecast quality in harmony with the forecasting aims and the specific needs of end-users. Multiple objective functions (i.e., here, verifying scores) for evaluating the forecasting performances of the H-EPSs are selected to guide the optimization process."*

*Detailed comments* 9 : l71: ".. to probe this topic.." Instead of "this topic" explicitly state the topic here. I assume it is the remaining challenge mentioned in l54-56: "how to improve the human interpretation of probabilistic forecasts and the communication of integrated ensemble forecast products to end-users (e.g., operational hydrologists, water managers, local conservation authorities, stakeholders and other relevant decision makers)." But then in l71 and further it should be explained how the testing of these two post-processing methods contributes to improving interpretation and communication. Or "this topic" refers to the paragraph l64-69, which needs reformulation to be more clear, but is directed towards harmonising forecast improvement and user-specific requirements and use in decision making, in which case l71 onwards also has to explain how the comparison of these two post-processing methods is contributing to this. In the present formulation of l71-75 it seems more as if the authors are addressing the benefits of distribution-free postprocessing methods. See my suggestion above under General Comment.

*Response* :

Thank you for your questions. Yes, this topic is referred to "how to improve the human interpretation of probabilistic forecasts and the communication of integrated ensemble forecast products to end-users (e.g., operational hydrologists, water managers, local conservation authorities, stakeholders and other relevant decision makers)." We reformulated the contents in the manuscript as:

*"Line 71: Here, two hydrological post-processors, namely the Affine kernel dressing (AKD) and the evolutionary multi-objective optimization (Non-dominated sorting genetic algorithm II, NSGA-II), were explored. Compared to conventional post-processing method, such as AKD, NSGA-II opens up the opportunity of improving the forecast quality in harmony with the forecasting aims and the specific needs of end-users. Multiple objective functions (i.e., here, verifying scores) for evaluating the forecasting performances of the H-EPSs are selected to guide the optimization process. The mechanisms of these two statistical post-processing methods are completely different. However, they share one similarity from another perspective, which is they can estimate the probability density directly from the data (i.e., ensemble forecast) without assuming any particular underlying distribution. As a more conventional method, Silverman (1986) firstly proposed the kernel density smoothing method to estimate the distribution from the data by centering a kernel function K that determines the shape of a probability distribution (i.e., kernel) fitted around every data point (i.e., ensemble members). The smooth kernel estimate is then the sum of those kernels. As for the choice of bandwidth* h *of each dressing kernel, Silverman's rule of thumb finds an optimal bandwidth* h *by assuming that the data*

185

190

195

200

205

210

215

*is normally distributed. Improvements to the original idea were soon to follow. For instance, the improved Sheather Jones (ISJ) algorithm is more suitable and robust with respect to multimodality (Wand and Jones, 1994). Roulston and Smith (2003) rely on the series of "best forecasts" (i.e., best-member dressing) to compute the kernel bandwidth h. Wang and Bishop (2005) as well as Fortin et al. (2006) further improved the best member method. The later advocated that the more extreme ensemble members are more likely to be the best member of raw under-dispersive forecasts, while the central members tend to be more "precise" for over-dispersive ensemble. They proposed the idea that different predictive weights should be set over each ensemble member, given each member's rank within the ensemble. Instead of standard dressing kernels that act on individual ensemble members, Bröcker and Smith (2008) proposed the AKD method by assuming an affine mapping between ensemble members and observation over the entire ensemble. They approximate the distribution of the observation given the ensemble.*

*Given the single-model H-EPSs studied here, the hydrologic ensemble is generated by activating two forecasting tools: the ensemble weather forecasts and the EnKF. Henceforth, enhancing the H-EPS forecasting skill by assigning different credibility to ensemble members becomes preferred than reducing the number of members. The post-processing techniques, like the Non-dominated sorting genetic algorithm II (NSGA-II), are now common (e.g., Liong et al., 2001; De Vos and Rientjes, 2007; Confesor and Whittaker, 2007). Such techniques are conceptually linked to the multiobjective parameter calibration of hydrologic models using Pareto approaches. Indeed, formulating a model structure or representing the hydrologic processes using a unique global optimal parameter set proves to be very subjective. Multiple optimal parameter sets exist with satisfying behavior given the different conceptualizations, albeit not identical Beven and Binley (1992). For example, Brochero et al. (2013) utilized the Pareto fronts generated with NSGA-II for selecting the "best" ensemble from a hydrologic forecasting model with a pool of 800 streamflow predictors, in order to reduce the H-EPS complexity. **Here, the expected output of NSGA-II method is a group of solutions, also known as Pareto front, that identify the trade-offs between different objectives, subject to the end-users' needs and requirements."***

*Detailed comments* 10 : l314: ".. for each day."

*Response* :

Thank you. We corrected this writing error

*" Line 316: Yet one may decide otherwise, such as implementing the calibration/validation procedures separately **for each day."***

*Detailed comments* 11 : l320-321: A training period of 30-days with moving window is mentioned here. Please kindly clarify, including whether that applies to re-training every next day both the AKD and NSGA-II post-processor, using only the past 30-days observational and forecast data (day-4 lead time).

*Response* :

Thank you for your suggestion. We added the clarification here:

*"Line 323: Here a 30-day moving window is selected so it contains enough training samples with coherent consistency.* **Which is to say, the NSGA-II post-processors were trained using only the past 30 days and day-4 forecast data and then re-trained every next day.** *Especially, from the operational perspective, a monthly moving window is more coherent and efficient in the real world, with limited length for time series."*

*Detailed comments* 12 : l354: I do not think the heading "Uncertainty analysis" covers what is presented and discussed here. I would suggest a separate heading for the NSGAII result, e.g. "4.2: NSGA-II convergence", and then for the comparison, e.g. "4.3 AKD and NSGA-II performance comparison".

*Response* :

Thank you for your suggestions. We added the separate headings.

**"Line 353: 4.2 NSGA-II convergence"**

**"Line 365: 4.3 AKD and NSGA-II performance comparison"**

*Detailed comments* 13 : l356: remove the space in "wi thout"

*Response* :

Thank you. We corrected this writing error.

*Detailed comments* 14 : l359-361: I do not understand what is done here and why, when referring to 'random selection from the pareto front'

*Response* :

Thank you for your question.

The un-repeated nondomination Pareto solutions is a set of optimal options for users to choose after the whole evolutionary multiobjective optimization search. The solutions in the elbow region of the Pareto front are the compromise between both two objective functions. For example, in the study, the optimal $NSE$ is inevitably accompanied with the highest $bias$ (e.g., $NES$= 0.846, $bias$=0.034), or vice versa. As one representative multiobjective evolutionary search result shown in Figure 5, 35 (nondominated) Pareto solutions are identified. The solutions can be obtained daily by setting the sliding window. Specifically speaking, the NSGA-II post-processors were trained using only the past 30 days and day-4 forecast data and then re-trained every next day. Therefore, we decided to pick a random solution in the Pareto front at each time step since they were all optimal options.

*Detailed comments* 15 : l367-369: Remove. I suggest to start the new section 4.3 with "The reliability of the raw.."

*Response* :

250        Thank you. We removed these contents in the manuscript.

**References**

Beven, K. and Binley, A.: The future of distributed models: Model calibration and uncertainty prediction, Hydrol. Process., 6(3), 279-298, https://doi.org/10.1002/hyp.3360060305, 1992.

Brochero, D., Gagné, C., and Anctil, F.: Evolutionary multiobjective optimization for selecting members of an ensemble streamflow forecasting model. Proceeding of the fifteenth annual conference on Genetic and evolutionary computation conference-GECCO, New York, United States, 6 July 2013, 13, 1221-1228, https://doi.org/10.1145/2463372.2463538, 2013.

Bröcker, J. and Smith, L.A.: From ensemble forecasts to predictive distribution functions, Tellus. A., 60(4), 663-678, https://doi.org/10.1111/j.1600-0870.2007.00333.x, 2008.

Buizza, R. Asensio, H. Balint, G. Bartholmes J, et al.: EURORISK/PREVIEW report on the technical quality, functional quality and forecast value of meteorological and hydrological forecasts, ECMWF Technical Memorandum, ECMWF Research Department: Shinfield Park, Reading, United Kingdom, 516, 1-21, http://www.ecmwf.int/publications/, 2007.

Crochemore, L., Ramos, M.H., Pappenberger, F., and Perrin, C.: Seasonal streamflow forecasting by conditioning climatology with precipitation indices, Hydrol.Earth Syst. Sci., 21, 1573-1591, https://doi.org/10.5194/hess-21-1573-2017, 2017.

Confesor, Jr.R.B. and Whittaker, G.W., 2007. Automatic Calibration of Hydrologic Models With Multi-Objective Evolutionary Algorithm and Pareto Optimization 1, JAWRA Journal of the American Water Resources Association, 43(4), 981-989, https://doi.org/10.1111/j.1752-1688.2007.00080.x, 2007.

De Vos, N.J. and Rientjes, T.H.M.: Multi-objective performance comparison of an artificial neural network and a conceptual rainfall—runoff model, Hydrolog. Sci. J., 52(3), 397-413, https://doi.org/10.1029/2007WR006734, 2007.

Deb, K., Pratap, A., Agarwal, S., and Meyarivan, T.A.M.T.: A fast and elitist multiobjective genetic algorithm: NSGA-II. IEEE. T. Evolut. Comput., 6(2), 182-197, https://doi.org/10.1109/4235.996017, 2002.

Fraley, C., Raftery, A.E., and Gneiting, T.: Calibrating Multimodel Forecast Ensembles with Exchangeable and Missing Members Using Bayesian Model Averaging, Mon. Weather. Rev., 138, 190-202, https://doi.org/10.1175/2009MWR3046.1, 2010.

Fortin, V., Favre, A.C., Saïd, M., 2006. Probabilistic forecasting from ensemble prediction systems: Improving upon the best-member method by using a different weight and dressing kernel for each member, Q. J. R. Meteorol Soc., 132, 1349-1369, https://doi.org/10.1256/qj.05.167, 2006.

Gaborit, É., Anctil, F., Fortin V., and Pelletier, G.: On the reliability of spatially disaggregated global ensemble rainfall forecasts, Hydrol. Process., 27(1), 45-56. https://doi.org/10.1002/hyp, 2013.

Klemeš, V.: Operational testing of hydrological simulation models, Hydrolog. Sci. J., 31(1), 13-24, https://doi.org/10.1080/02626668609491024, 1986.

Li, W., Duan, Q., Miao, C., Ye, A., Gong, W., and Di, Z.: A review on statistical postprocessing methods for hydrometeorological ensemble forecasting, Wires. Water., 4(6), e1246. https://doi.org/10.1002/wat2.1246, 2017.

Liong, S.Y., Khu, S.T. and Chan, W.T.: Derivation of Pareto front with genetic algorithm and neural network, J. Hydrol. Eng., 6(1), 52-61, https://doi.org/10.1061/(ASCE)1084-0699(2001)6:1(52), 2001.

Perrin, C.: Vers une amélioration d'un modèle global pluie-débit, PhD diss., Institut National Polytechnique de Grenoble-INPG, Grenoble, 287 pp, 2000.

Ramos, M.H., Mathevet, T., Thielen, J. and Pappenberger, F.: Communicating uncertainty in hydro-meteorological forecasts: mission impossible? Meteorol. Appl., 17(2), pp.223-235, https://doi.org/10.1002/met.202, 2010.

Roulston, M.S. and Smith, L.A.: Combining dynamical and statistical ensembles. Tellus. A., 55, 16-30, https://doi.org/10.3402/tellusa.v55i1.12082, 2003.

290 Seiller, G., Anctil, F., and Perrin, C.: Multimodel evaluation of twenty lumped hydrological models under contrasted climate conditions, Hydrol. Earth. Syst. Sc., 16, 1171-1189, https://doi.org/DOI : 10.5194/hess-1116-1171-2012, 2012.

Silverman, B.W.: Density estimation for statistics and data analysis, CRC press, London, 26, 1986.

Thiboult, A., Seiller, G., Poncelet, C., and Anctil, F.: The HOOPLA toolbox: a HydrOlOgical Prediction LAboratory to explore ensemble rainfall-runoff modeling, arXiv [preprint], Hydrol. Earth Syst. Sci. Discuss., https://doi.org/10.5194/hess-2020-6, 28 January 2020.

295 Thielen, J., Ramos,M.H., Bartholmes, J., De Roo, A., Cloke, H., Pappenberger, F., and Demeritt, D.: Summary report of the 1st EFAS workshop on the use of Ensemble Prediction System in flood forecasting, European Report EUR, Ispra., 22118. http://floods.jrc.ec.europa.eu/efas-documents, 2005.

Valéry, A., Andréassian, V., and Perrin, C.: As simple as possible but not simpler': What is useful in a temperature-based snow-accounting routine? Part 2 - Sensitivity analysis of the Cemaneige snow accounting routine on 380 catchments, J. Hydrol., 517: 1176-1187,
300 https://doi.org/10.1016/j.jhydrol.2014.04.058, 2014.

Wang, X. and Bishop, C.H.: Improvement of ensemble reliability with a new dressing kernel, 131(607), 965-986, https://doi.org/10.1256/qj.04.120, 2005.

Wand, M.P. and Jones, M.C.: Kernel smoothing, CRC press, vol. 60, 1 December 1994.